

# Version 2 of the EUMETSAT OSI SAF and ESA CCI Sea Ice Concentration Climate Data Records

Thomas Lavergne[1], Atle Macdonald Sørensen[1], Stefan Kern[2], Rasmus Tonboe[3], Dirk Notz[4], Signe Aaboe[1], Louisa Bell[4], Gorm Dybkjær[3], Steinar Eastwood[1], Carolina Gabarro[5], Georg Heygster[6], Mari Anne Killie[1], Matilde Brandt Kreiner[3], John Lavelle[3], Roberto Saldo[7], Stein Sandven[8], Leif Toudal Pedersen[7]

[1]Research and Development Department, Norwegian Meteorological Institute, Oslo, Norway
[2]Integrated Climate Data Center, CEN, University of Hamburg, Hamburg, Germany
[3]Danish Meteorological Institute, Copenhagen, Denmark
[4]Max-Planck Institut für Meteorologie, Hamburg, Germany
[5]Barcelona Expert Center, ICM-CSIC, Spain
[6]Institute of Environmental Physics, University of Bremen, Bremen, Germany
[7]Danish Technical University-Space, Copenhagen, Denmark
[8]Nansen Environmental and Remote Sensing Center, Bergen, Norway

*Correspondence to*: Thomas Lavergne (thomas.lavergne@met.no)

**Abstract.** We introduce the OSI-450, the SICCI-25km and the SICCI-50km climate data records of gridded global sea-ice concentration. These three records are derived from passive microwave satellite data and offer three distinct advantages compared to existing records: First, all three records provide quantitative information on uncertainty and possibly applied filtering at every grid point and every time step. Second, they are based on dynamic tie points, which capture the time evolution of surface characteristics of the ice cover and accommodate potential calibration differences between satellite missions. Third, they are produced in the context of sustained services offering committed extension, documentation, traceability, and user support. The three records differ in the underlying satellite data (SMMR & SSM/I & SSMIS or AMSR-E & AMSR2), in the imaging frequency channels (37 GHz and either 6 GHz or 19 GHz), in their horizontal resolution (25 km or 50 km) and in the time period they cover. We introduce the underlying algorithms and provide an initial evaluation. We find that all three records compare well with independent estimates of sea-ice concentration both in regions with very high sea-ice concentration and in regions with very low sea-ice concentration. We hence trust that these records will prove helpful for a better understanding of the evolution of the Earth's sea-ice cover.

## 1 Introduction

Satellite-retrieved records of Arctic and Antarctic sea-ice concentration differ widely in their estimates of a specific sea-ice concentration on a given day in a given region (e.g., Ivanova et al., 2015; Comiso et al., 2017). Integrated over the entire Arctic, these differences accumulate to an up to 20 % uncertainty in the long-term trends of sea-ice extent and sea-ice area (Comiso, 2017), which hinders a robust evaluation and bias correction of climate models, and in particular hinders a robust estimate of the future evolution of the Arctic sea-ice cover. For example, Niederdrenk and Notz (2018) found that



observational uncertainty is the main source of uncertainty for estimating at which level of global warming the Arctic will lose its summer sea-ice cover. In this contribution, we introduce three new climate data records of gridded global sea-ice concentration that address some of the shortcomings of existing records, and in particular provide additional information that allow users to judge the robustness of the sea-ice concentration estimates.

Our focus on sea-ice concentration is to a substantial degree driven by the fact that information on sea-ice concentration are key to the vast majority of approaches to understand the changing sea-ice cover of our planet. This importance of sea-ice concentration derives both from the availability of a long, continuous record of the underlying passive-microwave data, and from the central importance of sea-ice concentration for many physical processes connected to the sea-ice cover. For example, the albedo of the polar oceans directly depends on sea-ice concentration (e.g., Brooks, 1924), as does much of the

heat and moisture transfer between the ocean and the atmosphere (e.g., Maykut, 1978).

Information on sea-ice concentration are also used to derive total sea-ice area or extent. The latter have in the Arctic been found to be linearly related to global-mean temperature (e.g., Gregory et al., 2002; Niederdrenk and Notz, 2018), atmospheric $CO_2$ concentration (e.g., Johannessen, 2008; Notz and Marotzke, 2012) and anthropogenic $CO_2$ emissions (Zickfeld et al., 2012; Herrington and Zickfeld, 2014; Notz and Stroeve, 2016). These linear relationships allow one to

estimate the future evolution of Arctic sea ice directly from the observational record (e.g., Notz and Stroeve, 2016; Niederdrenk and Notz, 2018), to evaluate the sea-ice evolution in coupled climate models, and to bias correct estimates from climate models for improved projections of the future sea-ice cover (e.g., Mahlstein and Knutti, 2012; Screen and Williamson, 2017; Sigmond et al., 2018). For any of these applications, the reliability of the underlying sea-ice concentration record is crucial.

This importance of a reliable sea-ice concentration record is also reflected in the definition of Sea Ice Essential Climate Variables (ECV) by the Global Climate Observing System (GCOS), a body of the World Meteorological Organisation (WMO). In their most recent update (GCOS-IP, 2016), they request that reliable observational records of sea-ice concentration must be made available to the climate research community. However, the reliability and long-term stability of existing records is often not clear. This is for example reflected by substantial differences between existing estimates of sea-

ice concentration from various algorithms (e.g., Ivanova et al., 2015; Comiso et al., 2017).

With our three new climate data records of sea-ice concentration we aim at providing the users with new reference data sets that have three clear advantages over most existing records. First, all our three records provide quantitative information on uncertainty and possibly applied filtering at every grid point and every time step. Second, they are based on dynamic tie points, which capture the time evolution of surface characteristics of the ice cover and help minimize the impact of sensor

drift and change in satellite sensor. Third, they are produced in the context of sustained services offering committed extension, documentation, traceability, and user support.

The first of our three climate data records (CDR) is referred to as OSI-450. It is based on coarse resolution passive microwave (PMW) satellite data that are available from November 1978 onwards. This data is also at the heart of the two currently most widely used sea-ice concentration algorithms, namely the NASA Team algorithm (Cavalieri et al., 1996) and



the Bootstrap algorithm (Comiso et al., 2017a). OSI-450 has been released by the European Organisation for the Exploitation of Meteorological Satellites (EUMETSAT) Ocean and Sea Ice Satellite Application Facility (OSI SAF, http://www.osi-saf.org/) and is a fully revised version of its predecessor OSI-409 (Tonboe et al., 2016).

The second and third CDRs are called SICCI-25km and SICCI-50km. They are based on medium-resolution PMW satellite
data available from June 2002 onwards. These two SICCI CDRs are released by the European Space Agency (ESA) Climate Change Initiative (CCI, http://cci.esa.int/) programme.

All three Sea Ice Concentration (SIC) CDRs share the same algorithms, processing chains, and data format. In particular, they were all developed with their primary application as climate-data records in mind, putting very narrow constraints on the permissible long-term drift of the records. As such, the underlying algorithms are based on earlier work by the European
sea-ice remote-sensing community (Anderson et al., 1997; Tonboe et al., 2016) and provide sea-ice concentration estimates with a) low sensitivity to atmospheric noise including liquid water content and water vapour, b) low sensitivity to surface noise including wind roughening of the ocean surface, and variability of sea-ice emissivity and temperature, c) the capability to adjust to the climatological changes in the above-mentioned noise sources, and d) a quantification of the remaining noise at each time step for each pixel.

Together, the three new climate-data records are a unique joint contribution of the two leading European Earth Observation agencies for addressing the requirements of the climate research community and climate information services. The three CDRs are summarized up-front in Table 1, and the satellite data used as input are in Table 2. The values entering Table 1 and Table 2 will all be introduced in the course of the paper.

In this contribution, we outline the underlying algorithms and the philosophy behind them. We also provide an initial
evaluation of the resulting climate-data records. We start in section 2 by describing the satellite and ancillary data used as input. Section 3 describes the algorithms and processing steps implemented to process the data records. Afterwards, section 4 is devoted to the resulting data records, their initial validation results, and known limitations. Discussion, outlook, and conclusions are covered in section 5.

## 2   Data

This section summarizes the satellite as well as the numerical weather prediction (NWP) data used in the climate data records. Each of these data sources are fully described in dedicated technical documentation, web resources, and scientific literature, so that we keep here only the key information directly relevant to the discussion in this paper. Figure 1 shows the temporal coverage of the data sources entering the three SIC CDRs. Two ESA CCI data records (grey box marked "ESA CCI (2x)") are based on the Advanced Microwave Scanning Radiometer - Earth Observing System (AMSR-E) and AMSR2
instruments (orange and dark orange horizontal bars), while the EUMETSAT OSI SAF data record (grey box marked "OSI SAF (OSI-450)") is based on the Scanning Multichannel Microwave Radiometer (SMMR, purple bar), Special Sensor Microwave/Imager (SSM/I, dark blue bars), and Special Sensor Microwave Imager / Sounder (SSMIS, light blue bars) instruments. ERA-Interim reanalysis weather data from the European Centre for Medium-Range Weather Forecasts




(ECMWF) are also used throughout the period (not shown). Overlap of satellite missions and the 9-months data gap between AMSR-E and AMSR2 operations are clearly visible from Figure 1. Although there always was at least one satellite mission carrying a relevant passive microwave instrument after October 1978, some few data gaps exist in the satellite data record that are too short to appear in Figure 1. The more prominent are documented in Table 2. Figure 1 also shows other related

satellite missions that do not enter the new CDRs, but might be relevant for their future extension in a compatible Interim Climate Data Record (grey box marked "OSI SAF ICDR"). They are discussed in our Outlooks.

## 2.1    Input satellite data

More details about the satellite instruments and platforms are given in Table 2. It lists the satellite platforms, sensors, and time periods for brightness temperatures ($T_B$) used as input for the SIC CDRs. Some specific instrument characteristics like

channel frequencies, spatial resolution, view angle and width of the polar observation hole are also documented there. Table 2 documents that the instrument series might have quite different characteristics (e.g. channel frequencies or incidence angle). To build a consistent data record requires methodologies to carefully inter-calibrate and tune the algorithms to yield similar results when using all these sensors. This is the essence of the dynamic tuning approaches adopted in Tonboe et al. (2016) and further developed for the new CDRs (section 3).

Building CDRs from this suite of satellite sensors is best achieved if the selected algorithms use only channels that are consistently available throughout the period. Slight changes of incidence angle or wavelengths between the sensor series can be compensated for by the algorithms, but it is harder or even impossible to achieve temporal consistency in the event of sudden loss of channels. In that respect, it is noteworthy that the 23.0 GHz channels of the SMMR instrument were highly unstable since launch, and eventually ceased to function on 11th March 1985 (Njoku et al., 1998). There is thus no

continuous data record of brightness temperatures in the vicinity of the water vapour absorption line (22.235 GHz). Such wavelength is typically needed for filtering weather effects in other SIC CDRs (e.g. Meier et al. 2017). Our algorithms do not rely on such a channel (section 3.4.2).

Although not identical, the spatial resolution of the channels needed for the SIC algorithms is somewhat similar for the three coarse resolution sensor series (SMMR, SSM/I, and SSMIS) with about 70x45 km (resp. 38x30 km) instantaneous Field-of-

View (iFoV) diameters for the 19GHz (resp. 37 GHz) channels. The two medium-resolution radiometers AMSR-E and AMSR2 have finer resolution at these channels (27x16 km and 14x9km respectively), accompanied by increased sampling (10x10km instead of 25x25km for SSM/I). It is noteworthy that iFoV diameters, as reported in Table 2 and at several online resources such as the WMO OSCAR Space-based capabilities data base are not a measure of the true footprint of an individual measured pixel. This is because the iFoV neither takes into account the motion of the antenna (scan direction) nor

of the spacecraft (along its orbit) during the integration time needed to acquire a single pixel. The effective Field-of-View (eFoV) diameters includes the two effects, and is a better measure of the true footprint of the instrument. For example, the eFoV of the SSM/I 19GHz channels is closer to 70x75km.




One of the differences between the instrument series are the width of their observation swath, and the inclination of their orbit. This translates into different extent of the polar observation hole, and no data are available for sea-ice monitoring north of 84° (SMMR), 87° (SSM/I), 89° (SSMIS) and 89.5° (AMSR-E and AMSR2).

For our data records, a newly reprocessed version of the SMMR, SSM/I, and SSMIS data into a Fundamental Climate Data
Record (FCDR) was accessed from the EUMETSAT CM-SAF (Fennig et al. 2017). The AMSR-E data we use is the NSIDC FCDR AE_L2A V003 FCDR of Ashcroft and Wentz (2013). For AMSR2, we use re-calibrated (Version 2) L1R data that we directly accessed directly from the Japanese space agency (JAXA), covering 23rd July 2012 until 15th May 2017, that is the end of the SICCI-25km and SICCI-50km CDRs. For both AMSR-E and AMSR2, the nominal resolution of the channels is used, not the resolution-matched ones. It is noteworthy that the AMSR2 data is not from an FCDR, but rather from an
archive of an operational data stream. We use the data as they are provided by JAXA, without applying extra calibration towards AMSR-E (thus unlike Meier and Ivanoff, 2017) since our algorithms do not require such stringent calibration thanks to using dynamic tuning (section 3.3).

## 2.2 ERA-Interim data

The microwave radiation emitted by the ocean and sea ice travels through the Earth atmosphere before being recorded by the
satellite sensors. Scattering, reflection, and emission in the atmosphere add or subtract contribution to the radiated signal, and challenge our ability to accurately quantify sea-ice concentration. An initial step in our processing is thus the explicit correction of atmospheric contribution to the top-of-atmosphere radiation (see section 3.4.1). For this purpose, we accessed the global 3-hourly fields from ECMWF's ERA-Interim (Dee et al., 2011). Fields of 10m wind-speed, 2m air temperature, and total column water vapour are used. The ERA-Interim re-analysis starts in January 1979 and is available throughout the
time period of our CDRs. Unavailability of ERA-Interim prior to 1979 made it impractical to use the early period with SMMR data (October to December 1978).

## 3 Algorithms and processing details

This section introduces the algorithms and some processing elements that are used in the making of the SIC CDRs. In many cases, these algorithms are evolutions from those already applied for the previous version of the EUMETSAT OSI SAF
CDR (OSI-409, Tonboe et al., 2016).

### 3.1 Overview of the processing chain

Figure 2 gives an overview of the processing chain operated to process the three CDRs. The red boxes are data (stored in data files) and the blue boxes are processing elements that apply algorithms to the data. The whole process is structured into three chains, at Level 2 (left hand side), Level 3 (middle) and Level 4 (right hand side). The input Level 1 (L1) data files
hold the fields observed by the satellite sensors at top-of-atmosphere, in satellite projection: the brightness temperatures ($T_B$) structured in swath files. The Level 2 (L2) chain transforms these into the environmental variables of interest, but still on swath projection: the SIC, its associated uncertainties and flags. The Level 3 (L3) collects the L2 data files and produces





daily composited fields of SIC, uncertainties, and flags on regularly spaced polar grids. These fields can and will typically exhibit data gaps, e.g. in case of missing satellite data. The Level 4 (L4) chain performs the necessary steps to fill the gaps, apply extra corrections and format the data files that will appear in the CDR.

The next sub-sections are devoted to giving some more details about the main features of the several algorithms involved.

Before that, we note that the L2 chain holds an iteration (marked by the "2nd iteration" grey box) similarly to the work-flow in Tonboe et al. (2016) and stemming from the developments of Andersen et al. (2006). This iteration implements two key correction schemes, the atmospheric correction algorithm at low concentration range (section 3.4.1) and a novel correction for systematic errors at high concentration range (section 3.4.3).

### 3.2   A hybrid, self-tuning, self-optimizing sea-ice concentration algorithm

A new sea-ice concentration algorithm was developed during the ESA CCI Sea Ice projects and is used for the three CDRs. It is an evolution of the algorithms used in Tonboe et al. (2016). In this section, we describe both how the algorithm is trained to $T_B$ training data sets, and how it is then applied to actual $T_B$ measurements recorded by satellite sensors. The process of selecting training $T_B$ data is covered in section 3.3.

We call the SIC algorithm a *"hybrid"* algorithm because it combines two other SIC algorithms: one that is tuned to perform

better over open water and low-concentration conditions (named $B_{OW}$ for "Best Open Water"), and one that is tuned to perform better over closed-ice and the high-concentration conditions (named $B_{CI}$ for "Best Closed Ice"). The combination equation is quite simply a linear weighted average of $B_{OW}$ and $B_{CI}$ results, where $w_{ow}$ is the "open water" weight:

$$\begin{cases} w_{OW} = 1; \text{for } B_{OW} < 0.7 \\ w_{OW} = 0; \text{for } B_{OW} > 0.9 \\ w_{OW} = 1 - \frac{B_{OW} - 0.7}{0.2}; \text{for } B_{OW} \in [0.7; 0.9] \end{cases} ; \quad SIC_{hybrid} = w_{OW} \times B_{OW} + (1 - w_{OW}) \times B_{CI} , \qquad (1)$$

OSI-409 already used such a hybrid method. It combined the Bootstrap Frequency Mode (BFM) algorithm (Comiso, 1986)

as $B_{OW}$, and the Bristol (BRI) algorithm (Smith and Barrett, 1994; Smith, 1996) as $B_{CI}$. Andersen et al. (2007) and later Ivanova et al. (2015) confirmed that BFM (*resp* BRI) was so far the published algorithm (including NASA-Team and Bootstrap) performing best at low (*resp* high) SIC conditions. Notably, BRI is more accurate at high concentration than the Bootstrap Polarization Mode (BPM) algorithm. BFM and BPM are widely used for sea-ice monitoring in what is commonly known as the "Bootstrap" algorithm (Comiso and Nishio, 2008). Smith (1996) introduces the BRI algorithm as a

generalization of the BFM and BPM algorithms. BFM computes SIC values in the (19V, 37V) $T_B$ space and BPM in the (37V, 37H) $T_B$ space. BRI uses the three-dimensional (19V, 37V, 37H) $T_B$ space, where "19V" ("19H") is a notation for "the channel with a frequency near 19GHz and with Vertical (Horizontal) polarization".

Figure 3 illustrates the functioning of the BFM algorithm. Working with SMMR data for sea-ice monitoring, Comiso (1986) recognized that the typical signature of OW (SIC=0%) $T_B$ data clusters around an averaged point location (the OW tie-point,

$H$) in the (19V, 37V) $T_B$ space. Conversely, the CI (SIC=100%) $T_B$ data clusters along a line (the consolidated ice line A-D). Comiso (1986) thus designed a SIC algorithm where isolines of constant SIC are parallel to the A-D line and pass through



the measured $T_B$ in point P. A geometric algorithm using the intersection of the (H,P) and (A,D) lines in point I returns the SIC value (in our example SIC=68%). In the same study, similar aggregation of typical $T_B$ signatures and geometric algorithm were also used in the (37V, 37H) $T_B$ space (BPM algorithm). For easing later discussion, we note here that in winter Arctic conditions, the typical multiyear sea-ice signature is to the "left" of the ice line -close to A- while first-year sea

ice and young sea ice is to the right -closer to D- (Comiso et al. 2012).

The left-hand panel of Figure 4 is originally adapted from Smith (1996) and modified with colors to describes how BFM ("Frequency scheme"), BPM ("Polarisation scheme") and BRI (Bristol algorithm) view the open water (scatter around $H$) and closed ice (scatter along the $D − A$ line) data in the three-dimensional (19V, 37V, 37H) $T_B$ space. The view direction of BRI is equivalent to projecting the $T_B$ data onto a "data plane", which Smith (1996) choses as the plane containing both the

closed ice line ($D − A$) and the open water point H. Because this particular plane offers the largest dynamic range between the closed-ice and open water signatures, Smith (1996) states it to be an optimum projection plane. This however fails to recognize that the scatter of the closed-ice points around the line, and that of open water $T_B$ samples around the point H are anisotropic in the (19V, 37V, 37H) $T_B$ space. The open water scatter has increased variance along the directions resulting from weather effects (including wind speed, cloud liquid water, water vapour, etc…) on the emissivity of water. The closed-

ice scatter also has increased variance directions, e.g. due to ice type and snow characteristics. Because of these anisotropies, the optimal projection plan will generally not be that of BRI.

Our new algorithm is a generalization of BRI. Its principle is also introduced on Figure 4 (left panel). Like in BRI we seek an optimum "data plane" on which to project the $T_B$ data, and we impose that this plane holds the closed-ice line (the $D − A$ line, supported by unit vector $u$). Vector $u$ is computed by Principal Component Analysis (PCA), and is the direction with

highest variance in the CI $T_B$ samples. Conversely to BRI, we do not impose that $H$ is onto the projection plane. We rather rotate the plane around $u$, and seek the optimum rotation angle θ that yields best SIC accuracy. On Figure 4 (left panel), we mark three unit vectors $v$, corresponding to three different rotation angles and thus projection planes. By convention, we define that θ = 0° defines the BFM (19V, 37V) plane, and θ = +90° defines the BPM (37V, 37H) plane. The BRI plane typical has values around θ = +30°. By varying θ the optimization process samples several planes and eventually returns the

optimal angles $θ_{OW}$ and $θ_{CI}$ that define respectively the $B_{OW}$ and $B_{CI}$ algorithms. This optimization step allows us to cope for the anisotropy of the OW and CI $T_B$ samples in the (19V, 37V, 37H) $T_B$ space. The right-hand side panel of Figure 4 shows the process of such an optimization in a case using AMSR2 data from the Northern Hemisphere. The solid lines plot the variation of the accuracy (measured as standard deviation of SIC, on the y-axis) of the SIC algorithms defined by the rotation angle (x-axis) against the OW (blue) and CI (red) training $T_B$ data. The minimum of the blue and red curves are not achieved

at the same angle. This is a clear illustration that there cannot be a single SIC algorithm that performs best both on low-concentration and high-concentration conditions and confirms the strategy already adopted by Comiso (1986), Andersen et al. (2007) and Tonboe et al. (2016) to construct hybrid algorithms.



Still on Figure 4 (right panel), we can see that the optimum rotation angle for OW cases is generally not exactly at $\theta = 0°$ (BFM). Likewise, the optimum rotation angle for CI cases is generally not the same as that corresponding to the BRI plane. $\theta_{OW}$ (blue disc) and $\theta_{CI}$ (red disc) thus indeed define more accurate algorithms than BFM and BRI. In that particular example, the improvement is mostly for OW conditions and limited for CI conditions. The value of $\theta_{OW}$ and $\theta_{CI}$ will vary

with the exact frequencies, calibration, or viewing angle of the instrument (Table 2), as well as with the OW and CI signatures that exhibit regional, seasonal and inter-annual variations. The new hybrid, self-optimizing algorithms described in this section can always be tuned to available training data (see section 3.3) and deliver optimum and time-consistent performance.

We can draw some additional information from the right-hand side panel of Figure 4. First, we seem to confirm the findings

of Smith (1996) that BRI performs better than BPM (that corresponds to $\theta = +90°$). Indeed, the red curve increases all the way to $\theta = +90°$ and shows poor algorithm accuracy for the (37V, 37H) projection plane. Second, we observe that both the blue and red curves hit a maximum standard-deviation (minimum accuracy) somewhere around $\theta = -60°$ (the peak value is outside the y-range of the plot). This quite simply corresponds to the worst possible choice of projection plane, for which the OW $T_B$ data are projected onto the CI ice line, resulting in the smallest dynamic range between OW and CI signatures.

The geometric descriptions above were all carried out in a (19V, 37V, 37H) space. The same reasoning can however be carried within other 3D $T_B$ spaces, as long as such space offer a clustering of the CI conditions along an ice line, and sufficient dynamic range between the OW signature and the CI line. In the new CDRs, we use two different $T_B$ spaces: The OSI-450 and SICCI-25km CDRs use the (19V, 37V, 37H) space, while the SICCI-50km CDR uses the (6V, 37V, 37H) space. Both $T_B$ spaces feature two « higher frequency » channels with same wavelength but alternate polarization (37 GHz in

both cases), and a « lower frequency » vertically polarized channel (19V or 6V). The role of the "higher" frequencies is to ensure a significant spread of the CI $T_B$ samples along the ice line, and thus offer a good base for computing vector $u$ with PCA. They also bring higher spatial resolution to the retrieved SIC, since higher frequency channels achieve higher spatial resolution (Table 2). The role of the "lower" vertically polarized channel is to ensure sufficient dynamic range between OW and CI signatures, and thus aim at reducing retrieval noise. This is at the cost of bringing coarser spatial resolution into the

algorithm.

This section so far covered how the new algorithms are designed and tuned to training data. At the end of the tuning process, the unit vector $u$ defining the closed-ice line, the two angles $\theta_{OW}$ and $\theta_{CI}$, and the $T_B$ coordinates of the OW and CI mean tie-points are recorded and stored to disk for later use. These values are the tuned parameters needed to apply the algorithms. To apply the algorithm to a set of new $T_B$ data (e.g. a new swath of instrument data) is then straightforward. Each $T_B$ triplet -

(19V, 37V, 37H) or (6V, 37V, 37H) - is projected onto the two optimal planes (defined by $u$ and each of the $\theta$ angles), and a BFM-like geometric SIC algorithm is applied in both planes (like in Figure 3 but the x-axis and y-axis are now along directions in the projection plane), yielding two values : $SIC_{BOW}$ and $SIC_{BCI}$. The two SIC values are combined by Eq. 1 to yield the final SIC estimate.



### 3.3 Dynamical tuning of the SIC algorithm

As described in the previous section, tuning the algorithms requires two sets of training data, one from OW areas (SIC=0%), and one from areas we assume have fully CI cover (SIC=100%). As in Tonboe et al. (2016), the training of the algorithms is performed separately for each instrument, and for each hemisphere. In addition, the training is updated for every day of the data record, and is based on [-7;+7 days] worth of daily samples (where Tonboe et al. 2016 used a [-15;+15 days] sliding window).

The dynamic training of our algorithms allows us to a) adapt to inter-season and inter-annual variations of the sea-ice and open water emissivity, b) cope with different calibration of different instruments in a series, or between different FCDRs, c) cope with slightly different frequencies between different instruments (e.g. SMMR, SSM/I, and AMSR-E all have a different frequency around 19 and 37 GHz, see Table 2), d) mitigate sensor drift (if not already mitigated in the FCDR), e) compensate for trends potentially arising from use of NWP re-analysed data to correct the $T_B$ (see section 3.4.1).

As in Tonboe et al. (2016), the CI training sample is based on the results of the NASA Team (NT) algorithm (Cavalieri et al., 1984). It is assumed that locations for which the NT value is greater than 95% are in fact mostly a representation of 100 % ice (Kwok, 2002). The tie-points for applying the NT algorithm are taken from Appendix A in Ivanova et al. (2015). The same tie-points are used for AMSR2 (not covered by Ivanova et al., 2015) than as for AMSR-E. Recent investigations, e.g. during the ESA CCI Sea Ice projects documented that NT is an acceptable choice for the purpose of selecting closed-ice samples all-year-round, even in the summer melt season (Kern et al. 2016). To ensure temporal consistency between the SMMR and later instruments, the closed-ice samples for NH are only used for algorithm tuning if their latitude is less than 84°N, which is the limit of the SMMR polar observation hole (Table 2).

The selection of the OW tie-point samples was revised since Tonboe et al. (2016) which used fixed ocean areas at mid-high latitudes. The training areas now varies on a monthly basis, and follows sea-ice cover more closely. In practice, the OW locations are those falling in a 150km wide belt just outside the monthly varying maximum ice extent climatology (which is itself described in section 3.6).

### 3.4 Strategies to further reduce systematic errors and random noise

The algorithms described in section 3.2 are self-optimizing to yield highest accuracy at high and low concentration ranges. Nevertheless, all $T_B$ triplets with a departure from the mean CI or OW signatures will yield departure from 0% and 100% sea-ice concentration. Random departure that do not have apparent spatial or temporal structures are often referred to as *random noise*, while departure that are somewhat stable (correlated) in space and time are referred to as *systematic errors*. Analysis of time-series of sea-ice concentration maps retrieved from the algorithm from section 3.2 reveal that the departure at low concentration range (open water) is typically a random noise, while more systematic errors are observed at high concentration range (closed ice). This is explained by the different nature of the error sources playing a role at these two ends of the sea-ice concentration range: weather-related effects at synoptic scales over open water, and surface emissivity variability (due to ice type, temperature of the emission layer, snow depth, etc…) over closed ice. In this section, we describe



strategies implemented in the processing chain to further reduce random noise over open water, and systematic errors over closed ice. Both correction steps are applied during the second iteration of the L2 chain (Figure 2) and we note $SIC_{ucorr}$ (uncorrected) the uncorrected SIC value before the start of the second iteration.

### 3.4.1 Radiative Transfer Modelling for correcting atmosphere influence on brightness temperatures

As described in Andersen et al. (2006) and confirmed in Ivanova et al. (2015), the accuracy of retrieved sea-ice concentration can be greatly improved when the brightness temperatures are corrected for atmospheric contribution by using a Radiative Transfer Model (RTM) combined with surface and atmosphere fields from NWP re-analysis. The correction using NWP data is only possible in combination with a dynamical tuning of the tie-points, so that trends from the NWP model are not introduced into the sea-ice concentration dataset. The correction scheme implemented in the new CDRs is

based on a "double-difference" scheme, similar (but not identical) to that described in Andersen et al. (2006) or Tonboe et al. (2016).

The scheme evaluates the correction offsets $\delta T_B$ (one per channel), the difference between two runs of the RTM: $T_{Bnwp}$ uses estimates from NWP fields (in our case ERA-Interim), while $T_{Bref}$ uses a reference atmospheric state with the same air temperature as $T_{Bnwp}$, but zero wind, zero water vapour, and zero cloud liquid water. $\delta T_B$ is thus an estimate of the

atmospheric contribution at the time and location of the observation.

$$\text{Tb}_{nwp} = F(W_{nwp}, V_{nwp}, L_{nwp} = 0; T_s, \text{SIC}_{ucorr}, \theta_0)$$
$$\text{Tb}_{ref} = F(0,0,0; T_s, \text{SIC}_{ucorr}, \theta_{instr})$$
$$\delta\text{Tb} = \text{Tb}_{nwp} - \text{Tb}_{ref}$$
$$\text{Tb}_{corr} = \text{Tb} - \delta\text{Tb}, \qquad\qquad (2)$$

The RTM function F simulates the brightness temperature emitted at view angle $\theta_0$ by a partially ice-covered scene with sea-ice concentration SIC, and with surface and atmospheric states described by $W_{nwp}$ (10m wind-speed, m.s⁻¹), $V_{nwp}$ (total columnar water vapour, mm), $L_{nwp}$ (total columnar liquid water content, mm), and Ts (2m air temperature). $\theta_{instr}$ is the nominal incidence angle of the instrument series (see Table 2). Our double-difference scheme is thus both a correction for the atmosphere influence on the $T_B$ (as predicted by the NWP fields) and a correction to a nominal incidence angle. The

latter is required for stabilizing the SSM/I F10 signal, whose view angle varied significantly: the peak-to-peak daily average incidence angle variation due to the platform's orbital drift was 52.6°–53.7° for F10 according to Colton and Poe (1999). The typical values of $\delta T_B$ range from about 10 K over open water to few tenths of a Kelvin over consolidated sea-ice. The liquid water content (L) fields from global NWP fields (and ERA-Interim in particular) were found to not be accurate enough for being used in our atmospheric correction scheme (Lu et al. 2018). The $T_B$ are thus not corrected for L (L=0 in

both $T_{Bnwp}$ and $T_{Bref}$), and the induced remaining noise transfers into uncertainty in SIC.





We use the Remote Sensing Systems (RSS) RTM documented in Wentz (1983) for SMMR, Wentz (1997) for SSM/I and SSMIS, and Wentz and Meissner (2000) for AMSR-E and AMSR2. It is a parametrized, fast RTM optimized for the frequencies and view angles covered by the passive microwave sensors at hand. It originally allows ocean and atmosphere simulations, and was later extended to cover sea ice surface conditions (Andersen et al, 2006). Since the RTM is used in the double-difference scheme described above, accurate calibration of the RTM simulation with the measured brightness temperatures is not critical since such offsets cancel out.

### 3.4.2  Open Water Filtering

The Weather Filters (WFs) of Cavalieri et al. (1992) have been used in basically all available SIC CDRs except the earlier EUMETSAT OSI SAF ones (Andersen et al, 2007, Tonboe et al. 2016). Weather filters are algorithms that combine $T_B$ channels to detect when rather large SIC values are in fact noise due to atmospheric influence (mainly wind, water vapour, cloud liquid water effects), and should be reported as open water (SIC=0%). The concept of WFs is very different from the atmospheric correction of $T_B$ described in the previous section: the atmospheric correction reduces noise in the resulting SIC fields (but does not yield exactly SIC=0% over open water) while the WF is a binary test to decide if a pixel should be set to exactly SIC=0% or left un-affected. In the new CDRs, we combine both approaches as we apply the WFs after the atmospheric correction.

While WFs are effective at removing false sea ice in open water regions, they will always falsely remove (detect as open water) some amount of low concentration (and/or thin) sea ice, especially along the ice edge (Ivanova et al. 2015). This is why the OSI SAF SIC CDRs so far did not adopt WFs and why the effect of WFs can be fully reverted from our new SIC CDRs by using ad-hoc status flags in the product files (see section 4.1).

The Weather Filter by Cavalieri et al. (1992) detects as open water (and consequently forces SIC to 0%) all observations with either GR3719v > 0.050 and/or GR2219v > 0.045. The GR notation stands for Gradient Ratio and this quantity is computed as e.g. GR3719v = $(T_B37v-T_B19v)/(T_B37v+T_B19v)$. Many investigators re-used these thresholds unchanged, while they should really be adapted to the different wavelengths and calibration of the different instruments. Spreen et al. (2008) adapted the GR3719v threshold to 0.045 and GR2219v to 0.040 when processing sea-ice concentration with AMSR-E data. The NOAA/NSIDC Sea Ice Concentration CDR uses the Cavalieri et al. (1992) thresholds, to the exception of Southern Hemisphere processing with SSMIS F17 for which the GR3719v threshold is 0.053 (Algorithm Theoretical Basis Document for Meier et al. 2017).

Following Lu et al. (2018), we use a WF computed from $T_B$ that have been corrected for atmospheric influence, and features a test on GR3719v only. There are two reasons for not using GR2219v: 1) a near 22 GHz channel is not available throughout the satellite time-series (section 2.1); and 2) the correction of water vapour using ERA-Interim data is effective enough in polar regions so that very limited additional screening is triggered by GR2219v when applied after $T_B$ correction. Indeed, GR2219v is mostly effective at detecting water vapour effects, while GR3719v is effective at screening cloud liquid water and wind roughening effects (Cavalieri et al, 1995).





The functioning of the WF is illustrated on Figure 3. In the (19V, 37V) diagram of Figure 3, the GR3719v=$T$ isolines are steeper than the consolidated ice line (A-D). For selected values of $T$, the isoline intersect the regions of typical open water and low concentration ice (the solid blue isoline GR3719v=0.058 is plotted as an illustration). All $T_B$ data falling below the GR3719v isoline will result in GR3719v>$T$ and will thus be flagged as OW (SIC=0%) by the GR3719v test. Most of the OW

$T_B$ data (grey triangle symbols) are thus flagged as OW, as expected. Some low-concentration $T_B$ data (not shown, but falling between H and (A-D), closer to H will also be detected as OW by the GR3719v test. This is an illustration how WFs based on this gradient ratio will not only successfully detect false sea ice as open water, but also wrongly result into ice-free conditions where some true sea ice should have been observed. The greediness of the GR3719v filter is controlled by the threshold $T$, whose tuning is of paramount importance for the temporal consistency of the climate data record. The varying

signature of sea-ice and ocean emissivity with time and hemisphere, the different frequencies of the 19 and 37 GHz channels for different instruments, the varying effects of atmospheric correction all prevent the adoption of fixed thresholds. Instead, we adopt a dynamic approach to tune the threshold. Our WF is tuned to not remove true ice with concentration larger than 10%, on average as show on Figure 3. First, the coordinates for the point $J$ are computed: $J$ falls where the SIC=10% isoline crosses the line between the OW signature point $H$ and a point at the right-most end of the line A-D. Then, the GR3719v

value corresponding to $J$ is computed, and used as a threshold $T$. Since the exact location of H, A, and D vary for each instrument, hemisphere, and day in the data record, our threshold T will change (although slightly so) during the whole data record, without the need for prescribed values (such as T=0.05 for the Cavalieri 1992 WF). The value of 10% SIC is chosen to be below the threshold commonly used for defining sea-ice extent (15% SIC), to ensure that the Weather Filter does not interfere when computing sea-ice extent.

We note finally that the naming "Weather Filter" can be mis-leading as the non-expert could understand that it is meant for filtering out weather effects (false sea ice) from "calm" open water and low ice concentration conditions. As seen in Figure 3, this is not how the GR3719v filter works, as it is set to remove true sea ice as well, even in "calm" weather conditions (OW samples below $J$). For this reason, we rather refer to such filters as "Open Water Filter" (OWF) and add a test on the SIC value. The OWF implemented in the new CDRs are thus defined by the following two tests (corresponding to the thick

solid blue line in Figure 3):

$$\begin{cases} \text{GR3719v} \geq T \\ \text{or SIC} \leq 10\% \end{cases},\qquad (3)$$

Noticeably, we compute OWFs in swath projection, in the Level-2 chain (Figure 2). As a result, each FoV observation at Level-2 is attached a binary flag as to if the OWF detected it as "probably" open water or not. This binary flag is combined during gridding and daily averaging to yield Level-3 fields of OWFs. This is a better approach than computing WFs from

daily averaged gridded $T_B$ data which will smooth and smear rapidly changing weather effects such as cloud liquid water content or wind roughening. The impact of the dynamic tuning of the OWF is evaluated in section 4.2.1.





### 3.4.3 Reducing systematic errors at high concentration range

Winter-time, monthly averaged maps of $SIC_{ucorr}$ exhibit systematic errors at high concentration range, especially visible in the central Arctic Ocean. A novel correction scheme is implemented as part of the second algorithm iteration (Figure 2) that effectively mitigates most of these systematic errors over the basin.

By construction, SIC algorithms BFM, BPM, BRI, and our new dynamic algorithms likewise, consider that the SIC is exactly 100% when the input $T_B$ fall on the consolidated ice line (Figure 3). The concept of an ice line has sustained the development of SIC algorithms for decades, since it allows algorithms to return SICs close to 100% for all consolidated ice conditions, whatever the type of sea ice (multiyear ice, first-year ice, mixture of types). However, careful analysis of the spread of consolidated ice samples along the ice line reveals that systematic deviations exist and are stable with time. These

deviations best appear in a coordinate system whose abscissae are computed as $u.T$ (dot product of $u$ the unit vector sustaining the consolidated ice line, and T a 3D $T_B$ triplet in $T_{CI}$, Figure 4) and the ordinate as $B_{CI}(T)$ (the result of the "best ice" SIC algorithm for a given $T_B$ triplet). We refer to the quantity $u.T$ as the Distance Along the ice Line (DAL). Since $u$ points from multiyear ice to first-year sea ice (section 3.2 , and Figure 4), older ice have lower DAL values than younger ice. In winter Arctic conditions, it is typical to observe that $B_{CI}(T)$ values are constantly lower than 100% (down to 85-90%) for

old ice (low values of DAL), and constantly higher than 100% (up to 105-110%) for new and first-year ice (high values of DAL). In between these two extremes, the $B_{CI}(T)$ values oscillate between being below and over the SIC=100% line. Our novel correction scheme moves the concept of an ice line to an ice curve, that more closely follow the $B_{CI}(T)$ samples along the $u$ axis. A new ice curve is tabulated for each day in the record by binning the $B_{CI}(T)$ values by their DAL values. This consolidated ice curve defines the SIC 100% isoline during the second iteration of Level 2.

Figure 5 (left panel) shows the spatial distribution of the average correction for January 2015 (SIC minus $SIC_{ucorr}$). The regional patterns of the correction are clearly visible and seem to match variations in sea-ice age: large positive correction (increase SIC) north of the Canadian Arctic Archipelago and Ellesmere island (intense red color) where the ice is oldest in the Arctic, moderate negative correction over a large part of the central basin (extending from the central Beaufort Sea, over to North Pole, and to Northern Greenland, light blue color, second year ice) and a slightly positive correction again over

large parts of the Siberian Arctic (light red color, first year ice). The mean January 2015 DAL is shown on Figure 5 (right) (blue-green-yellow color shade).

On Figure 5 (right) we observe an overall increase of the DAL value from the Canadian Arctic Archipelago (multiyear ice) across the pole and towards Laptev and Kara Sea (first-year ice). To confirm the link between DAL and sea-ice age, we overlay contours >= 1 year, >= 2 years old, and >= 3 years old sea ice from Korosov et al. (2018) on right panel. Korosov et

al. (2018) developed an improved Lagrangian-based sea-ice age tracking algorithm using the sea-ice drift product of the EUMETSAT OSI SAF (Lavergne et al., 2010). The correspondence in the transitions of DAL values with the contour lines of sea-ice age is very good, indicating that a combination of DAL (right panel) and ice curve correction (left panel) could be used for sea-ice type (if not age) classification studies. This is outside the scope of our study which is focused on SIC algorithms and the new data records.



Figure 5 (center) shows the result of the ice curve correction averaged for January 2015. It plots the difference between the variability (standard deviation) of the un-corrected SIC values (SIC$_{ucorr}$) and that of the SIC after correction. Black solid lines show the mean sea-ice edge region (at 15% and 70% SIC values). In the regions covered with sea-ice (>= 70% SIC), the shades of light blue indicate that the variability at high concentration is rather consistently reduced by about 1-2% SIC by the ice curve correction. A limited number of regions show no improvement (white color) or slight degradation. This reduction of the variability comes in addition to the correction for the systematic errors (e.g. underestimation north for Canadian Arctic Archipelago, see panel a) for which the ice curve correction was designed. In the open water regions (<= 15%) the reduction of variability is even larger (3-4% SIC). This reduction is the result of the atmospheric correction step, that what described in section 3.4.1, and has most impact over open water (Andersen et al. 2006). The increased variability (red tones) between the 15% and 70% isolines follows logically from the two above mentioned reductions: the corrections enable more accurate retrievals of SICs, thus the ice edge is more sharply defined in the daily SIC fields, and this results in higher variability on a monthly basis.

In this section, we described the strategies we implemented to improve the accuracy of the SIC algorithms. In the next section, we discuss how the remaining noise is quantified and reported to the users of the data records in the form of uncertainties.

### 3.5 Uncertainties

Spatially and temporally varying uncertainty estimates for each and every SIC value are required of state-of-the-art CDRs (GCOS-IP, 2016). Uncertainties are needed as soon as the data are compared to other sources (e.g. other similar data records), or when data is assimilated into numerical models. However, there is no unique way to derive nor to present uncertainties in EO data (Merchant et al., 2017).

The approach to derive and present uncertainties in the new SIC CDRs is mostly similar to those of Tonboe et al. (2016): we make the assumption that the total uncertainty $\sigma_{tot}$ is given by two uncertainty components, i.e.:

$$\sigma_{tot}^2 = \sigma_{algo}^2 + \sigma_{smear}^2, \qquad (4)$$

where $\sigma_{algo}$ is the inherent uncertainty of the SIC algorithm (algorithm uncertainty) including sensor noise and the residual geophysical noise quantified as variability around the OW and CI mean signatures, and $\sigma_{smear}$ is the representativeness uncertainty due to resampling from satellite swath to a grid (smearing uncertainty) and footprint mismatch.

The derivation of $\sigma_{algo}$ is to a large extent similar to that described in Tonboe et al. (2016). This term is derived from the accuracy (estimated as statistical variance) of the algorithm to retrieve 0% (*resp* 100%) when applied onto the OW (*resp* CI) training data samples (section 3.3). This uncertainty term is computed at Level 2 (Figure 2). Each Level 2 SIC estimate in the data record has an associated $\sigma_{algo}$ value.

The uncertainty term $\sigma_{smear}$ is a representativeness uncertainty. It measures the increase of uncertainty due to mismatching spatial dimensions such as when a) the satellite sensor footprint potentially covers a larger area than that of a target grid cell, or when b) the imaging channels used by the SIC algorithms do not have the same FoV diameter. Table 2 lists the



dimensions relevant to discuss these two effects. Effect a) : the size of the 3 dB footprint of the 19 GHz channels of the SMMR, SSM/I, and SSMIS instruments is larger than the resolution of the grid used to present the SIC field (25x25 km, see Table 1). Effect b) : the 3 dB footprint of the 37 GHz channels is smaller than that of the 19 GHz ones, so that the two frequencies entering the SIC algorithms do not cover the same area of Earth surface. Intuitively, both effects should have no

or limited impact where the sea ice cover is homogeneous (fully consolidated sea ice, or open water). It should be at a maximum where sharp spatial gradients occur, typically at the sea-ice edge. The smearing contribution $\sigma_{smear}$ is difficult to derive analytically and we carry on the approach of Tonboe et al. (2016) that is to parametrize $\sigma_{smear}$ as a function of a proxy. For the three new CDRs we parametrize $\sigma_{smear}$ as a function of the $(MAX-MIN)_{3x3}$ value, that is the difference between the highest and lowest SIC value in a 3x3 grid cells neighbourhood around each location in the grid. Specifically:

$$\sigma_{smear} = K \times (MAX - MIN)_{3\times3}, \tag{5}$$

where $K$ is a scalar whose value depends on the FoV diameter of the instrument channels used for the SIC computation, and the spatial spacing of the target grid. Several other proxies for the local variability of the SIC field (among others the 3x3 standard deviation, the Laplacian, power-to-mean-ratio...) were tested and this one was selected for its simplicity and robustness. Values of K were tuned using a foot-print simulator and selected cloud-free scenes of the marginal ice zone

imaged by the Moderate-Resolution Imaging Spectroradiometer (MODIS) as described in Tonboe et al. (2016). A value of $K$=1 was found to yield good results for all three CDRs. The value for $\sigma_{smear}$ is computed as part of the Level 3 chain (Figure 2), after gridding and daily averaging. The total uncertainty $\sigma_{tot}$ is finally computed using Eq. 4. In the data files, both the total, the algorithm, and the smearing uncertainty fields are made available.

### 3.6    Other relevant algorithms and processing steps

This section shortly introduces some other algorithms and processing steps that are important to the generation of the data records, but are either less critical for prospective users of the data, or presenting less evolution since Tonboe et al. (2016).

Due to the coarse resolution of the sensors used, especially SMMR, SSM/I, and SSMIS (Table 2), the $T_B$ data are influenced by land emissivity several tens of km away from the coastline. The emissivity of land is comparable to sea-ice emissivity and much higher than water emissivity. This means that sea-ice concentration will be consistently overestimated in coastal

regions. In Tonboe et al. (2016), a statistical method similar to Cavalieri et al. (1999) was implemented as a post-processing to the daily-gridded sea-ice concentration maps. Such a method showed limitation and the new SIC CDRs now introduce explicit land spill-over correction of the $T_B$ at all used channels, and on swath projection. The correction algorithm is described in details in Maass and Kaleschke (2010). The basic principle is that a fine-resolution land mask is used together with the antenna viewing geometry to estimate (and correct for) the simulated contribution of land emissivity to the observed

$T_B$. The algorithm of Maass and Kaleschke (2010) was adopted with some modification and tuning, among others: a) the computation of the fraction of land in each FoV is computed in the view geometry of the antenna (not after projection to a map), b) the antenna pattern functions are approximated as Gaussian (Normal distribution) shapes indexed on the aperture angle from central view direction, instead of distance on a projection plane. At the end of this step, $T_B$ of FoV that overlap




land and ocean are corrected for contribution by land, and can enter the Level 2 sea-ice concentration algorithms. Note that although this swath-based correction step is quite efficient at reducing land spill-over contamination, a statistical method similar to that of Cavalieri et al. (1999) still had to be applied at Level 3.

The land masks and climatology for the new SIC CDRs were revised since Tonboe et al. (2016). New land masks for the target 25x25km grids (one for NH and one for SH) where computed based on the Operational Sea Surface Temperature and Sea Ice Analysis (OSTIA) 0.05x0.05° land mask (Donlon et al., 2011). This mask was re-used in the ESA CCI Sea Surface Temperature (SST) L4 data records and was selected as input mask for the new SIC CDRs to increase cross-ECV consistency. The masks are tuned to closely match that of the NSIDC SIC CDR (the NSIDC "SSM/I" 25km Polar Stereographic mask). On average, in the NH, this corresponds to setting all 25x25km grid cells with a fraction of land lower than 30% to water (and these cells can thus potentially be covered with sea ice). There is no right or wrong binary land masks at such coarse resolution, and the choice of tuning to the NSIDC SIC CDR land mask is to help intercomparison of data records. By the same token, the monthly varying maximum sea-ice extent climatology implemented in Meier et al. (2017) were used as a base for our own climatology. The modifications included manual editing of some single pixel based on US National Ice Center, Canadian Ice Service, and Norwegian Ice Service ice charts (e.g. along the coast of Northern Norway, for some summer months in the vicinity of New Scotland, etc…). The climatology of peripheral seas and large fresh water bodies (e.g. Bohai and Northern Yellow Seas, Great Lakes, Caspian Sea, Sea of Azov, etc...) was also revisited. The cleaned climatologies were then expanded with a buffer zone of 150km in the NH and 250km in the SH. The larger expansion in SH is to cope with the positive trends in SH sea-ice extent (Hobbs et al., 2016). The expanded monthly sea-ice climatology is used both for masking of the final product and for defining the monthly varying area where to select the Open Water training samples (section 3.3).

As described in the sections above, all the geophysical processing is performed on swath projection (Level 2 processing). Gridding and daily averaging of the swath data is tackled as an initial step of the Level 3 chain (Figure 2). The methodology is mostly similar to that of Tonboe et al. (2016) as swath data from all available instruments of similar spatial resolution are combined into daily maps of the NH and SH polar regions. It is noteworthy that full advantage of the overlap of satellite missions (see Figure 1 and Table 2) was taken in order to reduce as much as possible the occurrence of missing data areas in the daily composited fields. This is conversely to the SIC CDR of Meier et al. (2017) that uses one SSM/I or SSMIS sensor at a time.

Despite using all the sensors, some data gaps still appear in the daily SIC maps, especially in the early part of the data record (late 1970s to mid-1990s). These data gaps are filled by interpolation (both spatially and temporally) to yield a more user-friendly data record. The polar observation gap (largest for SMMR and SSM/I, see Table 2) is filled by interpolation as well. All interpolation of missing data is performed with basic isotropic schemes, and no model data, nor advanced methods (among others Strong and Golden, 2016) were implemented. All interpolated data are clearly marked in the product files using status flags.





## 4    The resulting data records and their initial evaluation

### 4.1    The data records and selected examples

The SIC CDR released by the EUMETSAT OSI SAF (OSI-450) extends from January 1979 throughout December 2015. It uses data from SMMR, all SSM/I (F08, F10, F11, F13, F14, F15), and three SSMIS (F16, F17, and F18). It is delivered on

two Equal Area Scalable Earth 2 (EASE2) grids with 25x25km spacing (Brodznick et al. 2012 and 2014), one for the Northern and one for the Southern Hemisphere. SMMR data for the period October to December 1978 are not included in the CDR because of the unavailability of ERA-Interim data (section 2.2). OSI-450 has the following Digital Object Identifier (DOI): 10.15770/EUM_SAF_OSI_0008 and data is freely available to any users from the EUMETSAT OSI SAF web pages (http://www.osi-saf.org/).

The two SIC CDRs released by the ESA CCI Sea Ice project (SICCI-25km and SICCI-50km) extend over two disjointed periods and process data from AMSR-E (June 2002 to October 2011) and AMSR2 (July 2012 to May 2017). SICCI-25km (DOI: 10.5285/f17f146a31b14dfd960cde0874236ee5) is delivered on the same EASE2 25x25km grids as the OSI SAF CDR. SICCI-50km (DOI: 10.5285/5f75fcb0c58740d99b07953797bc041e) is delivered on an EASE2 50x50km grid, whose cells exactly cover four 25x25km cells of SICCI-25km and OSI-450 grids. Both SICCI-25km and SICCI-50km are freely

available to any user from the ESA CCI Data Portal (http://cci.esa.int/data/). Figure 6 shows the OSI-450 (top left panel), SICCI-25km (top right) and SICCI-50km (bottom left) SIC fields over the Weddell Sea region on 25th September 2015. The two SIC fields on the top row are rather similar except in the Marginal Ice Zone where the better resolution of the AMSR2 instrument (SICCI-25km) with respect to that of the SSMIS (OSI-450) leads to resolving finer details. The SICCI-50km SIC has increased granularity due to the lower resolution of the 6 GHz channels wrt to 19 GHz.

All three data records share the same data format, which is Network Common Data Format (NetCDF) version 4 (classic format). Files abide by the Climate and Forecast (CF) convention (CF-1.6) and the Attribute Convention for Data Discovery (ACDD-1.3). The variables inside the file enable a flexible use of the data. The main variable is named `ice_conc` and holds a SIC field where all the filters (among others the Open Water Filter, section 3.4.2) and correction steps (among others the statistical coastal correction scheme, section 3.6) are applied. This is the entry point for most prospective users of these

new SIC CDRs and is the variable plotted in the top row and bottom left panel of Figure 6. In addition, a variable named `raw_ice_conc_values` gives access to the original ("raw") values of sea-ice concentration, before filtering is applied. Bottom right panel in Figure 6 shows the content of variable `raw_ice_conc_values` from the OSI-450 CDR on the same date and location as the three other panels. A blue-yellow-red color scale is used for the low-range of SIC values. Both negative (blue) and positive (red) values appear that corresponds to the intrinsic retrieval noise level of the SIC algorithm

before the OWF is applied. All these values are indeed set to exactly 0% by the OWF in variable `ice_conc`. Note how the belt of low SIC values is bordered by a dark red region. This is very probably true low-concentration or thin sea ice that is removed by the OWF at the marginal ice zone. Removal of true sea ice by OWF was discussed in section 3.4.2. Still on the bottom right panel, a yellow-green color scale is used to plot large off-range SIC>100% values, as nominally returned by the



SIC algorithm. These raw values are non-physical (like the blue-shaded SIC<0% values) and are set to exactly 100% in variable `ice_conc`. They might be interesting for advanced users interested in accessing the full Probability Distribution Function (PDF) of retrieved SIC values, for example for Data Assimilation (DA) applications. The off-range SIC values are also needed to compute temporal averages (e.g. monthly means) to avoid introducing biases if only SIC>=0% or SIC=<100% values enter the averaging.

In each file, a `status_flag` variable indicate which flags (OWF, maximum extent climatology,…) or corrective steps (land spill-over correction) were applied in each grid cell.

Example fields of uncertainties from the OSI-450 CDR are shown in Figure 7. The two uncertainty components $\sigma_{algo}$ (left panel), and the smearing uncertainty $\sigma_{smear}$ (center), as well as the total uncertainty $\sigma_{tot}$ (right) are shown. The algorithm uncertainty is typically between 2 and 3 % SIC. It is lower for sea ice than for open water because the global variability of closed sea ice is lower than the SIC variability over open water. It is noted that this variability is not due to real SIC variability but rather to ice and open water signature variability reflected in the estimated SIC, thus an uncertainty. The smearing uncertainty is largest, up to 40% SIC, at the ice edge and low, near 0% SIC, in areas where all contributing satellite footprints are covered by the same SIC (e.g. open water). The total uncertainty, which is the sum (in variance) of $\sigma_{algo}$ and $\sigma_{smear}$ (section 3.5) is dominated by $\sigma_{smear}$. The patterns seen in Figure 7 are representative of the uncertainties of all three CDRs, for both hemispheres, during winter.

## 4.2 Initial evaluation results

The evaluation of a CDR needs to cover several aspects. One is to demonstrate consistency of the methods used to derive the CDR. Key elements of our new suite of algorithms are i) its application to different sensors (various SSM/I, AMSR-E and AMSR2), ii) a self-optimizing algorithm which dynamically tunes tie points to minimize SIC errors at 0% and 100%, and iii) a dynamic open-water filtering (OWF) to mitigate spurious SIC values caused by residual weather influences while keeping actual low SIC. For the three SIC CDRs published here we investigate time-series plots of the optimized skills of the SIC algorithms, and the temporal stability of the OWF (section 4.2.1).

The second aspect is to evaluate the SIC CDRs with independent SIC values. In the present paper we focus on an evaluation at 0% and 100% SIC; results of the evaluation at inter-mediate SIC with various independent SIC will be published elsewhere. The methodology used and the results are given in section 4.2.2.

The third aspect is to evaluate the uncertainty estimates provided with the SIC CDRs. The uncertainties should provide the range within which the SIC CDRs values are allowed to vary around the true value, and this is evaluated at 0% and 100% SIC in section 4.2.3.

## 4.2.1 Monitoring stability and internal consistency

Many time-series plots can be produced to illustrate the stability and internal consistency of the three CDRs. As an example, Figure 8 shows the time-series of the algorithm training statistics at the Open Water target. As described in sections 3.2 and



3.3, the algorithms implemented in the three CDRs dynamically tune their parameters to yield zero bias and minimum standard deviation of the computed SICs (*aka* best accuracy) over the Open Water (OW) and Closed Ice (CI) training targets. Figure 8 shows the Northern Hemisphere (NH, top) and Southern Hemisphere (SH, bottom) temporal evolution of the standard deviation (solid lines) and bias (dotted lines) of the SIC algorithms over OW target areas. Prior to further describing

Figure 8, it is important to note that the biases and standard deviations discussed here are internal to the processing chains, not an evaluation of the CDRs against independent observations of SICs. An initial evaluation of the CDRs against independent ground-truth observations is the topic of section 4.2.2.

From Figure 8, it is easy to see that the algorithms implemented in the three CDRs achieve zero bias (dotted lines along the y=0 axis) for all instruments and both hemispheres, on a daily basis. To achieve zero bias albeit the changes in central

wavelengths and calibrations from one satellite to the next is one of the key advantages of using dynamically-tuned algorithms (section 3.3).

The impact of the explicit correction of brightness temperature from atmospheric noise effects is also clearly visible on Figure 8, since the standard deviations resulting from un-corrected $T_B$ data (thin solid lines) are consistently above those from corrected data (thick solid lines) by about 3% to 4% on average, depending on the season and hemisphere. The seasonal

variability is also larger from the un-corrected data, especially in the NH. It is noteworthy that the atmospheric noise reduction step does not improve much the OW standard deviation in the SH at the beginning of the OSI-450 period, for the SMMR instrument (1979-1987). As noted at the end of section 3.4.1, OSI-450 uses the Wentz (1983) RTM for SMMR, and the Wentz (1997) RTM for SSM/I and SSMIS. The parametrization implemented in the SMMR RTM are probably less developed than in the SSM/I and SSMIS RTM, which might explain why the impact on our standard deviation is more

limited for SMMR. Another possible cause would be that the re-analysed fields from ERA-Interim are less accurate in the SMMR era than from with the SSM/I, especially in the SH were other sources of conventional observations are scarcer. Even if not as large as later in the time-series, atmospheric correction does yield a positive impact on the accuracy of OW SICs during the SMMR era.

The SICCI-25km and SICCI-50km standard deviations are also plotted on Figure 8 (only those after atmospheric correction

so as not to clutter the plot area). SICCI-25km (reds) achieves sensibly the same OW standard deviation as OSI-450. Since SICCI-25km uses very similar frequency channels to those of OSI-450 (Table 1), it is not surprising they achieve similar accuracy. The central frequency of the AMSR-E and AMSR2 channels (18.7 GHz) is slightly further away from the water vapour absorption line (~22 GHz) than the SSM/I and SSMIS channels (19.3 GHz). This difference in frequency yields better accuracy for SICCI-25km than OSI-450 when using un-corrected $T_B$ data (not shown) but this effect is mostly

cancelled after atmospheric correction (though not fully in SH, bottom panel).

SICCI-50km (greens) is more accurate than both SICCI-25km and OSI-450, by nearly 1% in NH, and 0.5% in SH. This is expected from the choice of frequency channels, since SICCI-50km uses a C-band (6.9 GHz) channel, while SICCI-25km and OSI-450 use Ku-band (~19 GHz). Three effects lead to better accuracies of SIC retrievals at low-frequencies: 1) the atmosphere is more transparent, yielding better accuracy over OW, 2) the noise sources such as sea-ice type, snow depth,



snow scattering, etc... have less impact at low frequencies, and 3) the permittivity (and hence $T_B$) of sea ice and water are more different, resulting in a larger dynamic range for sea-ice concentration retrievals. SICCI-50km is designed to be the most accurate of the three SIC CDRs. However, it achieves a coarser spatial resolution (50 km) due to the limited size of the AMSR-E and AMSR2 antenna. The time-series in Figure 8 illustrate that the algorithms are internally consistent, behave as

expected, and are effectively tuned to achieve zero bias and smallest possible retrieval noise for each instrument in the time series.

The role of Open Water Filters (OWF) is to detect and remove weather-induced false sea ice over open water, while preserving the true low concentration values (typically at the ice edge) at best. As introduced in section 3.4.2, the threshold

of the OWF is tuned dynamically against the daily updated training data samples (thus by instrument, and by hemisphere) to preserve true SIC values down to 10%. A water/ice separation limit at 10% SIC is an ambitious goal, but is necessary to ensure that time-series of Sea Ice Extent (SIE, usually defined with a threshold of 15% SIC) are not influenced by the OWF and only by the evolution of true SIC. Figure 9 shows time-series of NH (solid lines) and SH (dashed lines, almost coinciding with NH lines) of the 1%-percentile value of all `ice_conc` values (thus after the OWF is applied) that are

strictly positive and below 30% SIC for the OSI-450 (blue), SICCI-25km (red), and SICCI-50km (green) CDRs. These are thus time-series of the typical minimum detected SIC that are preserved by the OWFs. A solid horizontal line is drawn at 15% SIC value, the threshold commonly chosen for SIE computations. The OSI-450 curves are very stable with time and increase only slightly from around 9% SIC at the beginning of the period to around 10.5% SIC at the end. They are in any case well below the 15% threshold throughout the data record and very little few jumps are observed when transitioning

between sensors. The seasonal cycles are limited to few tens of a percent at the end of the period (few percent at the beginning). The SICCI-25km curves are close to the OSI-450 ones, but at a slightly larger value of 11%. The SICCI-25km curves are also well below 15%. The SICCI-50km curves are those showing the largest variation. The average value for SICCI-50km is at about 10%, but the seasonal variations are much larger, ranging from lowest 5% to highest 15%. The temporal stability of the time-series on Figure 9 document that the tuning of the OWFs at values close to 10% SIC is

successful for the two data records that rely on the 19 GHz and 37 GHz for computing their SICs (OSI-450 and SICCI-25km) and not as good for SICCI-50km that uses the 6 GHz and 37 GHz channels to compute the SIC values. Although SICCI-50km does not compute SICs from 19 GHz and 37 GHz channels, its OWF is still based on the GR3719v threshold (section 3.4.2). The mismatch in frequency and resolution between the channels used to compute the OWF and those used to compute SIC explains the larger variability of the SICCI-50km time-series in Figure 9.

We note in addition that both the OSI-450 and SICCI-25km CDRs dynamically tune their optimal data plane for low concentration range $\theta_{OW}$ in the (19V, 37V, 37H) 3D $T_B$ space, while the OWF is only tuned in the (19V, 37V) $T_B$ plane. The departure of the optimal data plane from the (19V, 37V) plane (by convention at $\theta=0°$, see right-hand side panel in Figure 4) causes the slight increase of the 1%-percentile curves of OSI-450 during the time period, and the different value obtained with SICCI-25km. Ideally, the OWF should be tuned in the same 3D $T_B$ space as used for the SIC algorithms. Such 3D-




based filters do not exist at present and additional research is needed in that field. All in all, we note that all three CDRs achieve a rather stable detection of true SIC below the 15% SIC threshold commonly used to define SIE. To the best of our knowledge the temporal consistency of the minimum detected SIC has not been documented for other available CDRs, although all use OWFs.

### 4.2.2    Evaluation against ground truth

For the evaluation of the SIC CDRs, we used a temporal extension of the Round Robin Data Package (RRDP) used by Ivanova et al. (2015) to study the strengths and weaknesses of more than 30 published SIC algorithms. Among other datasets, the RRDP v2 holds ground-truth locations for Open Water cases (OW, 0% SIC) for the period 2002-2015, as well as ground-truth locations for Closed Ice (CI, 100% SIC) for the period 2007-2016. The OW locations are situated just

outside the climatological mask delineating maximum sea-ice extent but, well inside the buffer zone added to it in section 3.6. They are distributed as evenly as possible in longitude. The CI locations are selected in areas of high sea-ice concentration and after 24h of convergent sea ice motion, as computed from a highly accurate SAR-based sea-ice drift product from the Copernicus Marine Environment Monitoring Service (CMEMS, http://marine.copernicus.eu). The OW and CI datasets of RRDP are described in more details in Ivanova et al. (2015).

For the evaluation of the SIC CDRs over open water, we extracted OSI-450, SICCI-25km and SICCI-50km CDR SIC (variable `raw_ice_conc_values`) and total uncertainty $\sigma_{tot}$ data at the grid cell closest to the OW locations in the RRDP v2 from two months in summer: August and September in the Arctic and January and February in the Antarctic, and from three months in winter: January through March in the Arctic and July through September in the Antarctic. For the evaluation at 100% SIC conditions, we collocated the SIC CDRs with the SAR-based CI locations in the RRDP v2 for months

November through March (Arctic) and May through September (Antarctic) in the same way as we did for open water; no spatial or temporal interpolations are performed. We note that CI ground-truth data from East Antarctic are missing completely, however, because of a lack of enough SAR image acquisitions. Using the `status_flag` variable, any SIC being contaminated by land spill-over effects or by too high air temperatures were discarded.

For open water, we find quite similar SIC distributions around 0% for all three CDRs for both hemispheres (Figure 10).

During winter (blue curves) OSI-450 and SICCI-25km are skewed a bit towards negative SIC in the Arctic but not in the Antarctic. During summer (red curves) we find SIC distribution to be skewed to negative SIC for all CDRs except OSI-450 in the Antarctic. Distributions are generally more narrow for SICCI-50km than for the other two CDRs. Figure 10 (a) and b), black crosses) illustrates the very similar accuracies for OSI-450 and SICCI-25km with a mean SIC of 0% or -0.2% during summer and of ~0.5% during winter in both hemispheres. For SICCI-50km, the accuracy varies more: summer: ~ -0.5% and

winter: 0.2% to 0.5%, than for the other two CDRs. The standard deviation of the mean SIC (black bars), e.g. the precision, ranges between 1% and 2%. Without exception the precision is better (smaller) in summer than winter. For both hemispheres, we find that the precision of OSI-450 and SICCI-25km SIC CDRs is similar to each other and less good than that for SICCI-50km, which is in line with the findings in Figure 8.



For sea ice, we find almost identical SIC distributions around 100% for OSI-450 and SICCI-25km for both hemispheres (Figure 11, a), b), and d), e)). Distributions for SICCI-50km are considerably narrower (Figure 11, c), f)) and, in comparison to OSI-450 and SICCI-25km, have a modal value closer to 100%. All three CDRs exhibit a negative bias, i.e. a modal SIC < 100%. Figure 11 c), d) further illustrates that SICCI-50km provides the smallest bias (best accuracy) in both hemispheres

with a mean SIC of 99.5% and 99.3% for Arctic and Antarctic, respectively. In addition, SICCI-50km also offers the smallest SIC standard deviation of the mean (black bars), i.e. the best precision, of ~2% and ~3% for Arctic and Antarctic, respectively. OSI-450 and SICCI-25km provide a less good accuracy with a mean SIC of ~98% in the Arctic and ~98.5% in the Antarctic, which comes also with a higher SIC standard deviation of the mean: 3.5% to 4.0%. Accuracy and precision are quite similar for OSI-450 and SICCI-25km.

### 4.2.3    Evaluation of the uncertainties

We computed the mean SIC total uncertainty $\sigma_{tot}$ for OSI-450, SICCI-25km and SICCI-50km for exactly the same set of grid cells as used in section 4.2.2 (Figure 12, blue bars).

For open water, SIC = 0% (Figure 12, a), b)), we find that mean SIC total uncertainties differ by less than 0.3% between OSI-450 and SICCI-25km and take values of ~2% during summer and of ~2.5% during winter. For SICCI-50km, the mean

SIC total uncertainty is smaller than for the other two CDRs – particularly during summer in the Northern Hemisphere: ~1.5% compared to ~2% in winter. Without exception mean SIC total uncertainties exceed one standard deviation of the retrieval errors (compare black and blue bars in Figure 12 a), b)). Also without exception, mean SIC total uncertainties are smaller than two standard deviations of the retrieval errors (not shown).

For sea ice, SIC = 100% (Figure 12 c), d)), we find that mean SIC total uncertainties for OSI-450: ~3% are smaller than

those for SICCI-25km: ~3.5%, in both hemispheres. For SICCI-50km mean SIC total uncertainties are smaller than for the other two CDRs – particularly in the Northern Hemisphere: ~2% (Figure 12, c)). For OSI-450 and SICCI-25km, mean SIC total uncertainties are smaller than one standard deviation of the retrieval errors. For SICCI-50km, mean SIC total uncertainties are comparable to (Figure 12, c)) or larger than (Figure 12, d)) one standard deviation of the retrieval errors.

Thus, the results summarized in Figure 12 indicate that the uncertainty $\sigma_{tot}$ provided with the three CDRs are on average at

an appropriate level for the high sea-ice concentration range (SIC = 100%) but are (slightly) overestimated for the low sea-ice concentration range (SIC = 0%). The SIC total uncertainty $\sigma_{tot}$ has contributions from the algorithm uncertainty $\sigma_{algo}$ and the smearing uncertainty $\sigma_{smear}$. Because the locations for ground truth estimates generally are not at the ice edge, the smearing uncertainty term is close to zero and $\sigma_{algo}$ dominates the evaluation results summarized in Figure 12. As introduced in section 3.5, the algorithm uncertainty is computed as the standard deviation of the retrieval error at the dynamically

selected training data samples. For SIC = 100% cases, the dynamically selected training samples are spread mostly all over the high sea-ice concentration regions, and there are thus good odds that the training samples are representative of the geophysical conditions in the ground-truth dataset, and that in turn the reported uncertainties are in agreement with the retrieval errors for SIC=100% cases. For SIC = 0% however, the training data samples are selected at the outskirts of an



expanded maximum ice climatology, while the ground-truth locations are just outside the same climatology. The training samples thus generally correspond to lower latitude conditions (ocean surface, and atmosphere conditions) than the ground-truth locations. For example, training samples can be picked in regions of more frequent synoptic low-pressure paths than the conditions really prevailing at the ice edge, and to the least at the location of the ground-truth estimates used in this section.

More developed sea state as well as wetter atmosphere contribute to the overestimation of $\sigma_{algo}$ (hence $\sigma_{tot}$) by at maximum 1% SIC (one standard deviation) in SIC = 0% conditions. We finally note that the results from Figure 12 cover the end of the time period (the AMSR-E and AMSR2 years) while the maximum ice extent climatology driving the selection of training samples is computed for the whole almost 40 years of sea-ice data record. Trends in sea-ice decline (in the NH, especially summer) might thus have an amplification effect on the overestimation of the uncertainties, as the location for selecting
training samples is increasingly further away from sea-ice edge as decades pass.

### 4.3    Caveats and known limitations

Known limitations of the SIC CDR are listed in this section. All the aspects listed below apply in large extent to the other existing SIC data records based on Passive Microwave sensor data. Not all of these limitations are reflected in the uncertainty fields of the CDR, as presented below.

The Open Water Filter (aka Weather Filter) implemented in the new SIC CDR is based on combination of the PMR channels around 19 GHz and 37 GHz (section 3.4.2). Although the filter is efficient at detecting and removing weather-induced noise (false ice) over open water, it is also designed to remove some amount of true low-concentration ice, especially in the marginal ice zone. Although dynamic tuning strategies were developed for these new CDRs, users are explicitly warned to take close attention to filtered conditions, especially close to the ice edge. The un-filtered ("raw") SIC values can always be
accessed in field `raw_ice_conc_values` (section 4.1, Figure 6). The effect of the open water filter is not included in the uncertainty variables, which are pertaining to the un-filtered (raw) ice concentration values. See also the discussion on the temporal consistency of the OWF for the three CDRs in section 4.2.1.

All SIC algorithms based on the passive microwave data are very sensitive to melt-pond water on top of the ice (Kern et al. 2016). The radiation emitted at these wavelengths comes from a very thin layer at the surface of the ice, which does not
enable distinguishing between ocean water (in leads and openings) and melt water (in ponds). The `ice_conc` variable of the SIC CDRs, thus should hold an estimate of 1 minus the open water fraction in each grid cell, irrespective if this water is from lead and openings or ponds. The mis-interpretation of melt water on top of sea ice as open water is not included in the uncertainty variables (Yang et al., 2016). The uncertainties embedded in the files are those for "1 minus the open water fraction".

Due to many factors (including smooth surface, absence of snow, brine content) concentration of thin sea-ice (< 30cm) is underestimated by most of the PMR SIC algorithms (Cavalieri 1994). A complete, 100% cover of thin sea ice will be retrieved with a lower concentration, depending on the thickness (Ivanova et al. 2015). The effect of thin sea ice is not included in the uncertainty fields of the SIC CDRs.





The SIC data records aims at addressing needs from a wide range of users, from interested general public to climate modellers and climate services. It was decided to provide interpolated sea-ice concentration values in places where original input satellite data was missing, aiming at most complete daily maps. Both temporal and spatial interpolation is used (section 3.6). The locations where interpolation is used are clearly identified in the status_flag layer. These interpolated sea-ice

concentration values should generally be used with caution for scientific applications, especially the values obtained from spatial interpolation. The uncertainty variables are not interpolated where data was missing. Days where no satellite data was available, e.g. every other day in the SMMR time period, are not interpolated and corresponding files are missing from the data records.

OSI-450 is presented at 25 km grid spacing. However, a spatial sampling of 25 km does not fully represent the true spatial

resolution of the product since the footprint of the SMMR, SSM/I, and SSMIS channels used by the algorithms is coarser (Table 2). The mismatch of grid spacing to the true resolution of the instrument footprint is taken into account in the uncertainty model of the OSI SAF CDR and is a key contribution to the *smearing* uncertainty (section 3.5). The footprint of the channels used in the ESA CCI CDRs (SICCI-25km and SICCI-50km) are much more compatible with the 25x25km (SICCI-25km) and 50x50km (SICCI-50km) target grids, so that their grid spacing is closer to the true resolution.

The radiometric signature of land is similar to that of sea ice at the wavelengths used for estimating the SIC. Because of the large foot-prints and the relatively high brightness temperatures of land and ice compared to water, the land signature is "spilling" into the coastal zone open water and it will falsely look as intermediate concentration ice. This land-spill-over effect is corrected for as described in section 3.6. However, coastal correction procedures are not perfect, and some false sea-ice remains along some coastlines, especially for OSI-450 and SICCI-50km because of the larger foot-print of the

instrument. By the same token, some true coastal sea ice might be removed by the coastal correction scheme. Users are advised to check the values in the raw_ice_conc_values where the SIC estimates before the final coastal correction step are available. The uncertainty variables have larger values in the coastal regions where land spill-over effects are detected.

## 5 Discussion, Outlook and Conclusions

### 5.1 Discussion

This paper documents three new Sea Ice Concentration (SIC) Climate Data Records (CDR). One from EUMETSAT OSI SAF (OSI-450), and two from ESA CCI (SICCI-25km and SICCI-50km). All three share the same algorithm baseline, which is both a continuation of the EUMETSAT OSI SAF SIC approach (Andersen et al. 2006, Tonboe et al. 2016) and a series of innovations contributed mostly by the ESA CCI activities. The three CDRs are a family of data records that aim at addressing the GCOS Requirements for the Sea Ice ECV (GCOS-IP, 2016). The improvement with respect to earlier

versions of the CDRs include 1) using high-quality Fundamental Climate Data Records (FCDR) as input data (section 2.1), 2) a new family of self-tuning, self-optimizing SIC algorithms that dynamically adjust to the input $T_B$ data (sections 3.2, and 3.3), 3) novel noise reduction and filtering approaches (section 3.4), and per-pixel uncertainty estimates (section 3.5). The





product data files are designed so that interested users can revert some of the filtering steps and access the "raw" output of the SIC algorithms (section 4.1).

The three CDRs are designed to ensure temporal continuity throughout the almost 40 years of passive microwave data records. The OSI-450 dataset currently covers 1979 throughout 2015 with a consistent set of frequencies at 19 GHz and 37 GHz. Conversely to other CDRs (e.g. Meier et al. 2017 and its two components Bootstrap and NasaTeam), the channels around 22 GHz are not used for filtering water vapour contamination. The 23.0 GHz channels of the SMMR instrument were highly unstable since launch, and eventually ceased to function on March 1985. This is one of the reasons why the Meier et al. (2017) only starts with SSM/I F08 09 July 1987 as a fully-qualified CDR (according to https://nsidc.org/data/g02202). A key asset of the algorithms we adopted is that they are self-tuning and self-optimizing to the data, which greatly helps achieving temporal consistency between different satellite missions, both in the past and future (discussed later as outlook).

The self-tuning and self-optimizing algorithms allowed e.g. to consistently process SIC CDRs from the AMSR-E and AMSR2 instruments. The SICCI-25km is an attempt at closing the gap in spatial resolution between what can be achieved from coarse resolution sensors like SMMR, SSM/I and SSMIS and the requirements of GCOS for 10-15 km spatial resolution (GCOS-IP, 2016). The almost 15 years record of brightness temperature observations from these two instruments is a key complement to OSI-450.

The decision to produce distinct CDRs, one with SMMR, SSM/I, and SSMIS and the other two with AMSR-E and AMSR2 is mainly based on the difference in spatial resolution. To mix the two types of sensors (coarse resolution with medium resolution) into a single CDR will require careful consideration of the mismatch of spatial resolution, and possibly advanced enhanced resolution methods (e.g. Long and Daum, 1998; Long and Brodzik, 2016) which are not used here. It is in any case doubtful if the resulting single CDR had met the temporal consistency requirements of many climate applications.

An initial validation of the three CDRs and their uncertainties is reported upon in section 4.2. Time-series plots document that the dynamic tuning of the SIC algorithms and of the OWF perform as expected, and that temporal consistency is mostly achieved despite the changes of frequencies and calibration between sensors. Based on similar frequency channels at 19 GHz and 37 GHz, the OSI-450 and SICCI-25km CDRs achieve similar accuracies, both in the time-series plots of internal tuning parameters (section 4.2.1) and when validated against ground-truth (section 4.2.2). Over open water, the retrieval accuracy of these two CDRs is as good as 1.5% to 2% SIC (one standard deviation) and without biases. Over sea ice, the retrieval accuracy is somewhat poorer (3.5% to 4% SIC) and with a limited low bias (2% SIC in NH, 1% in SH). The SICCI-50km uses a 6 GHz frequency channel instead of 19 GHz. Theoretically 6 GHz is a better channel for estimating sea-ice concentration since the atmosphere is more transparent, the influence of error sources like sea-ice age or snow processes have less influence, and the contrast between ocean and ice is larger. This is confirmed in our validation results. Over open water, the retrieval accuracy of SICCI-50km is as good as 1% to 1.5% SIC (one standard deviation). Over sea ice, the accuracy is better than 2.5% SIC and the bias limited to below 1%. The SICCI-50km is thus the most accurate of our three new CDRs but is also that with coarsest spatial resolution due to the large footprint of the 6 GHz channels.



Our evaluation results reveal very similar accuracies in the Northern and Southern hemispheres, even though the sea-ice conditions can be very different. Regarding algorithm performance the Arctic is more challenging at first glance. At least two radiometrically different ice types, multiyear ice and first-year ice, and a pronounced seasonal cycle of sea-ice and snow properties during summer with regular wide-spread occurrence of melt ponds on the ice surface need to be accommodated by

the algorithm. Antarctic multiyear ice has a less well studied and different radiometric signature than Arctic multiyear ice, resulting from other summer melt processes, e.g. melt ponds occur rarely; one could say it differs less from that of first-year ice on the one hand. On the other hand, direct and indirect weather influences, causing an unwanted variation in the retrieved sea-ice concentration, have been quite regional in extent in the Arctic Ocean (largely encompassed by land masses) while these have been a common, wide-spread phenomenon on Antarctic sea ice (bordered by oceans and at lower latitudes).

Therefore, a very similar algorithm performance in both hemispheres is not a surprise and agrees with earlier findings (e.g. Ivanova et al. 2015). We note that because they automatically tune their coefficients (tie-points, plane angle θ, etc…) to the training data specific for each hemisphere, our new algorithms can best adapt to radiometric properties of sea ice being different in both hemispheres.

An analysis of the temporal consistency of the Open Water Filter (section 4.2.1) also revealed that our dynamic tuning of the

OWF does not perform as optimally on the SICCI-50km CDR than on SICCI-25km and OSI-450. This is explained by the larger mismatch in frequency and resolution between the channels entering the SIC algorithms, and those used in computing the OWF (19 GHz and 37 GHz only). For all practical purposes we note that the dynamic tuning of the OWF as implemented here secures a rather stable level for the minimum detectable true SIC, in the order of 10% SIC, well below the 15% SIC threshold commonly used for defining Sea Ice Extent.

An evaluation of the uncertainties, a key element of the CDRs, is reported upon in section 4.2.3. We compare the uncertainty values reported in the product files with the retrieval error of the SIC field in conditions of known 0% and 100% SIC. Over 100% SIC, there is a close correspondence between the reported uncertainty and the observed retrieval noise, for both hemispheres. In open water conditions, the uncertainties provided in the CDR product files overestimate (by maximum 1% SIC in terms of standard deviation) the observed retrieval noise by a couple of percent. This slight overestimation is probably

due to the use of a buffer zone outside of the monthly maximum ice climatology extent to dynamically select the data samples used to train the algorithms (section 3.3) and derive uncertainties (section 3.5).

## 5.2   Outlook

The Climate Data Records presented in this manuscript will be further developed and extended in the context of the EUMETSAT OSI SAF. A full-reprocessing of the OSI-450, SICCI-25km, and SICCI-50km CDRs is namely committed to

by OSI SAF (version 3 of the CDRs) and should happen in 2021. It will use updated versions of the FCDRs –if available- and the new ERA5 atmosphere re-analysis from the EU C3S. At time of writing, no radical change of algorithms and processing steps is foreseen, but our paper identifies several improvements and evolutions that would be beneficial for these upcoming versions, and that these are briefly described below.



Although the ESA Climate Change Initiative Sea Ice projects went far in the characterization of the impact of melting, and melt-ponds have on sea-ice concentration retrievals from passive microwave data (Kern et al. 2016), the question on how to limit and best convey the increased uncertainty to users will benefit from more efforts. Results of an inter-comparison between the same data set of melt-pond fraction, sea-ice concentration and net sea-ice surface fraction that was used in Kern
et al. (2016) and the three CDRs presented in this paper as well as other available sea-ice concentration products, including those based on NASA-Team and Bootstrap algorithms, will be reported in a forthcoming article.

The uncertainty model presented here is already a significant improvement over that used in the previous version of the SIC CDR (Tonboe et al. 2016). Nonetheless, additional research is needed to better quantify the uncertainties and validate that they are fit-for-purpose. Since the way we derive uncertainties is directly linked to the way we select training data samples, it
could be investigated if to selecting training samples closer to the ice edge could improve the uncertainty values, and for example reduce the slight overestimation of uncertainties at SIC = 0% conditions documented in section 4.2.3. Another challenging topic is the quantification of cross-correlation scales (both in the temporal and spatial dimension) necessary to fully aggregate such CDRs at the scales relevant for evaluation of models or higher-level climate indicators (Bellprat et al. 2017).

Despite being from all seasons and in both hemispheres, the validation results presented in this paper cover 0% and 100% SIC conditions, but not the intermediate range found in the marginal ice zone due to the lack of high quality validation data. Results of the evaluation of the three CDRs with independent data, i.e. ship-based visual observations the sea-ice cover, sea-ice area fraction derived from high-resolution optical satellite imagery, have been reported in the Product Validation and Intercomparison Report PVIR (available from http://cci.esa.int) and will also be published in forthcoming articles. These will
also include an inter-comparison of time-series of the sea-ice area (SIA) and SIE as derived from the three CDRs and from other sea-ice concentration products.

Already from the early assessment of the new CDRs presented here, we can outline a number of algorithm developments that have the potential to further improve the accuracy of future SIC estimates based on passive microwave data, both in climate and operational applications. The new self-tuning, self-optimizing algorithms introduced in this paper are currently limited to
3D $T_B$ spaces. This is because the optimization of the projection plane is handled via a rotation angle along a 3D axis, a geometrical concept that is difficult to upscale to more dimensions. The generalization of this optimization to $n$D (where $n$ could be any subset of the channels available on a given passive microwave imager) would open for exploring all possible $T_B$ channels combinations in a systematic manner, and maybe unveil algorithms achieving even better accuracy that the 3D ones used here. By the same token, it should be investigated if the concept of a consolidated ice curve (as opposed to an ice
line) could not be better embedded by SIC algorithms in the future, instead of being a correction step applied a-posteriori as is the case in our CDRs. A third algorithm development to be investigated is the generalization of the concept of Open Water Filter (*aka* Weather Filters) to 3D or even $n$D, so that the OWFs are always tuned and computed using the same $T_B$ channels as the SIC. This development has the potential to improve the temporal consistency of the OWF at low SIC values, across changes of wavelengths and calibration, or when using other $T_B$ channels than 19 GHz and 37 GHz. In any case and even



after almost 40 years of routinely available passive microwave observations of the polar regions, the underlying algorithms can be improved to yield improved accuracy and there is scope for continued research and development in the field.

Another development for using such SIC CDRs to evaluate models and perform Data Assimilation would be the definition and uptake of observation operators (aka satellite simulators, e.g. Kaminski and Mathieu, 2017). Once the remaining

systematic errors (such as underestimation of very thin ice, impact of melt-pond water...) have been described and quantified, the next step is for the EO science community to define observation operators. These operators are typically parametric formulations that express the quantity retrieved from Earth Observation techniques (in our case the sea-ice concentration values in the CDR) as a combination of physical variables in the model world (e.g. sea-ice area fraction, thickness of sea ice categories, area coverage of melt-pond…). We advocate these operators are built in a step-wise, pragmatic manner

(Lavergne, 2017). This development should happen in complement to building more "end-to-end" satellite simulators that aim at linking the physical variables in the model world directly to satellite radiances.

Thanks to using the C-band channels (4-8 GHz) the SICCI-50km CDR exhibits outstanding sea-ice concentration retrieval accuracy, both at low and high concentration range. The usability of this CDR can however be challenged by its rather coarse resolution (the 6 GHz channels of AMSR-E have a iFoV of 75x43 km (Table 2) and the CDR is presented on a 50 km

grid), which is a direct consequence of the limited antenna diameter of the AMSR-E (2.0 m) and AMSR2 (2.1 m) instruments. Our results fully support that a passive microwave mission measuring at the C-band frequency, and carrying a large-enough antenna to enable ground resolutions better than 15 km (at C-band) would be a clear asset for all-weather, global, daily-covering sea-ice concentration mapping for operational applications. At time of writing, such a satellite mission is under study as a High Priority Candidate Mission for the European Union's Copernicus Space Component Expansion: the

Copernicus Imaging Microwave Radiometer (CIMR, https://cimr.eu).

A key requirement of GCOS for addressing the needs of the climate modelling community as well as the Climate Information Services such as the EU Copernicus Marine Environment Monitoring Service (CMEMS, http://matine.copernicus.eu) and Copernicus Climate Change Service (C3S, http://climate.copernicus.eu) is the seamless extension of the CDRs in the context of operational services. These operational services aim at best temporal consistency

with the CDRs, but still may have to rely on different data streams. They are referred to as Interim Climate Data Records (ICDR) because they are meant as a temporary extension until a full-reprocessing of the CDRs is performed (Yang et al., 2016). For the SIC variable, both the EUMETSAT OSI SAF CDR of Tonboe et al. (2016) and the NOAA/NSIDC CDR (since late 2017, Version 3) are extended daily by such ICDR. We are naturally working towards starting an operational ICDR for our new CDRs, tentatively by late 2018, with a 16-days latency.

Aside from the technical aspects of reliably running the CDR processing chains on a daily basis, a major challenge that all SIC CDR data producers now face is the end of life for the U.S. Defense Meteorological Satellite Program (DMSP), that has been the work-horse for virtually all Sea Ice CDRs since SSM/I F08 in 1987 (Table 2 and Figure 1). At time of writing, the Japanese AMSR2 instrument is already passed its design lifetime (5 years, launched mid 2012), with no committed successor. For the continuation of the new OSI SAF SIC CDR, we are investigating the quality of the Micro-Wave Radiation



Imager (MWRI) on board China's Feng-Yun 3 (FY3) satellites. Preliminary results are encouraging and, when consolidated, will be presented in a follow-up paper. The first satellite of the European Polar System Second Generation (EPS-SG), that will carry a Microwave Image (MWI) will be launched in 2023. It can be used to further extend the SIC CDR up-until the late 2040s. It is noticeable that EPS-SG MWI implements quite a different frequency for Ka-band (26.5–40 GHz): 31.4 GHz

instead of 36 - 37 GHz for SSM/IS and AMSRs (Table 2). Because our algorithms are self-adapting to the data and their calibration, the implementation with MWI should be possible. The impact of using 31.4 GHz instead of 36 - 37 GHz for sea-ice concentration mapping still needs to be addressed.

### 5.3    Conclusions

Long-term consistency, traceability and an evaluation and documentation of uncertainties are arguably the three major

properties of any climate-data record. In this contribution, we have described how these requirements are reflected by the algorithm underlying the three new sea-ice concentration climate-data records OSI-450, SICCI-25km and SICCI-50km.

Long-term consistency is achieved by developing an algorithm that dynamically adjusts to changing environmental conditions and changing satellite sensors. In particular, applying the same algorithm to microwave products based on different frequencies and satellites allows users to combine the advantages of the length of the record of the OSI-450 product

with the high true spatial resolution of the SICCI-25km product and/or the low-noise product SICCI-50km.

Traceability of the algorithm and the resulting climate-data records is achieved by a combination of two approaches. First, the final product contains substantial information on the impact of the various processing steps. For example, they include at every time step per-pixel information on the impact of possible filtering. Second, the algorithm and the products are embedded into an operational context. This guarantees on the hand a long-term maintenance of these product, but in

particular establishes clear rules on version-tagging, documentation and availability of the underlying code, which allows other researchers to easily build on our work and to develop it further.

Uncertainties of all products is systematically documented in the final products and has carefully been evaluated. All products contain at every time step per-pixel information on uncertainties arising from the algorithm itself (e.g., sensor noise or residual geophysical noise) and the smearing uncertainty from spatial remapping. This information is in particular helpful

for data-assimilation purposes. The evaluation of uncertainties carried out in this provides some initial information on the remaining random per-pixel uncertainty which can be used as an estimate of observational uncertainty for example during model evaluation or data assimilation. We find in particular that our product has a long-term stable zero bias arising from the dynamical re-tuning of the tie points.

We hope that by explicitly addressing the three requirements of a climate-data record, our three new sea-ice concentration

records and the underlying algorithm will be a helpful resource for the climate-research community.

### Competing Interests

The authors declare that they have no conflict of interest.



**Acknowledgements**

The authors are grateful to Prof Edward Hanna and Dr Doug Smith for sharing insights and software for the dynamic tuning of the Bristol SIC algorithm. Our gratitude goes to the open-source community at large, and especially the maintainers of the Python language and its modules (numpy, scipy, matplotlib, pytroll,…).

The SMMR, SSM/I, and SSMIS FCDR (R3) was accessed from the EUMETSAT CM SAF (www.cmsaf.eu). Karsten Fennig and Marc Schröder, both at DWD, helped making best use of this data. The AMSR-E FCDR was accessed from NSIDC, and the AMSR2 data from JAXA. ECMWF ERA-Interim was accessed from the MARS archive.

This study and the development of the three new SIC CDRs was funded by EUMETSAT (through the 2nd Continuous Developments and Operation Phase of OSI SAF) and ESA (through the Climate Change Initiative SeaIce_cci project).

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





**Figures**

**Figure 1: Time-coverage diagram for the new ESA CCI and EUMETSAT OSISAF SIC CDRs. The ESA CCI CDR is based on medium resolution AMSR-E and AMSR2 sensors, while the EUMETSAT OSISAF CDR uses the coarse resolution SMMR, SSM/I, and SSMIS instruments. Other current and future passive microwave instruments, as well as the OSI SAF ICDRs are discussed in our Outlook.**

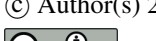



**Figure 2: From left to right, the three main elements (Level 2, Level 3, and Level 4) in the sea ice concentration processing workflow. The red boxes depict data files, the blue boxes correspond to individual steps (aka algorithms) in the processing. The files that exit a processing chain (e.g. the "L2 SIC and uncert and WF" at the bottom of the Level 2 processing chain) are the input for the next level of processing. NT is the Nasa Team algorithm.**







**Figure 3: Illustration of the Bootstrap Frequency Mode (BFM) and Open Water Filter (OWF) algorithms in a 36.5V (x-axis) and 18.7GHz (y-axis) T$_B$ space of AMSR-E (Winter NH conditions). The grey symbols are actual T$_B$ measurements over SIC=0% (triangles) and SIC=100% (disks) conditions. The SIC=100% measurements fall along a line (the consolidated ice line) while the mean water signature is point H. An example measurement P (black circle) falling on the SIC=68% isoline illustrates the functioning of BFM. The blue solid and dotted lines illustrate the tuning and functioning of the OWF (as described in section 3.4.2).**





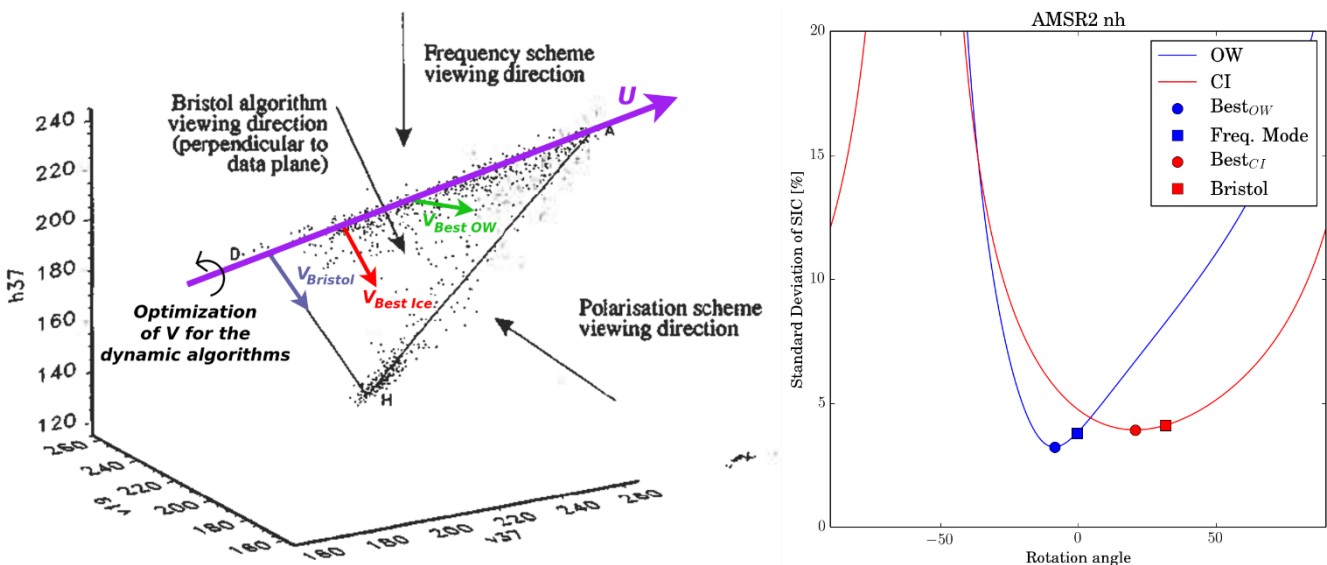

**Figure 4:** Left: three dimensional diagram of open water (H), and closed ice (ice line between D and A) brightness temperatures in a 19V, 37V, 37H space (black dots). The original figure is from Smith et al. (1996). The vectors u (violet), vBristol (blue), vBest-ice (red), and vBest-OW (green) are added, as well as an illustration of the optimization of the direction of V for the "dynamic" (self-optimizing) algorithms. Right: Evolution of the SIC algorithm accuracy for Open Water (blue) and Closed Ice (red) training samples as function of the rotation angle θ. Square symbols are used for the BFM (Freq. Mode) and BRI (Bristol) algorithms. Disk symbols locate the new, self-optimizing algorithms.

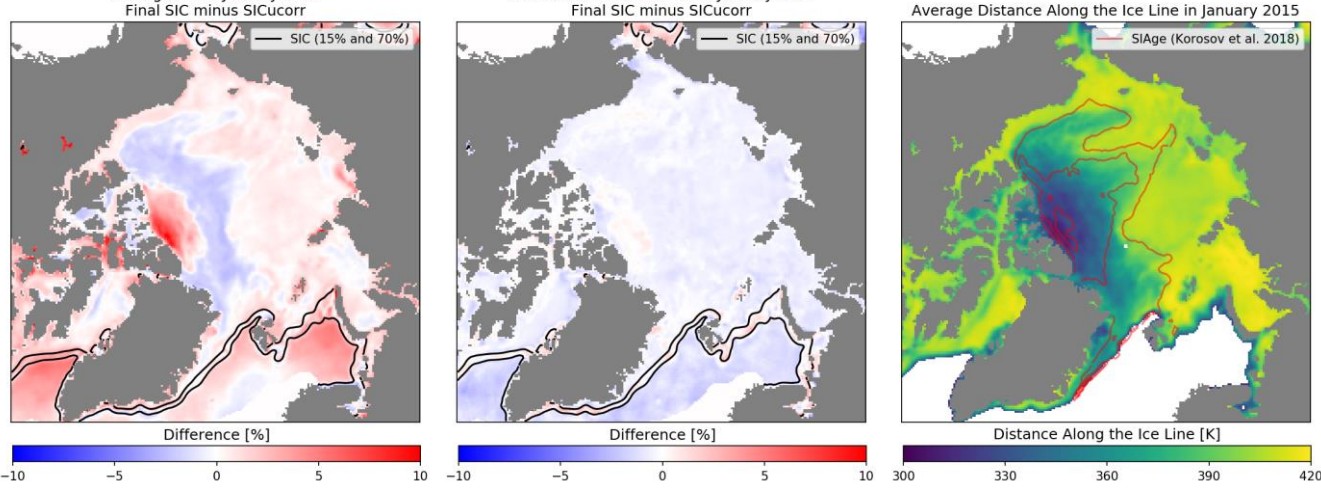

**Figure 5:** Left and center panels: difference maps between the January 2015 mean (left) and temporal variability (center) of the final SIC and the uncorrected SIC (SIC$_{ucorr}$) in the Arctic Ocean. Black solid lines are at the 15% and 70% SIC levels (marginal ice zone). Right panel: January 2015 mean Distance Along the Ice Line (DAL) values, red lines are transitions between 1$^{st}$ year sea ice, 2$^{nd}$ year sea ice, and older sea ice from Korosov et al. (2018).







**Figure 6: Example SIC fields on 25th September 2015 from the three CDRs (top left: OSI-450, top right: SICCI-25km, bottom left: SICCI-50km) over the Weddell Sea. Bottom right panel shows the content of variable raw_ice_conc_values from the OSI-450 file for the same date and area. Note the two discontinuous color scales for the bottom right panel.**





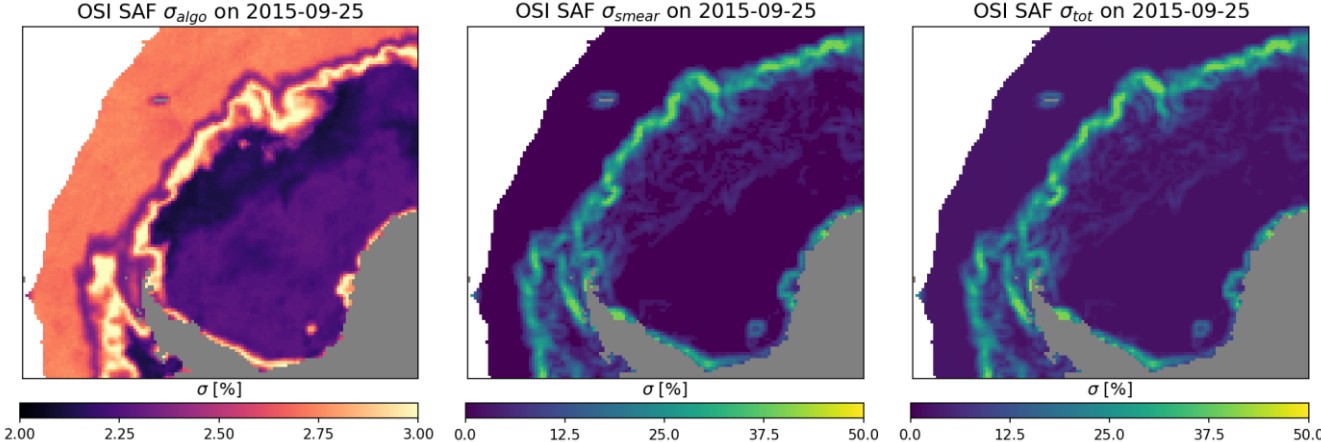

**Figure 7: Example fields of uncertainties on 25th September 2015 from the EUMETSAT OSI SAF CDR over the Weddell Sea. The component σ_algo (left), σ_smear (center), and the total uncertainty σ_tot (right) are shown. σ_tot is dominated by the σ_smear contribution.**





**Figure 8:** Time series of performance statistics for the three CDRs (blues: OSI-450, reds: SICCI-25km, greens: SICCI-50km) over the Open Water target for the Northern Hemisphere (top) and Southern Hemisphere (bottom). For OSI-450 and SICCI-25km, the color of the lines is for individual satellites, as used in Figure 1. For OSI-450, the thick (*resp* thin) solid lines plot the OW standard deviation of SIC (*resp* $SIC_{ucorr}$). The thin solid lines are only plotted for OSI-450 so as not to clutter the plot area. The bias of SIC is plotted with a dotted line.





**Figure 9: Time-series of the monthly mean 1%-percentile value of all strictly positive SICs that are below 30% (variable ice_conc) for the three CDRs (blues: OSI-450, reds: SICCI-25km, greens: SICCI-50km). Solid lines are for NH, dashed lines for SH. This time-series plot investigates if the dynamic tuning of the Open Water Filters results in temporal consistency of the minimum detected true SIC across all satellites.**





**Figure 10: SIC distribution around SIC = 0% at the selected open ocean locations for, from top to bottom, OSI-450, SICCI-25km and SICCI-50km in the Arctic (images a) to c)) and the Antarctic (images d) to f)). Blue (red) curves and numbers refer to results from winter (summer); the numbers in parenthesis behind the season denote the count of cases used. Numbers below the season denote the mean SIC plus/minus one standard deviation of the mean (in parenthesis) in percent SIC. Binsize is 0.5%. Distributions are normalized to give a total of 1.**



**Figure 11: SIC distribution around SIC = 100% from the RRDP-2 data set for, from top to bottom, OSI-450, SICCI-25km and SICCI-50km in the Arctic (images a) to c)) and the Antarctic (images d) to f)). Black, red and blue curves and numbers refer to all data and data limited to ERA-Interim 2m-air temperatures < -5°C and < -10°C, respectively. The numbers behind the limitation text (e.g. "all") denote the count of data used; the numbers below denote the mean SIC plus/minus one standard deviation (in parenthesis) in percent SIC. Binsize is 0.5%. Distributions are normalized to give a total of 1.**





**Figure 12:** Summary of histogram statistics from Figures 10 and 11 for SIC = 0% (images a) and b)) and SIC = 100% (images c) and d)) for the Arctic (left) and the Antarctic (right). Crosses and black bars denote the mean SIC plus/minus one standard deviation. Red horizontal bars denote the modal SIC. Blue bars denote the range covered by the mean SIC plus/minus one total standard error. Letters "S" and "W" in images a) and b) refer to summer and winter, respectively.




| | Instruments & [Channels] | Time Period | Grid Spacing | Originator | DOI |
|---|---|---|---|---|---|
| OSI-450 | SMMR, SSM/I, SSMIS [19V, 37V, 37H] | 1979-2015 | 25x25km | OSI SAF | 10.15770/EUM_SAF_OSI_0008 |
| SICCI-25km | AMSR-E, AMSR2 [19V, 37V, 37H] | 2002-2011, 2012-2017 | 25x25km | ESA CCI | 10.5285/f17f146a31b14dfd960cde0874236ee5 |
| SICCI-50km | AMSR-E, AMSR2 [6V, 37V, 37H] | 2002-2011, 2012-2017 | 50x50km | ESA CCI | 10.5285/5f75fcb0c58740d99b07953797bc041e |

**Table 1: Summary of the three SIC CDRs presented in this paper. The values entering the table are all described in the course of the paper.**



| Platform and Instrument | Start date | Stop date | Frequency, in GHz, (footprint resolution in km) of channels | Width of polar observation hole | View angle | Comment |
|---|---|---|---|---|---|---|
| Nimbus-7 SMMR | 01/01/1979 | 20/08/1987 | 18.0 (54x35), 37.0 (28x18) | 84° | 50.2° | Operates every other day. Two long periods with missing data are 29/03-23/06 1986, and 03/01-15/01 1987. |
| DMSP F08 SSM/I | 090/07/1987 | 18/12/1991 | 19.3 (70x45), 37.0 (38x30) | 87° | 53.1° | A long period with missing data is 03/12-31/12 1987. |
| DMSP F10 SSM/I | 07/01/1991 | 13/11/1997 | 19.3 (70x45), 37.0 (38x30) | 87° | 53.1° | Significant variation (slow oscillation) of the incidence angle during its life time. |
| DMSP F11 SSM/I | 01/01/1992 | 31/12/1999 | 19.3 (70x45), 37.0 (38x30) | 87° | 53.1° | |
| DMSP F13 SSM/I | 03/05/1995 | 31/12/2008 | 19.3 (70x45), 37.0 (38x30) | 87° | 53.1° | F13 operated longer but 31/12/2008 is the end of coverage in CM-SAF FCDR R3 |
| DMSP F14 SSM/I | 07/05/1997 | 23/08/2008 | 19.3 (70x45), 37.0 (38x30) | 87° | 53.1° | |
| DMSP F15 SSM/I | 28/02/2000 | 31/07/2006 | 19.3 (70x45), 37.0 (38x30) | 87° | 53.1° | F15 operated longer but 31/07/2006 is the end of coverage in CM-SAF FCDR R3 |
| DMSP F16 SSMIS | 01/11/2005 | 31/12/2015 | 19.3 (70x45), 37.0 (38x30) | 89° | 53.1° | |
| DMSP F17 SSMIS | 14/12/2006 | 31/12/2015 | 19.3 (70x45), 37.0 (38x30) | 89° | 53.1° | F17 operated longer but 31/12/2015 is the end of coverage in CM-SAF FCDR R3 |
| DMSP F18 SSMIS | 08/03/2010 | 31/12/2015 | 19.3 (70x45), 37.0 (38x30) | 89° | 53.1° | F18 operated longer but 31/12/2015 is the end of coverage in CM-SAF FCDR R3 |
| EOS Aqua AMSR-E | 01/06/2002 | 03/10/2010 | 6.9 (75x43), 18.7 (27x16), 36.5 (14x9) | 89.5° | 55° | |
| GCOM W1 AMSR2 | 23/07/2012 | 31/05/2017 | 6.9 (62x35), 18.7 (22x14), 36.5 (12x7) | 89.5° | 55° | AMSR2 operated longer but 31/05/2017 is the last date we fetched from JAXA for the CDRs. |

**Table 2: Platform, instrument, time period for input brightness temperatures used in the sea ice data records. All frequencies listed have both horizontal and vertical polarization channels**