# Peer review of "Version 2 of the EUMETSAT OSI SAF and ESA CCI Sea Ice Concentration Climate Data Records"

_The Cryosphere, 2018_

## Referee Comment (RC1) · Anonymous Referee #1 · 1 Sep 2018

Summary

This paper describes a new version of the OSISAF sea ice concentration product and the ESA sea ice CDR. The products are derived from passive microwave data. The new version includes several enhancements from the Version 1 OSISAF product. Comparisons with independent estimates show good agreement. The new version provides a consistent record of sea ice concentrations for the scientific community.

General Comment

The manuscript provides a thorough introduction of the new versions. The description of the algorithm and processing, including enhancements from Version is clear and

detailed. The initial evaluation results look reasonable and given that it builds on the previous version and thorough earlier validation, they are quite sufficient to provide high confidence in the quality of the product. The Level 4 filtered product is particularly beneficial for users who wish to have a "clean" concentration estimate and this is an excellent improvement from Version. I have only a few minor comments that the authors should address before publication.

(One further general comment: it would be helpful for readability to either indent new paragraphs and/or skip a line between paragraphs.)

Specific Comments (by page and line number):

P2, L9: while the albedo specifically depends on concentration, it is not only concentration: snow melt state and particularly melt ponds substantially affect albedo even for 100% concentration.

P4, L24-25: I find the (resp. XXXX) style awkward to read and somewhat confusing. I would just write out each in sequence rather than using parenthesis, but this may just be a preference by me.

P6, L1: "daily composited fields of SIC" – how is the compositing done? Is it simply drop-in-the-bucket?

P10, L12-20: What are the uncertainties in the NWP fields and the RTM? While the dynamical tiepoints and the double-difference approach may negate much of the influence, I do wonder how effective the correction is if the NWP data and/or the RTM have high uncertainties? This feeds into my next comment below.

P10, L28: the use of the NWP fields is novel and I like the physical approach. However, L is not reliable from NWP. Isn't L one of the largest if not generally the largest source of emission, at least over open water. So not being able to correct for that really limits the effectiveness of the NWP correction, doesn't it? The use of weather filters in the Level 4 fields eliminates this, which is good, but the quality of the Level 3 fields must

be limited, right?

P13, L17: It might be worth considering showing an example of the "ice curve". I can generally visualize, but a figure would perhaps better illustrate it.

P13, L20-26: I'm not sure I understand Figure 5. It appears to show an increase in open water concentration near the ice edge due to the correction (e.g., in Barents Sea and Davis Strait regions). Is that correct? Wouldn't that reduce the quality if the correction essentially added ice to open water regions?

P16, L3: This should be discussed further – why is the gridded land-spillover correction still needed after the swath correction? How much coastal contamination remains after the swath correction. If the swath correction is not sufficient on its own, is it worth doing – i.e., would the Cavalieri correction work just as well without the swath correction? I guess the basic question is whether there is a benefit to doing both corrections or is the Cavalieri correction just as good? If so, then why do the swath correction?

P16, L31: "basic isotropic schemes" is not very specific. Is it a bi-linear interpolation?

P19, L17-23: I can understand that the ERA-Interim fields are not as good earlier in the record and thus the correction for SMMR is not as good. However, there is a noticeable step-change between SMMR and SSMI in Figure 8. Did ERA-Interim undergo a step change in terms of data sources or other processing quality at the same time? If not, then it seems like it's not ERA-Interim (or at least not only), but rather something else causing the step change. Perhaps it's related to the change in frequency from 18 GHz for SMMR to 19.3 GHz for SSMI?

P21, L4: One thing not discussed is the potential impact of satellite crossing times on the retrievals. I assume the dynamic tiepoints should handle these discrepancies, but it might be worth mentioning.

P25, L12: Why not produce a 12.5 km or 10 km resolution AMSR-E and/or AMSR2 product, i.e., using the same channels (19, 37 GHz) as for SMMR-SSMI-SSMIS, but
obtaining a higher spatial resolution for the period of 2002-present? It seems like this would be more beneficial than at least the 25 km SICCI. I can see a benefit of using the 6V channel for the 50 km product, but that isn't in the 25 km SICCI.

Minor Comments (by page and line number):

P3, L17: use "in" instead of "entering"

P3, L26: use "share" or "provide" instead of "keep"

P16, L26: use "contrasts with" instead of "is conversely to"

P24, L3: "aiming at most complete" to "aiming to produce the most complete daily maps possible"

P25, L11: use "allowed, e.g., consistent processing of SIC CDRs. . .."

P26, L18: use "on the order of. . ."

P27, L1-2: use "the impact that melting and melt-ponds have. . ."

P27, L10: use "could be investigated if selecting. . ."

P28, L24: use "aim to have the best temporal consistency. . ."

---

## Referee Comment (RC2) · Anonymous Referee #2 · 17 Sep 2018

Review of Version 2 of the EUMETSAT OSI SAF and ESA CCI Sea Ice Concentration Climate Data Records, by Lavergne et al. (2018).

This paper gives a thorough, informative and detailed description of three important new climate data records of sea ice concentration. The science in the paper is comprehensive. I therefore only have suggestions for minor improvements (though there are quite a few) - mostly for clarifications to the text. The paper is clear and easy to read, despite a number of minor grammatical errors which are detailed below.

Minor comments

Page 2 line 1: Is this the observation uncertainty in assimilation for models? Unclear,

need to elaborate line 32: quantify what you mean by "coarse resolution"

Page 3 line 4: quantify what you mean by "medium resolution" line 19/20 & 22 (and throughout): Why only an "initial evaluation". Reading on shows that you have done more than just a cursory evaluation which is what this wording implies. Suggest reword.

Page 4 line 2: Suggest mentioning data gap in AMSR data earlier, perhaps when introducing Table 1. line 4: Suggest "documented in Table 2" should be "documented in the comments in Table 2". Would also be useful to have a full list of outages, perhaps a link to this in another document? Line 10 (and Table 2): "width of the polar observation hole" is not given, it's the bit that's viewed rather than the hole, also not a width as it's an angle, suggest rename this column line 23: Not sure that spatial resolution of SMMR is "somewhat similar" to SSM/I and SSMIS, suggest reword line 26: Clarify difference between sampling and resolution line 32: Consider showing eFoV in Table 2.

Page 5 line 4/5: Clarify if L1 data for SMMR, SSMI/S, SSMIS line 5: Add a line on what is an FCDR and what reprocessing has it undergone. Overlaps? Calibration? QC? line 6: add what period AMSR-E data covers line 9: more information needed on "resolution-matched"

Page 6 line 1: what type of grid? EASE? Line 2: what are the necessary steps? Can reference later on in paper if necessary lines 5-8: suggest moving these lines to page 5 line 32, after "flags". Would flow better. Line 18: clarify these numbers are sea ice fraction line 23: needs citation for BRI more accurate than BPM at high concentrations line 28: Figure 3 illustrates for AMSR-E data, example from Comiso (1986) is for SMMR. Need to clarify that these can be applied to other instruments.

Page 7 lines 4/5: show ice signatures on plot (mentions in text to left and right but not that clear) line 8: text says D-A, use A-D for consistency. Also A,D in figure 3 and D,A in figure 4, make consistent.

Page 8 line 4: What is the magnitude of the ice concentration change between algorithms for this example? Line 10: show theta = 90 on figure

Page 9 line 5: Have you also used a sliding window? Wording implies not, if it is suggest adding "similarly" before sliding. Why was the window changed from +-15 to +-7 days? Line 13: Why can this be assumed? Expand. Line 15: and SMMR, SSM/I, SSMIS? Also remove "than". Line 15/16: suggest moving sentence beginning "Recent investigations..." to line 13, before "It is assumed..."

Page 11 First paragraph: This is confusing as it sounds like different RTMs for each instrument but is it actually different optimisations? Reword. Line 10: quantify what is meant by "rather large" line 27: Is there a citation for the ATBD document itself? From line 20: As not using GR2219v suggest editing this section as don't need to describe in detail or give previous examples.

Page 12 line 1: Would be helpful to use a different symbol other than T to avoid confusion with temperature line 26: Implies that <10% will be removed anyway, even if GR3719v < T. If so need to clarify this in text. Note also in this section that GR3719v is also used for AMSR despite different channels. Also in this section, it is not really clear how the threshold values for the Gradient Ratios are selected, needs clarification.

Page 13 line 1: If you say it's visible, need to show on a figure line 12: Would be useful to show in a figure for visualisation line 13: "u" in italics is given as "U" on figure 4, needs to be consistent line 26: Why is there an increase in concentration due to the atmospheric correction (with reduced standard deviation) in figure 5? line 29: Are the contours specifically for 2015? Need to elaborate.

Page 14 line 2: confirm if this is the standard deviation of the differences, or the standard deviation over January for each pixel, then the difference of these (latter is as worded) line 8: Would be useful to see impact of ice curve correction and atmospheric correction separately on figures line 26: clarify footprint mismatch is between different channels

Page 15 lines 1&3: need to explain "3 dB footprint" or remove lines 2/3: also mention AMSR products line 10 and paragraph: Needs more information on how K was calculated line 21: land spill-over effects are critical for users in that missing data around coasts causes problems and has to be dealt with. Where you have removed data, have you done any filling?

Page 16 line 3: Does this improve things compared to Cavalieri et al. (1999) alone? line 6: Year for Donlon paper should be 2012. Also, not to change in the paper but note that I believe the mask has been updated for the SST CCI v2 processing. Line 15: "New Scotland" should be "Nova Scotia", no need to translate as still same in English lines 15&16: State whether you have done anything different in processing to get ice over inland regions and fresh water, either here or elsewhere in paper line 29&30: Clarify that you are not filling in missing days, e.g. in the SMMR period etc. Are you filling around coasts?

Page 17 Evaluation of the data: Have you simply looked through the data? Issues where processing has gone wrong, or the data looks strange have previously been an issue for OSI SAF CDRs. It would be very helpful for users not to have to do this QC. Line 7: add what the ERA-Interim data is used for in the processing line 28: colour scale is blue-red, not blue-yellow-red line 29: Is noise just characterised as below 10%? line 33: suggest move "as nominally returned by the SIC algorithm" to line 27 after "raw_ice_conc_values"

Page 18 line 16: what about summer?

Page 19 line 23: SMMR uncorrected is also better than for SSM/I and SSMIS, particularly in the NH. Why? Line 30: in winter? Line 31: need to give seasonal figures

Page 20 line 4: "internally consistent" - do you mean consistent over time? Line 5: Can't tell from figure 8 that it's the smallest possible. Suggest reword "and smallest possible retrieval noise" to "and a small retrieval noise" line 14: change "thus after the OWF is applied" to "thus after all the filters including the OWF are applied" for

clarification line 17&18: as the range changes are they stable with time? Also need to give separate summer and winter values and incorporate line 20 in the discussion. Also separate summer and winter values line 21. line 27: might be worth adding that this is addressed as future work later in the paper line 33: Need to elaborate on how this could cause an increase over time

Page 21 lines 17&18: Why 2 months in summer and 3 months in winter? line 23: Give the T2m threshold (if not mentioned elsewhere?) line 25: "skewed a bit" – could quantify the skewness, or reword to "slight negative skew" or similar, and elsewhere. Line 27,28,29: should refer to Figure 12, not Figure 10. Values given are not the same as on Figure 10, unclear. Line 33: In winter it looks fairly similar though.

Page 22 line 3: reference "(Figure 12)" after "100%" line 4: should be Figure 12, not Figure 11 line 7: Suggest replace "less good" with something like "poorer, but still acceptable". Suggest cut the last sentence of this paragraph as is a repetition. Line 12: The total uncertainty is described as "standard error" on Figure 12, need to reword this. Paragraph around line 20: Elaborate on why uncertainties for SICCI-50km are smaller than for the other two datasets. Line 25: For high sea-ice concentration range they are slightly underestimated, especially for OSI-450.

Page 23 line 1: Confusing wording. Ground truth locations are not outside expanded maximum ice climatology? Line 17: Reword "it is also designed to remove" as "it also has the effect of removing", as this is a side-effect of the filter, rather than a planned part of the design. Line 24: replace "these wavelengths" with "the wavelengths of the PMR channels" for clarity.

Page 24 line 3: Unclear what is meant by "at most" in this context line 13: add "AMSR-E and AMSR2" before "channels" for clarity. Line 21: add "variable" after "raw_ice_conc_values" for clarity. Line 29: Expand "ECV" acronym here

Page 25 line 5: change "two components" to "two algorithm components" for clarity line 10: add section number after "Outlook" line 16: add "data" after "AMSR2" line 23: add

"channel" before "frequencies" for clarity. Lines 26&31: add "closed" before "sea ice"

Page 26 lines 18-19: The level itself is not stable, though always remains below 15% - needs rewording. Also not accurate to say "well below 15% SIC threshold" for SICCI-50km. Lines 23&24: Confusing wording: "maximum 1%" and then "a couple of percent" - needs rewording.

Page 27 line 20: expand SIE acronym line 24: Add some more information on plans to implement improvements for CDRs into operational processing chains (a few lines).

Page 28 line 6: expand EO acronym (and use acronym on line 7) line 23: URL for CMEMS is "marine.copernicus.eu" (there is a typo)

Page 29 lines 2&3: Confusing wording – is it the first satellite or the first satellite with MWI? Reword. Line 14: Add "channel" before frequencies for clarity. Line 15: This implies users should combine the products (which they shouldn't if they want a consistent product). Clarify that different products are available for different user needs. Lines 25-27: how can this be used? Users will treat uncertainties provided with data as the observation uncertainty.

Page 30 Line 7: Would be useful to provide URLs for the data archives. Line 18: Update this, says "[Indicate subset used]" References in general: Provide URLs if available for Technical Reports etc. Some DOIs have come out as links and others not.

Figures and tables: Some acronyms are in figure and table captions before being introduced in the text. Suggest defining in captions.

Figure 1: Add section number for Outlook.

Figure 2: Add that L2 SIC is also swath, L3 is a single daily averaged file. Define acronyms used in figure in caption.

Figure 3: Title should be "AMSR-E" (currently "AMSR"). Labels in the figure need to

be closer to the points (or colour coding would help). In figure caption, give section numbers where BFM and OWF are described in the text. "mean water signature" should be "mean open water signature"

Figure 4: Left plot: Label "BRI", "BPM" and "BFM" on plot. Add theta label on plot. "u" in caption is labelled "U" on plot, make consistent. Axis labels should also match convention in caption, e.g. "37H" rather than "h37". Right plot: "Freq. Mode" should be "BFM", "Bristol" should be "BRI". Add "theta" symbol to "Rotation angle" axis label.

Figure 5: Centre panel: Difficult to see any detail using this scale, needs to be shortened. Doesn't have to be the same as left panel as showing different variables.

Figure 6: Need to show 0% as white (or similar) for SIC plots so can see detail around ice edge. Would also be helpful to plot ice_conc minus raw_ice_conc_values.

Figure 7: Need to show 0% as white (or similar) so can see detail at low uncertainties.

Figure 8: Figure legend - datasets should be capitalised for consistency

Figure 9: Figure legend - datasets should be capitalised for consistency. If SICCI-25km and OSI-450 lines were thinner (like SICCI-50km) it would be easier to see the lines for both hemispheres.

Figures 10, 11: Specify that the sea ice concentration is uncorrected. Numbers in parentheses are in front of the season, not behind. Unclear - "Numbers below the season denote the mean SIC plus/minus one standard deviation" - there's only one number so how can this be plus/minus? Also Figure 11: The SH plots are "bumpier" than the NH plots – add comments on this.

Figure 12: Standard error is not mentioned in the text.

Table 1: Give months in the time period. Worth adding that grid is EASE grid. Caption: "entering" should be "entered in".

Table 2: Start date for DMSP SSM/I has an error ("090"), check table for other errors.

Technical corrections

General comments: Throughout, need to ensure there is a space between numbers and their units.

Throughout have used "..." or "etc...", should probably just be "etc." or sometimes "e.g." but check journal style guide.

Have referred to e.g. F10, F11 satellites, suggest using full name (include DMSP) at the start of the paper for clarification.

Specific comments: Some of the following are corrections of grammatical errors, and some are rewording suggestions to improve the readability of the paper.

Page 2 line 4: "allow" should be "allows" line 5: "are" should be "is" line 6: "to understand" should be "for understanding" line 11: "are" should be "is", "have" should be "has" line 28: unclear what you mean by "possibly" in this context, if it's the possibility that filtering can be applied needs rewording

Page 3 line 17: remove "up-front" here, reads a bit strangely in this context. Also, "entering" should be "entered"

Page 4 line 3: "some" should be "a" line 4: "more" should be "most" line 6: give section number for Outlook. Line 20/21: "Such wavelength" should be "Such a wavelength" line 23: replace "needed for" with "used in" line 25: add "(Table 2)" after "channels" line 31: "diameters" should be "diameter"

Page 5 line 1: I think "One" should be "Two", also change "swath" to "swaths" line 2: change "orbit" to "orbits", "extent" to "extents" line 5: expand CM-SAF acronym line 7: "directly accessed directly" should be "accessed directly", "Japan space agency" should be "Japan Aerospace Exploration Agency" line 15: "contribution" should be "contributions" line 18: "ERA-Interim" should be "ERA-Interim reanalysis" line 20: "ERA-Interim prior" should be "ERA-Interim data prior" (or similar), "early period with" should be "earliest period of" line 24: "from" should be "of", "for" should be "in" line

27: "operated to process" should be "for" line 32: "(L3) collects" should be "(L3) chain collects"

Page 6 line 3: "apply" should be "applies", "format" should be "formats" line 5: "similarly" should be "similar" line 29: define OW (given above in context of algorithm but worth defining here again), same for CI line 30.

Page 7 line 1: "TB in point" should be "TB at point", similarly "lines in point" should be "lines at point" line 2: "and geometric" should be "and a geometric" line 6: remove "originally", "describes" should be "describe" line 20: "onto" should be "on" line 25: "cope for" should be "cope with"

Page 8 line 1: would read better as "Figure 4 (right panel) also shows that the optimum..." line 16: "space" should be "spaces" lines 19/20: replace arrows with " " line 26: "section so" should be "section has so" line 32: "by Eq. 1" should be "using Eq. 1"

Page 9 line 20: "was" should be "has been", comma before "which" line 21: "varies" should be "vary", "follows" should be "follow" line 25: "yield highest" should be "yield the highest" line 26: "yield departure" should be "yield a departure" line 27 and 28: "departure" should be "departures"

Page 10 line 3: add commas both before and after "the uncorrected SIC value" line 7: "re-analysis" should be "re-analyses" line 20: Add "For Tb_nwp" at the start of the line line 22: Add "For Tb_ref" at start of sentence before "Theta_instru" line 26: remove "for F10" (already mentioned in this sentence) line 29: "for being" should be "to be"

Page 11 line 3: "allows" should be "allowed" line 9: "ones" would read better as "datasets". Also, having introduced the acronym WFs should use on this line instead of "Weather filters" (also on line 20). Line 14: no hyphen in unaffected line 18: suggest changing "so far did not adopt" to "have so far not adopted", also "from" should be "in" line 19: change "by using adhoc status flags" to "on an adhoc basis by using status flags" (as the flags themselves are not adhoc) line 22: "re-used" should be "reuse" (or

"have reused") line 23: Suggest add "For example," before Spreen et al. Line 25: "to" should be "with" line 26: "with" should be "for", "for which" should be "where", suggest changing "threshold is 0.053" to "threshold is set to 0.053"

Page 12 line 2: "intersect" should be "intersects" line 6: missing close bracket after (A-D), also "illustration how" should be "illustration of how" line 7: "into" should be "in" line 11: add "and" before "the varying effects" line 12: suggest replace "not remove" with "avoid removing" line 13: "show" should be "shown" lines 16,17: "T" should be in italics line 20: "naming" should be "name" line 22: change "is set to" to "will" as this is an unintended consequence line 23: suggest changing "we rather refer to such filters as 'Open Water Filter'" to "we refer to such a filter as an 'Open Water Filter'", also suggest "add" changed to "include" line 24: "are" should be "is" line 27: "Noticeably" should be "Notably" line 28: "attached a" should be "attached to a", also change "as to if the OWF detected it" to "corresponding to OWF detection"

Page 13 lines 1&2: "high concentration range" should be either "a high concentration range" or "high concentration ranges line 5: remove "likewise" line 10: change "best appear" to "are best shown" line 11: "T" in B_CI(T) should be bold lines 14&15: "constantly" should be "consistently" line 28: "Laptev and Kara Sea" should be "the Laptev and Kara Seas" line 29: would read better to remove "old" after "2 years", also "on right panel" should be "on the right panel"

Page 14 line 6: "north for Canadian" should be "north of Canadian" line 8: remove "that what" line 9: change "and" to "which" after "section 3.4.1," line 19: "data is assimilated" should be "data are assimilated" line 21: "those" should be "that" line 28: "algorithm to retrieve" should be "algorithm for retrieving"

Page 15 line 1: suggest change "relevant to discuss" to "relevant for discussion of" line 4: "Earth surface" should be "the Earth's surface" line 7: remove "that is" line 9: "cells" should be "cell" line 20: remove "shortly" line 21: suggest change "presenting less" to "have undergone little" line 28: "details" should be "detail" line 30: suggest

change "among others" to "including" line 31: remove "is computed" line 32: change "the antenna pattern functions are approximated" to "the approximation of antenna pattern functions" line 33: "from central" should be "from the central"

Page 16 line 1: "for contribution" should be "for the contribution" line 4: "were" should be "have been" line 5: "where" should be "were" line 7: "as input" should be "as the input" line 13: "were" should be "was", suggest change "base" to "basis", "pixel" should be "pixels" line 18: "in SH" should be "in the SH" line 19&20: change "where to select the Open Water training samples" to "where the Open Water training samples were selected" line 26: "conversely" should be "converse", suggest "CDR of" should be "CDR method of"

Page 17 line 14: "of SICCI-25km" should be "of the SICCI-25km" line 22: "file" should be "files" line 27: "Bottom" should be "The bottom" line 29: "corresponds" should be "correspond" line 32: "by OWF" should be "by the OWF"

Page 18 line 6: "indicate" should be "indicates" line 14: replace "are covered by" with "cover" line 18: Suggest replace "several" with "three", "One" with "The first" line 19: "its" should be "their" line 25: no hyphen in intermediate

Page 19 line 9: "albeit" should be "despite" line 12: "from" should be "for" line 15: "from" should be "for" line 16: "improve much" should be "much improve" line 18: "parametrization" should be "parametrizations" line 21: "from with" should be "for", also "were" should be "where" line 25: "sensibly" - do you mean "ostensibly"?

Page 20 line 9: remove "at best" and add "ideally" before "preserving" line 19: reword "very little few jumps are" to "very little change is" (or similar) line 23: could remove "lowest" and "highest" as it's already clear this is the range

Page 21 line 10: remove comma after "but" line 14: "details" should be "detail" line 21: "East Antarctic" should be "the East Antarctic" (or "East Antarctica") line 23: remove "being", suggest replace "by too" with "with" lines 29&30: suggest move "than for the

other two CDRs" after "more" on line 29 line 30: "e.g." should be "i.e." line 31: add "for all three CDRs" after "2%". lines 32&33: suggest change "less good than that" to "poorer than"

Page 22 lines 5&6: change "Arctic" to "the Arctic"

Page 23 line 3: Suggest reword "can be picked" to "may be selected" line 4: Suggest reword "and to the least at the location of the ground-truth estimates used in the section" to "where the ground-truth estimates used in the section are located" line 5: Change "More developed" to "A more developed", "as wetter" to "as a wetter" line 6: Change "We finally" to "Finally, we" line 12: "in large extent" needs to be reworded, perhaps replace with "generally" or "to a large extent" line 13: Capitalisation of "Passive Microwave" varies, be consistent line 15: "on combination" should be "on a combination", also need to define acronym "PMR" line 19: "take" should be "pay" line 20: "in field" should be "in the field", also use "OWF" acronym for consistency line 21: "are pertaining" should be "pertain" line 25: Remove "distinguishing between" and add "to be distinguished" to end of sentence.

Page 24 line 1: "aims" should be "aim", also "from interested" should be "from the interested" line 6: replace "was" with "were" twice line 17: "is 'spilling' " should be " 'spills' ", also "appear" would be better than "look" line 19: "foot-print" sometimes has a hyphen, sometimes not, needs to be consistent line 20: "instrument" should be "instruments" line 29: "improvement" should be "improvements"

Page 25: line 7: "on March 1985" should be "in March 1985" line 8: add "dataset" before "only", also add "on" before "09 July" lines 9&10: change "achieving" to "to achieve" line 11: suggest reword "algorithms allowed e.g. to consistently process SIC" to "algorithms also allowed consistent processing of SIC" line 14: change "15 years record" to "15-year record" line 18: change "will" to "would" line 20: change "had met" to "would meet" line 33: add "the" before "coarsest"

Page 26 lines 3&4: Confusingly worded: "seasonal cycle of sea-ice and snow properties during summer". Should this be sea ice extent? (Also be consistent throughout about whether to use a hyphen in sea ice or not) line 15: "than" should be "as" line 17: Suggest remove "For all practical purposes" line 29: Remove "namely" line 33: remove "that"

Page 27 line 1: "impact of melting" should be "impact that melting" line 3: Suggest change "more efforts" to "further effort" line 4: Remove "same" and "that was" line 10: "if to selecting" should be "if selecting" line 12: "dimension" should be "dimensions" line 17: "the sea ice cover, sea ice area" should be "of sea ice cover, and sea ice area" line 27: "exploring" should be "exploration" line 28: "channels" should be "channel", "that" should be "than" line 30: Suggest change "could not be better embedded by SIC" to "could be better embedded in SIC" line 32: "Filter" should be "Filters"

Page 28: line 2: Suggest adding "still" after "can" line 24: Change "at best" to "to achieve" line 33: "passed" should be "past"

Page 29 line 5: Add "However," before "Because" (as shouldn't start a sentence with because) line 17: "product contains" should be "products contain" as there is more than one product. Line 19: Replace "on the hand a" with "ease of", "product" should be "products" line 20: remove "of all products", "is" should be "are", "has" should be "have" line 25: "this provides" should be "this paper provides" (or similar)

Page 30 line 6: "making" should be "make".

---

## Author Response (AR1)

In this document, the authors provide answers to the two reviews of paper tc-2018-127.

Lavergne, T., Sørensen, A. M., Kern, S., Tonboe, R., Notz, D., Aaboe, S., Bell, L., Dybkjær, G., Eastwood, S., Gabarro, C., Heygster, G., Killie, M. A., Kreiner, M. B., Lavelle, J., Saldo, R., Sandven, S., and Pedersen, L. T.: Version 2 of the EUMETSAT OSI SAF and ESA CCI Sea Ice Concentration Climate Data Records, The Cryosphere Discuss., https://doi.org/10.5194/tc-2018-127, in review, 2018.

We thank the two anonymous reviewers for thorough comments on our manuscript, and many suggestions to improve both the content and the language.

**Anonymous Referee #1**

**Summary:**

This paper describes a new version of the OSISAF sea ice concentration product and the ESA sea ice CDR. The products are derived from passive microwave data. The new version includes several enhancements from the Version 1 OSISAF product. Comparisons with independent estimates show good agreement. The new version provides a consistent record of sea ice concentrations for the scientific community.

**General Comment:**

The manuscript provides a thorough introduction of the new versions. The description of the algorithm and processing, including enhancements from Version 1 is clear and detailed. The initial evaluation results look reasonable and given that it builds on the previous version and thorough earlier validation, they are quite sufficient to provide high confidence in the quality of the product. The Level 4 filtered product is particularly beneficial for users who wish to have a "clean" concentration estimate and this is an excellent improvement from Version 1. I have only a few minor comments that the authors should address before publication.

Thank you for your positive appreciation of our manuscript. Your comments are very valuable and are addressed below.

(One further general comment: it would be helpful for readability to either indent new paragraphs and/or skip a line between paragraphs.)

This is done in the revised manuscript.

Specific Comments (by page and line number):

*P2, L9: while the albedo specifically depends on concentration, it is not only concentration: snow melt state and particularly melt ponds substantially affect albedo even for 100% concentration.*

We agree and have changed the wording to now read "For example, the albedo of the polar oceans is strongly influenced by sea-ice concentration"

P4, L24-25: I find the (resp. XXXX) style awkward to read and somewhat confusing. I would just write out each in sequence rather than using parenthesis, but this may just be a preference by me.

We changed this sentence to read: "Although not identical, the spatial resolution of the channels needed for the SIC algorithms is similar for the three coarse resolution sensor series (SMMR, SSM/I, and SSMIS) with about 70x45 km instantaneous Field-of-View (iFoV) diameters for the 19 GHz frequency channels, and 38x30 km for the 37 GHz ones (Table 2)".

**P6, L1: "daily composited fields of SIC" – how is the compositing done? Is it simply drop-in-the-bucket?**

We use a weighted average with Gaussian spatial weights, and equal weights in the temporal domain. This is added to section 3.6 where gridding and daily compositing was briefly covered.

P10, L12-20: What are the uncertainties in the NWP fields and the RTM? While the dynamical tiepoints and the double-difference approach may negate much of the influence, I do wonder how effective the correction is if the NWP data and/or the RTM have high uncertainties? This feeds into my next comment below.

The quality of RTMs and NWP fields indeed play a role in the effectiveness of the  $T_B$  correction. In our experience however, the  $T_B$  correction always yields more accurate SIC fields over open water when correcting for wind speed, and water vapour. As others we do not correct for Cloud Liquid Vapour (L) which is not reliable in NWP data (see next comment).

P10, L28: the use of the NWP fields is novel and I like the physical approach. However, L is not reliable from NWP. Isn't L one of the largest if not generally the largest source of emission, at least over open water. So not being able to correct for that really limits the effectiveness of the NWP correction, doesn't it? The use of weather filters in the Level 4 fields eliminates this, which is good, but the quality of the Level 3 fields must be limited, right?

The use of NWP fields and RTM-based correction schemes is one of the specificities of the OSI SAF approach, and was introduced in Andersen et al. (2007). So far, Cloud Liquid Water (CLW, symbol L in the paper) in global NWP field has rarely been found to be reliable enough for correcting  $T_B$ . One factor Is probably that the modelled fields are not at the same temporal and spatial scales as the satellite data.

Although CLW is not used, the correction based only on 10m wind speed and water vapour is still quite efficient (see for example the offset of 1% to 1.5% standard deviation in Figure 8). Correcting  $T_B$  for WS and WV leads to a 35% reduction in  $T_B$  variance at 18/19 GHz and 22% reduction in  $T_B$  variance at 37 GHz. This subsequently leads to a 35-45% reduction of SIC variance for standard algorithms (ESA SICCI PVASR p151-159). The noise associated to CLW is rather localized on small geographical domains, and is indeed taken care of by the weather filter (and maximum extent climatology) at Level-4. As in previous version of the CDR (Tonboe et al. 2016), the Level-3 product files indeed present some remaining noise, more pronounced in the case of high CLW. Noticeably: the statistics of the remaining noise is integrated in the uncertainty fields: not correcting for CLW leads to higher product uncertainties.

P13, L17: It might be worth considering showing an example of the "ice curve". I can generally visualize, but a figure would perhaps better illustrate it. Yes, this is added to Figure 3, and discussed in section 3.4.3.

**P13, L20-26: I'm not sure I understand Figure 5. It appears to show an increase in open water concentration near the ice edge due to the correction (e.g., in Barents Sea and Davis Strait regions). Is that correct? Wouldn't that reduce the quality if the correction essentially added ice to open water regions?**

You are referring to the "open water" region of Figure 5 left panel (outside the 15% SIC contour). This part of the plot was actually not described nor discussed in the text, something that was clearly missing and triggers your comment (also from Reviewer #2).

Figure 5 (left and center panels) shows the effect of the total correction, including both the correction due to the ice curve (described in this section 3.4.3), and the effect of the RTM-based atmospheric correction (section 3.4.1). The ice curve correction has most of its impact in high-concentration regions (inside the 70% SIC contour) while the atmospheric correction has most of its impact over open water regions (outside the 15% SIC contour). Outside the 15% SIC contour, it is correct that Figure 5 (left) shows increase in SIC after correction. This was confirmed by plotting similar maps for other months. This is because the SIC before correction SICucorr is mostly slightly negative there, and the correction step brings it closer to 0%. This is linked to the way our OW tie-point is tuned. As explained in section 3.4, the OW tie-point is tuned dynamically against open ocean cases that are outside a maximum ice extent climatology, thus potentially more representative of "open ocean" Tie-point than the conditions closer to the edge. Prior to atmospheric correction, the open-water tie-point is thus "warmer" than the TB conditions closer to the edge, thus the uncorrected SICs are slightly negative there. After correction, our OW tie-point is re-tuned and is more representative of  $T_{\rm B}$  close to the ice edge, hence the increase (reds in Figure 5). The net effect is a reduction in variability over ocean (blues on Figure 5, center panel) which indicates that the atmospheric correction step on average does a good job reducing weather-induced noise over the open ocean.

Your comment prompted several edits: in section 3.4.1 we added that the effect of the RTM-based correction is largest over open water, and very limited over sea-ice. in section 3.4.3, we stated the "ice curve" correction has most impact over consolidated ice, and little effect over open water. Then we started that Figure 5 shows the combined effects of both correction. We also reworked the description of Figure 5 to first address the "ice curve" correction (including the discussion with ice-age on right panel), before addressing the atmospheric correction (including addressing your specific comment above). This results mostly in a re-arranging of text for improved readability. Thank you for this comment.

P16, L3: This should be discussed further – why is the gridded land-spillover correction still needed after the swath correction? How much coastal contamination remains after the swath correction. If the swath correction is not sufficient on its own, is it worth doing – i.e., would the Cavalieri correction work just as well without the swath correction? I guess the basic question is whether there is a benefit to doing both corrections or is the Cavalieri correction just as good? If so, then why do the swath correction?

We believe there is a benefit of combining the two approaches: first perform a physically based correction, then a statistical-based correction/filtering. We have however not studied in details if the statistical method alone could have done a good enough job alone, and cannot answer your (very valid) questions above. We added the following text in section 5.2 "Outlooks" when discussing algorithm improvements.

Other steps in the processing chain can further be improved upon, e.g. the land spill-over correction schemes. In section 3.6 we described how land spill-over was corrected in two steps, first through a physically-based algorithm on swath TB data (adapted from Maass and Kaleschke, 2010), followed by a statistically-based correction of gridded SICs (adapted from Cavalieri et al. 1999). Several reasons can have led to the swath-based correction to not be enough. For example, the method relies heavily on accurate geolocation of the TB measurements, however its uncertainty for the SSM/I and SSMIS instrument is known to be large (Poe et al. 2008), and is not corrected for in the current version of the FCDR (R3) we used (Fennig et al. 2017). We used approximated iFoVs weighting functions instead of eFoVs (see section 2.1) when convolving antenna pattern with the land mask, thus neglecting the effect of the measurements integration period. Finally, strategies to avoid gridding land-contaminated FoVs when building Level 3 maps might help in the future. It will also be beneficial to use high-resolution SIC maps from coastal regions (e.g. from navigational ice charts) to tune the various thresholds embedded in the statistically-based correction. To improve further on the land spill-over correction will be an objective for upcoming versions of the CDRs.

*P16, L31: "basic isotropic schemes" is not very specific. Is it a bi-linear interpolation?* It is a interpolation with gaussian weights of the distance. This is now specified in the text.

P19, L17-23: I can understand that the ERA-Interim fields are not as good earlier in the record and thus the correction for SMMR is not as good. However, there is a noticeable step-change between SMMR and SSMI in Figure 8. Did ERA-Interim undergo a step change in terms of data sources or other processing quality at the same time? If not, then it seems like it's not ERA-Interim (or at least not only), but rather something else causing the step change. Perhaps it's related to the change in frequency from 18 GHz for SMMR to 19.3 GHz for SSMI?

This is a very good point, also made by Reviewer #2. We added a sentence discussing the impact of 18.0 GHz Ku-band.

Concerning the quality of ERA-Interim in the SMMR era: the main ERA-Interim reference is Dee et al. 2011, but it only describes the "1st production stream" of ERA-Interim (post 1989). A second stream covering 1979-1989 was added at a later stage, but there are no publications. We contacted the ECMWF team, and obtained a personal communication that can give some insight. We modified the manuscript P19 to read: "Another plausible explanation would be that the re-analysed fields for wind speed and water vapour from ERA-Interim are less accurate in the SMMR era than in the SSM/I and SSMIS era. We note that clear-sky radiances from SSM/I and SSMIS were directly assimilated in ERA-Interim over the ocean (Dee et al. 2011), but not SMMR radiances (Paul Poli, personal communication). This can especially have an impact in the SH, were other sources of conventional observations are scarcer".

**P21, L4: One thing not discussed is the potential impact of satellite crossing times on the retrievals. I assume the dynamic tiepoints should handle these discrepancies, but it might be worth mentioning.**

The dynamic tuning of tiepoints and OWF threshold work with samples gathered at an hemispheric scale, and over a [-7:+7d] sliding time window. This technique can thus not handle intra-daily differences -arising from one region to the next- that are due to not observing the surface at the same time. It can however mitigate the potential impacts due to different missions observing at different times (if any). We added the following sentence when describing how OWFs are tuned and applied at Level-2 (P12): "To compute OWFs at Level 2 can also help mitigate the potential impacts of changes in satellite crossing time between different missions".

P25, L12: Why not produce a 12.5 km or 10 km resolution AMSR-E and/or AMSR2 product, i.e., using the same channels (19, 37 GHz) as for SMMR-SSMI-SSMIS, but obtaining a higher spatial resolution for the period of 2002-present? It seems like this

would be more beneficial than at least the 25 km SICCI. I can see a benefit of using the 6V channel for the 50 km product, but that isn't in the 25 km SICCI.

This is an excellent question. The channels we use for AMSR-E and AMSR2 have the following iFoVs (reproduced from Table 2).

| IFoV   | 18.7GHz  | 36.5GHz |
|--------|----------|---------|
| AMSR-E | 16x27 km | 9x14 km |
| AMSR2  | 14x22 km | 7x12 km |

They all have a 10x10 km spacing. The diameters given here are those of the 3dB ellipses of the main lobe of the antenna pattern. Considering in addition that eFoVs are larger (mostly in the across track direction), we approximate that the eFoV diameter for the 18.7GHz channels are about 25km, while the eFoVs of 36.5GHz are about 15km. These two resolution are then merged into a SIC algorithm (that uses one 18.7GHz channel, and two 36.5GHz channels). What is the spatial resolution (eFoV) of the computed SIC? Probably somewhere between 15km and 25km, but in any case larger than 10km (the spacing) or 12.5km (half the grid spacing used for OSI-450).

The choice of a grid spacing for SIC products is very much based on "feelings" and historical reasons. Because the SSM/I brightness temperature daily maps were originally provided on a polar stereographic 25km grid, the NOAA/NSIDC SIC CDR is also on a 25km grid, and OSI-450 as well. This is probably too fine a spacing as we discuss in our section 4.3. The choice of 25km grid for the SICCI product based on 19 and 37 GHz channels from AMSR-E and AMSR2 is potentially too conservative (not by much), but this choice was made to ease uptake by users (only one spacing to refer to).

Your comment prompted the following revision:

A sentence was added section 4.3: "The true resolution of the SICCI-25km CDR might be slightly better than 25x25 km, but this grid spacing was retained to ease uptake by users, and comparison with OSI-450."

A sentence was added section 5.2 (Outlooks) when discussing needs for further research efforts: "Finally, research is needed to assign a true spatial resolution to SIC fields computed from combinations of n  $T_B$  channels, themselves at different spatial resolutions. Some knowledge is embedded in our parametrization of smear, but it is currently not enough to e.g. choose and fully justify a grid spacing for SIC data records."

**Minor Comments (by page and line number):**

P3, L17: use "in" instead of "entering"
P3, L26: use "share" or "provide" instead of "keep"
P16, L26: use "contrasts with" instead of "is conversely to"
P24, L3: "aiming at most complete" to "aiming to produce the most complete daily maps possible"
P25, L11: use "allowed, e.g., consistent processing of SIC CDRs. ..."
P26, L18: use "on the order of. . ."
P27, L1-2: use "the impact that melting and melt-ponds have. ..."
P27, L10: use "could be investigated if selecting. . .."
P28, L24: use "aim to have the best temporal consistency. . ."
Thank you, all your suggestions were implemented.

**Anonymous Referee #2**

This paper gives a thorough, informative and detailed description of three important new climate data records of sea ice concentration. The science in the paper is comprehensive. I therefore only have suggestions for minor improvements (though there are quite a few) - mostly for clarifications to the text. The paper is clear and easy to read, despite a number of minor grammatical errors which are detailed below.

Thank you for your positive evaluation of our manuscript. Your "quite a few" suggestions for minor improvements were processed thoroughly and led to an improvement of our text and figures. Thank you for having taken the time.

**Minor comments**

**Page 2 line 1: Is this the observation uncertainty in assimilation for models? Unclear, need to elaborate**

We have added a sentence to clarify this statement: "This is because both the bias correction of large-scale climate models and the extrapolation of observed relationships between forcing and sea-ice coverage can only be carried out robustly if observational uncertainty is sufficiently small."

*line 32: quantify what you mean by "coarse resolution"* Done: "coarse resolution (30-60 km)".

Page 3 line 4: quantify what you mean by "medium resolution" line 19/20 & 22 (and throughout): Why only an "initial evaluation". Reading on shows that you have done

**more than just a cursory evaluation which is what this wording implies. Suggest reword.**

Done: "medium resolution (15-25 km)".

"initial" is here meant as "a first set of evaluation results". More evaluation is underway, that will be published at a later stage. Since both reviewers estimate that the evaluation presented in this manuscript is enough for a publication, we will remove "initial". That more evaluation will come in later publications is already announced in our Outlooks section.

Page 4 line 2: Suggest mentioning data gap in AMSR data earlier, perhaps when introducing Table 1. line 4: Suggest "documented in Table 2" should be "documented in the comments in Table 2". Would also be useful to have a full list of outages, perhaps a link to this in another document? Line 10 (and Table 2): "width of the polar observation hole" is not given, it's the bit that's viewed rather than the hole, also not a width as it's an angle, suggest rename this column line 23: Not sure that spatial resolution of SMMR is "somewhat similar" to SSM/I and SSMIS, suggest reword line 26: Clarify difference between sampling and resolution line 32: Consider showing eFoV in Table 2.

P4L2: this would require discussing acronyms earlier, we feel it is not worth the rewriting since the information comes shortly after. L4 done, we refer to the Product User Guides (PUGs) for extensive list of missing dates. L10&T2: done. L23 done (removed "somewhat"). L26: done (add sentence "The dimensions of the iFoV and eFoV are referred to as the resolution of the channels. The sampling is how close in space the FoVs are acquired. Most channels are thus oversampled."). L32: unfortunately there is no authoritative source for eFoVs across all the instruments. iFoVs is what is generally documented (e.g. at WMO OSCAR database).

**Page 5 line 4/5: Clarify if L1 data for SMMR, SSMI/S, SSMIS line 5: Add a line on what is an FCDR and what reprocessing has it undergone. Overlaps? Calibration? QC? line 6: add what period AMSR-E data covers line 9: more information needed on "resolution-matched"**

P5L4: done. L5: done (add sentence "In the FCDR, the  $T_B$  are re-computed from Antenna Temperatures (TA), screened and corrected for known artefacts like solar intrusion, and intercalibrated between missions.") L6: done, L9: done. The sentence is edited to read : "For both AMSR-E and AMSR2, the  $T_B$  are available both at their nominal resolution (documented in Table 2), and post-processed at lower resolution matching those of other channels (e.g. the 36.5GHz  $T_B$  at the resolution of the 6.9GHz channel). We use the nominal resolution of the  $T_B$  channels, not the resolution-matched ones."

Page 6 line 1: what type of grid? EASE? Line 2: what are the necessary steps? Can reference later on in paper if necessary lines 5-8: suggest moving these lines to page 5 line 32, after "flags". Would flow better. Line 18: clarify these numbers are sea ice fraction line 23: needs citation for BRI more accurate than BPM at high concentrations line 28: Figure 3 illustrates for AMSR-E data, example from Comiso (1986) is for SMMR. Need to clarify that these can be applied to other instruments.

P6L1: Yes. The type and definition of grids is covered later in the text. L2: the sentence was simplified to "The Level 4 (L4) chain fills the gaps, apply extra corrections, and format the data files that will appear in the CDR.". L5-8: done, L18: done, L23: same references as the sentence before, so we merged the two sentences. L28: the reference to SMMR was not needed and was removed.

Page 7 lines 4/5: show ice signatures on plot (mentions in text to left and right but not that clear) line 8: text says D-A, use A-D for consistency. Also A,D in figure 3 and D,A in figure 4, make consistent.

Well spotted. We made this consistent.

Page 8 line 4: What is the magnitude of the ice concentration change between algorithms for this example? Line 10: show theta = 90 on figure The improvement is only few tens of %s RMSE, but can be more significant in other conditions. We specified the [-90;90] range for Figure 4.

Page 9 line 5: Have you also used a sliding window? Wording implies not, if it is suggest adding "similarly" before sliding. Why was the window changed from +-15 to +-7 days? Line 13: Why can this be assumed? Expand. Line 15: and SMMR, SSM/I, SSMIS? Also remove "than". Line 15/16: suggest moving sentence beginning "Recent investigations..." to line 13, before "It is assumed..."

P9L5. Done. We add a sentence: "Our sliding window is made shorter so that tie-points react more rapidly to seasonal cycles, e.g. onset of melting." L13, L15, L15/16: all done by a refactoring: "As in Tonboe et al. (2016), the CI training sample is based on the results of the NASA Team (NT) algorithm (Cavalieri et al., 1984): locations for which the NT value is greater than 95% are used as a representation of 100 % ice. Recent investigations, e.g. during the ESA CCI Sea Ice projects confirmed that NT was an acceptable choice for the purpose of selecting closed-ice samples."

Page 11 First paragraph: This is confusing as it sounds like different RTMs for each instrument but is it actually different optimisations? Reword. Line 10: quantify what is meant by "rather large" line 27: Is there a citation for the ATBD document itself? From line 20: As not using GR2219v suggest editing this section as don't need to describe in detail or give previous examples.

P11: we re-worded to avoid confusion of different RTMs. L10 done ("sometimes up to 50%") L27: we are not aware of a citation for the ATBD. L20: we kept the text as-is.

Page 12 line 1: Would be helpful to use a different symbol other than T to avoid confusion with temperature line 26: Implies that

Donlon paper: done. Thank you for the update on the SST CCI land mask, we will act upon this for next version. L15&16: We added this information at the end of section 4.3: ("Ice resulting from freezing of fresh and brackish waters does not have the same emissivity as that from sea water. The retrieval of ice area fraction in these conditions would call for dedicated tie-points (e.g. Ghaffari et al. 2011), which we did not implement here. In addition to the difficulty of computing dynamic tie-points over such small areas, it is unclear if such dedicated tie-points would make a large difference in the end, because of the combination of many error sources in these close water bodies (land spill-over, thin sea-ice, larger atmospheric influence, etc...). A layer in the status\_flag variable indicates fresh and brackish water bodies."

Fully missing days: we added the sentence "Days with fully missing input data (e.g. every other day in the SMMR period) are not created by interpolation, and the files are missing."

Page 17 Evaluation of the data: Have you simply looked through the data? Issues where processing has gone wrong, or the data looks strange have previously been an issue for OSI SAF CDRs. It would be very helpful for users not to have to do this QC. Line 7: add what the ERA-Interim data is used for in the processing line 28: colour scale is blue-red, not blue-yellow-red line 29: Is noise just characterised as below 10%? line 33: suggest move "as nominally returned by the SIC algorithm" to line 27 after "raw\_ice\_conc\_values"

P17: The data was thoroughly looked at. We hope no artefacts are left. The situation should also be improved wrt OSISAF v1 thanks to using QCed FCDR as input (instead of an archive of operational data stream). L7 ERA-Interim: Done. L28: done, L29: no, "noise" characterises that the true SIC is 0% (unless close to the edge), before the OWF is applied. L33: done.

**Page 18 line 16: what about summer?**

Good question. The following sentence was added: "During summer, sigma\_algo is larger by few percents, and the increased variability inside the ice pack yields higher sigma\_smear, leading to larger sigma\_tot."

**Page 19 line 23: SMMR uncorrected is also better than for SSM/I and SSMIS, particularly in the NH. Why? Line 30: in winter? Line 31: need to give seasonal figures**

P19, L23: Indeed, SMMR uncorrected is also better than SSM/I and SSMIS. This is due to the center frequency of the Ku-band channel (18GHz) being farther away from the water vapour absorption line (22GHz) than the SSM/I channel (19.3GHz). 18GHz is less influenced by water vapour. This explanation was added in the manuscript.

L30 and L31: the offset between SICCI-50km and the others is mostly constant in all seasons.

Page 20 line 4: "internally consistent" - do you mean consistent over time? Line 5: Can't tell from figure 8 that it's the smallest possible. Suggest reword "and smallest possible retrieval noise" to "and a small retrieval noise" line 14: change "thus after the OWF is applied" to "thus after all the filters including the OWF are applied" for clarification line 17&18: as the range changes are they stable with time? Also need to give separate summer and winter values and incorporate line 20 in the discussion. Also separate summer and winter values line 21. line 27: might be worth adding that this is addressed as future work later in the paper line 33: Need to elaborate on how this could cause an increase over time

P20L4&5: clarified as suggested. L14: done as suggested. L17&18 We added a values for summer and winter. L33: this is an hypothesis, and is now clearly marked as such. The mechanism would go via improving atmospheric correction via better

re-analysis field, that would lead to stronger separation of the projection plane in (19v,37v,37h) and the (19v,37v) OWF plane. We changed the sentence to: "The departure of the optimal SIC data plane from the OWF plane (by convention at theta=0°, see right-hand side panel in Figure 4) could be the cause for the slight increase of the 1%-percentile curves of OSI-450 during the time period (via an improvement of the reanalysis data entering the atmosphere correction step over time), and the different value obtained with SICCI-25km".

**Page 21 lines 17&18: Why 2 months in summer and 3 months in winter?**

The motivation doing so is the temporal duration of sea-ice conditions being close to the annual sea-ice extent minimum and maximum. This period lasts longer in winter than summer. We also chose to limit the comparison to these months because the climatological ocean mask varies least during these time periods and allows us to put the locations of the reference 0% sea-ice concentration as close as possible to the maximum extent of sea ice. This way we make sure to perform the evaluation in "polar"-type waters and atmospheric conditions.

*line 23: Give the T2m threshold (if not mentioned elsewhere?)*

The T2m threshold is +5C, this is now added in the text.

*line 25: "skewed a bit" – could quantify the skewness, or reword to "slight negative skew" or similar, and elsewhere.*

Reworded.

Line 27,28,29: should refer to Figure 12, not Figure 10.

To refer to Figure 12 instead of 10 is correct. Done.

Values given are not the same as on Figure 10, unclear. Line 33: In winter it looks fairly similar though.

This was a rounding issue in the figure text. Figure 10 (and 12) are now revised to show the same values as in the text.

**Page 22 line 3: reference "(Figure 12)" after "100%"**

The reference to Figure 11 at the end of the sentence is actually covering quite well the information given on this sentence, not changed.

*line 4: should be Figure 12, not Figure 11* Indeed, this was changed.

*line 7: Suggest replace "less good" with something like "poorer, but still acceptable". Suggest cut the last sentence of this paragraph as is a repetition.*

We replaced "less good accuracy" by "slightly larger bias", and removed last sentence.

Line 12: The total uncertainty is described as "standard error" on Figure 12, need to reword this.

This is now better captured in the caption of Figure 12: black error bars are for plus/minus one standard deviation of the standard error, while blue error bars are for plus/minus one standard deviation of the total uncertainties.

Paragraph around line 20: Elaborate on why uncertainties for SICCI-50km are smaller than for the other two datasets.

The following sentence was added: "These results are in agreement with those introduced in section 4.2.1 and are mainly explained but the frequency channels used in the three CDRs: 18.7 GHz for SICCI-25km, instead of 19.3 GHz for OSI-450 (less noise contribution from atmospheric water vapour content), and 6.9 GHz for SICCI-50km (smaller sensitivity to atmosphere and surface snow and sea-ice property variations)."

*Line 25: For high sea-ice concentration range they are slightly underestimated, especially for OSI-450.*

Indeed. We reworded the sentence to: "Thus, the results summarized in Figure 12 indicate that the uncertainty tot provided with the three CDRs are slightly underestimated, especially for OSI-450, for the high sea-ice concentration range (SIC = 100%), and are slightly overestimated for the low sea-ice concentration range (SIC = 0%)."

Page 23 line 1: Confusing wording. Ground truth locations are not outside expanded maximum ice climatology?

This was reworded as: "For SIC = 0%, the ground-truth open water locations are selected just outside the maximum sea-ice climatology, while we used an expanded version of this climatology for the selection of the open water training data samples (sections 3.3 and 3.6)"

Line 17: Reword "it is also designed to remove" as "it also has the effect of removing", as this is a side-effect of the filter, rather than a planned part of the design. Line 24: replace "these wavelengths" with "the wavelengths of the PMR channels" for clarity.

P23L17: done as suggested. L24: done as suggested.

Page 24 line 3: Unclear what is meant by "at most" in this context line 13: add "AMSR-E and AMSR2" before "channels" for clarity. Line 21: add "variable" after "raw\_ice\_conc\_values" for clarity. Line 29: Expand "ECV" acronym here P24L3: reworded. L13: done. L21: done, L29: done.

Page 25 line 5: change "two components" to "two algorithm components" for clarity line 10: add section number after "Outlook" line 16: add "data" after "AMSR2" line 23: add "channel" before "frequencies" for clarity. Lines 26&31: add "closed" before "sea ice"

P25L5: done, L10: done, L16: done L23: done, L26&31: done

Page 26 lines 18-19: The level itself is not stable, though always remains below 15% - needs rewording. Also not accurate to say "well below 15% SIC threshold" for SICCI50km. Lines 23&24: Confusing wording: "maximum 1%" and then "a couple of percent" - needs rewording.

P26L18/19: we reworded but still find that this is quite stable over >30 years. L23&24: fixed (kept couple of percent).

Page 27 line 20: expand SIE acronym line 24: Add some more information on plans to implement improvements for CDRs into operational processing chains (a few lines).

P27L20: done. L24: done, but on the page after (when discussing ICDR).

Page 28 line 6: expand EO acronym (and use acronym on line 7) line 23: URL for CMEMS is "marine.copernicus.eu" (there is a typo) P28L6: done. L23: done (thanks!)

Page 29 lines 2&3: Confusing wording – is it the first satellite or the first satellite with *MWI*? Reword. Line 14: Add "channel" before frequencies for clarity. Line 15: This implies users should combine the products (which they shouldn't if they want a consistent product). Clarify that different products are available for different user needs. Lines 25-27: how can this be used? Users will treat uncertainties provided with data as the observation uncertainty

P29L2&3: We reworded: "The first satellite of the European Polar System Second Generation (EPS-SG) series to carry a Microwave Imager (MWI) is scheduled for launch in 2023." L15: interesting question. Users can combine information they retrieve separately from the three datasets. They can also attempt the combination of the products, but have to take into account the difference in spatial resolutions, which requires more advanced techniques that we could use here. We did not modify the text. L25/27: based on our evaluation of the observation uncertainties, users 1) are confident that our uncertainties mostly correspond to the statistical observed error, and 2) our uncertainties are slightly too large over open water, and users can thus decide to shrink them a bit if relevant for their application.

Page 30 Line 7: Would be useful to provide URLs for the data archives. Line 18: Update this, says "[Indicate subset used]" References in general: Provide URLs if available for Technical Reports etc. Some DOIs have come out as links and others not.

P30L7: Rather than the URLs, we provide the DOIs (when available) that allow link to documentation. The list of references was thoroughly checked.

*Figures and tables:* Some acronyms are in figure and table captions before being introduced in the text. Suggest defining in captions.

*Figure 1: Add section number for Outlook.* Done

Figure 2: Add that L2 SIC is also swath, L3 is a single daily averaged file. Define acronyms used in figure in caption Done

Figure 3: Title should be "AMSR-E" (currently "AMSR"). Labels in the figure need to be closer to the points (or colour coding would help). In figure caption, give section numbers where BFM and OWF are described in the text. "mean water signature" should be "mean open water signature" Done

Figure 4: Left plot: Label "BRI", "BPM" and "BFM" on plot. Add theta label on plot. "u" in caption is labelled "U" on plot, make consistent. Axis labels should also match convention in caption, e.g. "37H" rather than "h37". Right plot: "Freq. Mode" should be "BFM", "Bristol" should be "BRI". Add "theta" symbol to "Rotation angle" axis label. As noted in the text and figure caption, the original figure is from Smith et al. (1996), so that we cannot change the labels on the arrows. The other suggestions are implemented as text in the caption to Figure 4.

Figure 5: Centre panel: Difficult to see any detail using this scale, needs to be shortened. Doesn't have to be the same as left panel as showing different variables Done.

Figure 6: a) Need to show 0% as white (or similar) for SIC plots so can see detail around ice edge. b) Would also be helpful to plot ice\_conc minus raw\_ice\_conc\_values.

- a) We tried your suggestion, but it gives the impression that the SIC fields have missing value (instead of 0% SIC). We did not observe it added much information in the ice edge region. Readers interested in such details would probably open the netCDF files and inspect this more closely, while we aim here at a high-level feel of what is in the variables. We did not change figure 6.
- b) raw ice conc values holds non-masked values iif ice conc = 0% (in places OWF ice conc was triggered) and = 100% the (in places ice conc raw values is larger than 100%). Thus, a plot of "ice conc minus raw ice conc values" would be very similar plot to our of "raw ice conc values". Because it is the first time users are presented with

such "raw" ice concentration values, we feel it is more important to illustrate them what they find in the file. We did not add or change on Figure 6.

**Figure 7: Need to show 0% as white (or similar) so can see detail at low uncertainties.**

There are no grid cells with exactly 0% in sigma\_algo (left) and thus sigma\_tot (right). There are some 0% values in sigma\_smear (center) but as in Figure 6, using white for them gives the impression that the sigma\_smear field has missing values. Readers interested in such details would probably open the netCDF files and inspect this more closely, while we aim here at a high-level feel of what is in the variables. We did not change Figure 6.

*Figure 8: Figure legend - datasets should be capitalised for consistency* Done.

Figure 9: Figure legend - datasets should be capitalised for consistency. If SICCI-25km and OSI-450 lines were thinner (like SICCI-50km) it would be easier to see the lines for both hemispheres.

Capitalization done. We did not change the line width as NH lines were too difficult to read. As per your suggestion, we added some description of the NH and SH curves in the text with discussing Figure 9.

Figures 10, 11: Specify that the sea ice concentration is uncorrected. Numbers in parentheses are in front of the season, not behind. Unclear - "Numbers below the season denote the mean SIC plus/minus one standard deviation" - there's only one number so how can this be plus/minus? Also Figure 11: The SH plots are "bumpier" than the NH plots – add comments on this.

The sea-ice concentration are corrected but not filtered (the OWF and 100% thresholding are not applied). This is now specified in the legend to both Figures. The description of the numbers appearing in the plot area was revised. The SH plots are "bumpier" simply because of the reduced number of data pairs, as indicated in the plot area.

*Figure 12: Standard error is not mentioned in the text.* This is now done.

Table 1: Give months in the time period. Worth adding that grid is EASE grid. Caption: "entering" should be "entered in". Done

*Table 2: Start date for DMSP SSM/I has an error ("090"), check table for other errors* Done, thank you.

**Technical corrections**

General comments: Throughout, need to ensure there is a space between numbers and their units. *Done*.

Throughout have used "..." or "etc...", should probably just be "etc." or sometimes "e.g." but check journal style guide. We will check when editing final version.

Have referred to e.g. F10, F11 satellites, suggest using full name (include DMSP) at the start of the paper for clarification. Done (introduce DMSP acronym early in the text).

Specific comments: Some of the following are corrections of grammatical errors, and some are rewording suggestions to improve the readability of the paper. Thank you very much for compiling all these suggestions!

Page 2 line 4: "allow" should be "allows" line 5: "are" should be "is" line 6: "to understand" should be "for understanding" line 11: "are" should be "is", "have" should be "has" line 28: unclear what you mean by "possibly" in this context, if it's the possibility that filtering can be applied needs rewording.

Implemented all suggestions. We reworded "possibly filtered" to "access to filtered as well a raw values".

Page 3 line 17: remove "up-front" here, reads a bit strangely in this context. Also, "entering" should be "entered" Done.

Page 4 line 3: "some" should be "a" line 4: "more" should be "most" line 6: give section number for Outlook. Line 20/21: "Such wavelength" should be "Such a wavelength" line 23: replace "needed for" with "used in" line 25: add "(Table 2)" after "channels" line 31: "diameters" should be "diameter" Done.

Page 5 line 1: I think "One" should be "Two", also change "swath" to "swaths" line 2: change "orbit" to "orbits", "extent" to "extents" line 5: expand CM-SAF acronym line 7: "directly accessed directly" should be "accessed directly", "Japan space agency" should be "Japan Aerospace Exploration Agency" line 15: "contribution" should be "contributions" line 18: "ERA-Interim" should be "ERA-Interim reanalysis" line 20: "ERA-Interim prior" should be "ERA-Interim data prior" (or similar), "early period with"

should be "earliest period of" line 24: "from" should be "of", "for" should be "in" line 27: "operated to process" should be "for" line 32: "(L3) collects" should be "(L3) chain collects"

Done.

Page 6 line 3: "apply" should be "applies", "format" should be "formats" line 5: "similarly" should be "similar" line 29: define OW (given above in context of algorithm but worth defining here again), same for CI line 30. Done.

Page 7 line 1: "TB in point" should be "TB at point", similarly "lines in point" should be "lines at point" line 2: "and geometric" should be "and a geometric" line 6: remove "originally", "describes" should be "describe" line 20: "onto" should be "on" line 25: "cope for" should be "cope with" Done.

Page 8 line 1: would read better as "Figure 4 (right panel) also shows that the optimum..." line 16: "space" should be "spaces" lines 19/20: replace arrows with " " line 26: "section so" should be "section has so" line 32: "by Eq. 1" should be "using Eq. 1"

Done.

Page 9 line 20: "was" should be "has been", comma before "which" line 21: "varies" should be "vary", "follows" should be "follow" line 25: "yield highest" should be "yield the highest" line 26: "yield departure" should be "yield a departure" line 27 and 28: "departure" should be "departures" Done.

Page 10 line 3: add commas both before and after "the uncorrected SIC value" line 7: "re-analysis" should be "re-analyses" line 20: Add "For Tb\_nwp" at the start of the line line 22: Add "For Tb\_ref" at start of sentence before "Theta\_instru" line 26: remove "for F10" (already mentioned in this sentence) line 29: "for being" should be "to be" Done.

Page 11 line 3: "allows" should be "allowed" line 9: "ones" would read better as "datasets". Also, having introduced the acronym WFs should use on this line instead of "Weather filters" (also on line 20). Line 14: no hyphen in unaffected line 18: suggest changing "so far did not adopt" to "have so far not adopted", also "from" should be "in" line 19: change "by using adhoc status flags" to "on an adhoc basis by using status flags" (as the flags themselves are not adhoc) line 22: "re-used" should be "reuse" (or "have reused") line 23: Suggest add "For example," before Spreen et

al. Line 25: "to" should be "with" line 26: "with" should be "for", "for which" should be "where", suggest changing "threshold is 0.053" to "threshold is set to 0.053" Done.

Page 12 line 2: "intersect" should be "intersects" line 6: missing close bracket after (AD), also "illustration how" should be "illustration of how" line 7: "into" should be "in" line 11: add "and" before "the varying effects" line 12: suggest replace "not remove" with "avoid removing" line 13: "show" should be "shown" lines 16,17: "T" should be in italics line 20: "naming" should be "name" line 22: change "is set to" to "will" as this is an unintended consequence line 23: suggest changing "we rather refer to such filters as 'Open Water Filter"" to "we refer to such a filter as an 'Open Water Filter", also suggest "add" changed to "include" line 24: "are" should be "is" line 27: "Noticeably" should be "Notably" line 28: "attached a" should be "attached to a", also change "as to if the OWF detected it" to "corresponding to OWF detection" Done.

Page 13 lines 1&2: "high concentration range" should be either "a high concentration range" or "high concentration ranges line 5: remove "likewise" line 10: change "best appear" to "are best shown" line 11: "T" in B\_CI(T) should be bold lines 14&15: "constantly" should be "consistently" line 28: "Laptev and Kara Sea" should be "the Laptev and Kara Seas" line 29: would read better to remove "old" after "2 years", also "on right panel" should be "on the right panel"

Page 14 line 6: "north for Canadian" should be "north of Canadian" line 8: remove "that what" line 9: change "and" to "which" after "section 3.4.1," line 19: "data is assimilated" should be "data are assimilated" line 21: "those" should be "that" line 28: "algorithm to retrieve" should be "algorithm for retrieving" Done.

Page 15 line 1: suggest change "relevant to discuss" to "relevant for discussion of" line 4: "Earth surface" should be "the Earth's surface" line 7: remove "that is" line 9: "cells" should be "cell" line 20: remove "shortly" line 21: suggest change "presenting less" to "have undergone little" line 28: "details" should be "detail" line 30: suggest change "among others" to "including" line 31: remove "is computed" line 32: change "the antenna pattern functions are approximated" to "the approximation of antenna pattern functions" line 33: "from central" should be "from the central" Done.

Page 16 line 1: "for contribution" should be "for the contribution" line 4: "were" should be "have been" line 5: "where" should be "were" line 7: "as input" should be "as the input" line 13: "were" should be "was", suggest change "base" to "basis", "pixel"

should be "pixels" line 18: "in SH" should be "in the SH" line 19&20: change "where to select the Open Water training samples" to "where the Open Water training samples were selected" line 26: "conversely" should be "converse", suggest "CDR of" should be "CDR method of" Done.

Page 17 line 14: "of SICCI-25km" should be "of the SICCI-25km" line 22: "file" should be "files" line 27: "Bottom" should be "The bottom" line 29: "corresponds" should be "correspond" line 32: "by OWF" should be "by the OWF" Done.

Page 18 line 6: "indicate" should be "indicates" line 14: replace "are covered by" with "cover" line 18: Suggest replace "several" with "three", "One" with "The first" line 19: "its" should be "their" line 25: no hyphen in intermediate Done.

Page 19 line 9: "albeit" should be "despite" line 12: "from" should be "for" line 15: "from" should be "for" line 16: "improve much" should be "much improve" line 18: "parametrization" should be "parametrizations" line 21: "from with" should be "for", also "were" should be "where" line 25: "sensibly" - do you mean "ostensibly"? Done.

Page 20 line 9: remove "at best" and add "ideally" before "preserving" line 19: reword "very little few jumps are" to "very little change is" (or similar) line 23: could remove "lowest" and "highest" as it's already clear this is the range Done.

Page 21 line 10: remove comma after "but" line 14: "details" should be "detail" line 21: "East Antarctic" should be "the East Antarctic" (or "East Antarctica") line 23: remove "being", suggest replace "by too" with "with" lines 29&30: suggest move "than for the other two CDRs" after "more" on line 29 line 30: "e.g." should be "i.e." line 31: add "for all three CDRs" after "2%". lines 32&33: suggest change "less good than that" to "poorer than"

Page 22 lines 5&6: change "Arctic" to "the Arctic" Done.

Page 23 line 3: Suggest reword "can be picked" to "may be selected" line 4: Suggest reword "and to the least at the location of the ground-truth estimates used in the section" to "where the ground-truth estimates used in the section are located" line 5:

Change "More developed" to "A more developed", "as wetter" to "as a wetter" line 6: Change "We finally" to "Finally, we" line 12: "in large extent" needs to be reworded, perhaps replace with "generally" or "to a large extent" line 13: Capitalisation of "Passive Microwave" varies, be consistent line 15: "on combination" should be "on a combination", also need to define acronym "PMR" line 19: "take" should be "pay" line 20: "in field" should be "in the field", also use "OWF" acronym for consistency line 21: "are pertaining" should be "pertain" line 25: Remove "distinguishing between" and add "to be distinguished" to end of sentence. Done.

Page 24 line 1: "aims" should be "aim", also "from interested" should be "from the interested" line 6: replace "was" with "were" twice line 17: "is 'spilling' " should be " 'spills' ", also "appear" would be better than "look" line 19: "foot-print" sometimes has a hyphen, sometimes not, needs to be consistent line 20: "instrument" should be "instruments" line 29: "improvement" should be "improvements" Done.

Page 25: line 7: "on March 1985" should be "in March 1985" line 8: add "dataset" before "only", also add "on" before "09 July" lines 9&10: change "achieving" to "to achieve" line 11: suggest reword "algorithms allowed e.g. to consistently process SIC" to "algorithms also allowed consistent processing of SIC" line 14: change "15 years record" to "15-year record" line 18: change "will" to "would" line 20: change "had met" to "would meet" line 33: add "the" before "coarsest" Done.

Page 26 lines 3&4: Confusingly worded: "seasonal cycle of sea-ice and snow properties during summer". Should this be sea ice extent? (Also be consistent throughout about whether to use a hyphen in sea ice or not) line 15: "than" should be "as" line 17: Suggest remove "For all practical purposes" line 29: Remove "namely" line 33: remove "that" Done.

Page 27 line 1: "impact of melting" should be "impact that melting" line 3: Suggest change "more efforts" to "further effort" line 4: Remove "same" and "that was" line 10: "if to selecting" should be "if selecting" line 12: "dimension" should be "dimensions" line 17: "the sea ice cover, sea ice area" should be "of sea ice cover, and sea ice area" line 27: "exploring" should be "exploration" line 28: "channels" should be "channel", "that" should be "than" line 30: Suggest change "could not be better embedded by SIC" to "could be better embedded in SIC" line 32: "Filter" should be "Filters"

Done.

Page 28: line 2: Suggest adding "still" after "can" line 24: Change "at best" to "to achieve" line 33: "passed" should be "past" Done.

Page 29 line 5: Add "However," before "Because" (as shouldn't start a sentence with because) line 17: "product contains" should be "products contain" as there is more than one product. Line 19: Replace "on the hand a" with "ease of", "product" should be "products" line 20: remove "of all products", "is" should be "are", "has" should be "have" line 25: "this provides" should be "this paper provides" (or similar) Done.

Page 30 line 6: "making" should be "make" Done.

==============

**Version 2 of the EUMETSAT OSI SAF and ESA CCI Sea Ice Concentration Climate Data Records**

Thomas Lavergne1, Atle Macdonald Sørensen1, Stefan Kern2, Rasmus Tonboe3, Dirk Notz4, Signe Aaboe1, Louisa Bell4, Gorm Dybkjær3, Steinar Eastwood1, Carolina Gabarro5, Georg Heygster6, Mari

5 Anne Killie1, Matilde Brandt Kreiner3, John Lavelle3, Roberto Saldo7, Stein Sandven8, Leif Toudal Pedersen7

1Research and Development Department, Norwegian Meteorological Institute, Oslo, Norway
2Integrated Climate Data Center, CEN, University of Hamburg, Hamburg, Germany
3Danish Meteorological Institute, Copenhagen, Denmark

4Max-Planck Institut für Meteorologie, Hamburg, Germany 5Barcelona Expert Center, ICM-CSIC, Spain 6Institute of Environmental Physics, University of Bremen, Bremen, Germany 7Danish Technical University-Space, Copenhagen, Denmark 8Nansen Environmental and Remote Sensing Center, Bergen, Norway

15

Correspondence to: Thomas Lavergne (thomas.lavergne@met.no)

[revised manuscript text omitted]

- 5 needed to acquire a single pixel. The effective Field-of-View (eFoV) diameters includes the two effects, and is a better measure of the true footprint of the instrument. For example, the eFoV of the SSM/I 19\_GHz channels is closer to 70x75\_km. The dimensions of the iFoV and eFoV are referred to as the resolution of the channels. The sampling is how close in space the FoVs are acquired. Most channels are thus oversampled.
- 10 One Two of the differences between the instrument series are the width of their observation swaths, and the inclination of their orbits. This translates into different extents of the polar observation hole, and no data are available for sea-ice monitoring north of 84° (SMMR), 87° (SSM/I), 89° (SSMIS) and 89.5° (AMSR-E and AMSR2).

For our data records, a newly reprocessed version of the SMMR, SSM/I, and SSMIS data into a Fundamental Climate Data

- 15 Record (FCDR, L1) was accessed from the EUMETSAT Climate Monitoring Satellite Application Facility (CM-SAF-( Fennig et al. 2017). In the FCDR, the TB are re-computed from Antenna Temperatures (TA)-, screened and corrected for known artefacts like solar intrusion, and intercalibrated between missions. The AMSR-E data we use is the NSIDC FCDR AE\_L2A V003 FCDR of Ashcroft and Wentz (2013), covering the full lifetime of the mission from 1st June 2002 to 10th October 2010. For AMSR2, we use re-calibrated (Version 2) L1R data that we directly-accessed directly from the Japanese Japan Aerospace space-Exploration agency-Agency (JAXA), covering 23rd July 2012 until 15th May 2017, that is the end of the SICCI-25km and SICCI-50km CDRs. For both AMSR-E and AMSR2, the TB are available both at their nominal resolution (documented in Table 2), and post-processed at lower resolution matching those of other channels (e.g. the 36.5 GHz TB at the resolution of the 6.9 GHz channel). We use the nominal resolution of the TBehannels is used, not the resolution-matched ones. It is noteworthy that the AMSR2 data is not from an FCDR, but rather from an archive of an
- 25 operational data stream. We use the data as they are provided by JAXA, without applying extra calibration towards AMSR-E (thus unlike Meier and Ivanoff, 2017) since our algorithms do not require such stringent calibration thanks to using dynamic tuning (section\_-3.3).

**2.2 ERA-Interim data**

The microwave radiation emitted by the ocean and sea ice travels through the Earth atmosphere before being recorded by the satellite sensors. Scattering, reflection, and emission in the atmosphere add or subtract contributions to the radiated signal, and challenge our ability to accurately quantify sea-ice concentration. An initial step in our processing is thus the explicit correction of atmospheric contribution to the top-of-atmosphere radiation (see section\_-3.4.1). For this purpose, we accessed the global 3-hourly fields from ECMWF's ERA-Interim reanalysis (Dee et al., 2011). Fields of 10m wind-speed, 2m air

| -  | Formatted: Subscript   |  |
|----|------------------------|--|
| 1  | Formatted: Subscript   |  |
| -1 | Formatted: Superscript |  |
| 4  | Formatted: Superscript |  |

temperature, and total column water vapour are used. The ERA-Interim re-analysis starts in January 1979 and is available throughout the time period of our CDRs. Unavailability of ERA-Interim data prior to 1979 made it impractical to use the early earliest period with of SMMR data (October to December 1978).

**3 Algorithms and processing details**

[revised manuscript text omitted]

Figure\_-3 illustrates the functioning of the BFM algorithm. Working with SMMR data for sea ice monitoring, Comiso (1986) recognized that the typical signature of Open Water (OW, SIC=0%, grey triangles) TB data clusters around an averaged point location (the OW tie-point, *H*) in the (19V, 37V) TB space. Conversely, the Closed Ice (CI, SIC=100%, grey discs) TB data mostly clusters along a line (the consolidated ice line A-D). Comiso (1986) thus designed a SIC algorithm where isolines of constant SIC are parallel to the A-D line and pass through the measured TB in-at point P. A geometric algorithm using the intersection of the (H,P) and (A,D) lines in-at point I returns the SIC value (in our example SIC=68%). In the same study, similar aggregation of typical TB signatures and a geometric algorithm were also used in the (37V, 37H) TB space (BPM algorithm). For easing later discussion, we note here that in winter Arctic conditions, the typical multiyear

25 sea-ice signature is to the "left" of the ice line -close to AD- while first-year sea ice and young sea ice is to the right -closer to DA- (Comiso et al. 2012). The AMSR-E TB samples on Figure 3 are from Pedersen et al. (2018), the ESA CCI Sea Ice Round Robin Data Package (RRDP).

[revised manuscript text omitted]

- are only used for algorithm tuning if their latitude is less than 84°N, which is the limit of the SMMR polar observation hole (Table 2).
- 25 The selection of the OW tie-point samples was-has been revised since Tonboe et al. (2016), which used fixed ocean areas at midhigh latitudes. The training areas now varies vary on a monthly basis, and follows sea-ice cover more closely. In practice, the OW locations are those falling in a 150 km wide belt just outside the monthly varying maximum ice extent climatology (which is itself described in section\_-3.6).

**3.4 Strategies to further reduce systematic errors and random noise**

30 The algorithms described in section 3.2-3.2-are self-optimizing to yield the highest accuracy at high and low concentration ranges. Nevertheless, all TB triplets with a departure from the mean CI or OW signatures will yield a departure from 0% and 100% sea-ice concentration. Random departures that do not have apparent spatial or temporal structures are often referred to as *random noise*, while departure that are somewhat stable (correlated) in space and time are referred to as *systematic errors*.

Analysis of time-series of sea-ice concentration maps retrieved from the algorithm from section 3.2-3.2 reveal that the departure at low concentration range (open water) is typically a random noise, while more systematic errors are observed at high concentration range (closed ice). This is explained by the different nature of the error sources playing a role at these two ends of the sea-ice concentration range: weather-related effects at synoptic scales over open water, and surface emissivity variability (due to ice type, temperature of the emission layer, snow depth, etc...) over closed ice. In this section, we describe strategies implemented in the processing chain to further reduce random noise over open water, and systematic errors over closed ice. Both correction steps are applied during the second iteration of the L2 chain (Figure\_-2) and we note SICucorr (uncorrected), the uncorrected SIC value, before the start of the second iteration.

**3.4.1 Radiative Transfer Modelling for correcting atmosphere influence on brightness temperatures**

- 10 As described in Andersen et al. (2006) and confirmed in Ivanova et al. (2015), the accuracy of retrieved sea-ice concentration can be greatly improved when the brightness temperatures are corrected for atmospheric contribution by using a Radiative Transfer Model (RTM) combined with surface and atmosphere fields from NWP re-analysisanalyses. The correction using NWP data is only possible in combination with a dynamical tuning of the tie-points, so that trends from the NWP model are not introduced into the sea-ice concentration dataset. The correction scheme implemented in the new CDRs
- 15 is based on a "double-difference" scheme, similar (but not identical) to that described in Andersen et al. (2006) or Tonboe et al. (2016).

The scheme evaluates the correction offsets δTB (one per channel), the difference between two runs of the RTM: TBnwp uses estimates from NWP fields (in our case ERA-Interim), while TBref uses a reference atmospheric state with the same air temperature as TBnwp, but zero wind, zero water vapour, and zero cloud liquid water. δTB is thus an estimate of the atmospheric contribution at the time and location of the observation.

 $Tb_{nwp} = F(W_{nwp}, V_{nwp}, L_{nwp} = 0; T_s, SIC_{ucorr}, \theta_0)$  $Tb_{ref} = F(0,0,0; T_s, SIC_{ucorr}, \theta_{instr})$  $25 \quad \delta Tb = Tb_{nwp} - Tb_{ref}$  $Tb_{corr} = Tb - \delta Tb,$

30

(2)

For  $T_{\text{Brurp}}$ . The the RTM function F simulates the brightness temperature emitted at view angle  $\theta_0$  by a partially ice-covered scene with sea-ice concentration SIC, and with surface and atmospheric states described by  $W_{nwp}$  (10 m wind-speed, m.s-1),  $V_{nwp}$  (total columnar water vapour, mm),  $L_{nwp}$  (total columnar liquid water content, mm), and Ts (2 m air temperature). For TBrer  $\theta_{instr}$  is the nominal incidence angle of the instrument series (see Table\_-2). Our double-difference scheme is thus both a correction for the atmosphere influence on the TB (as predicted by the NWP fields) and a correction to a nominal incidence angle. The latter is required for stabilizing the DMSP SSM/I F10 signal, whose view angle varied significantly: the peak-to-

11

peak daily average incidence angle variation due to the platform's orbital drift was  $52.6^{\circ}-53.7^{\circ}$  for F10 according to Colton and Poe (1999). The typical values of  $\delta T_B$  range from about 10 K over open water to few tenths of a Kelvin over consolidated seaice. The liquid water content (L) fields from global NWP fields (and ERA-Interim in particular) were found to not be accurate enough for beingto be used in our atmospheric correction scheme (Lu et al. 2018). The TB are thus not corrected for L (L=0 in both TBnwp and TBref), and the induced remaining noise transfers into uncertainty in SIC.

We use the Remote Sensing Systems (RSS) RTM, whose tuning to different instruments is documented in Wentz (1983) for SMMR, Wentz (1997) for SSM/I and SSMIS, and Wentz and Meissner (2000) for AMSR-E and AMSR2. It is a parametrized, fast RTM optimized for the frequencies and view angles covered by the passive microwave sensors at hand. It originally allows-allowed ocean and atmosphere simulations, and was later extended to cover sea ice surface conditions (Andersen et al, 2006). Since the RTM is used in the double-difference scheme described above, accurate calibration of the RTM simulation with the measured brightness temperatures is not critical since such offsets cancel out. The atmospheric correction step has more impact over open water and low concentration values than over closed ice conditions. This is because 1) a generally dryer atmosphere above the consolidated ice pack, 2) of the effect of wind speed on ocean (and not sea-ice) emissivity, and 3) of the low emissivity and high reflectivity of water at the frequencies we use in SIC algorithms

15 sea-ice) emissivity, and 3) of the low emissivity and high reflectivity of water at the frequencies we use in SIC algorithm (Andersen et al. 2006).

**3.4.2 Open Water Filtering**

5

30

The Weather Filters (WFs) of Cavalieri et al. (1992) have been used in basically all available SIC CDRs except the earlier EUMETSAT OSI SAF ones-datasets (Andersen et al, 2007, Tonboe et al. 2016). Weather filterWFs are algorithms that combine TB channels to detect when rather large SIC values (sometimes up to 50% SIC) are in fact noise due to atmospheric influence (mainly wind, water vapour, cloud liquid water effects), and should be reported as open water (SIC=0%). The concept of WFs is very different from the atmospheric correction of TB described in the previous section: the atmospheric correction reduces noise in the resulting SIC fields (but does not yield exactly SIC=0% over open water) while the WF is a binary test to decide if a pixel should be set to exactly SIC=0% or left un-affected. In the new CDRs, we combine both approaches as we apply the WFs after the atmospheric correction.

While WFs are effective at removing false sea ice in open water regions, they will always falsely remove (detect as open water) some amount of low concentration (and/or thin) sea ice, especially along the ice edge (Ivanova et al. 2015). This is why the OSI SAF SIC CDRs has so far did not adopted WFs and why the effect of WFs can be fully reverted from-in our new SIC CDRs by using on an ad--hoc basis by status flags in the product files (see section -4.1).

[revised manuscript text omitted]

5 conditions (OW samples below J). In addition, GR3719v contains information on sea-ice type (Cavalieri et al. 1984) and it is desirable the filter should work equally for first-year and multiyear sea ice. For this these reasons, we rather refer to such a filters as an "Open Water Filter" (OWF) and add-include a test on the SIC value. The OWF implemented in the new CDRs are is thus defined by the following two tests (corresponding to the thick solid blue line in Figure 3):

 $\begin{cases} \text{GR3719v} \ge T\\ \text{or SIC} \le 10\% \end{cases},$

20

(3)

10 NoticeablyNotably, we compute OWFs in swath projection, in the Level-2 chain (Figure 2). As a result, each FoV observation at Level-2 is attached to a binary flag as to if the corresponding to OWF detected detectionit as "probably" open water or not. This binary flag is combined during gridding and daily averaging to yield Level-3 fields of OWFs. This is a better approach than computing WFs from daily averaged gridded TB data which will smooth and smear the sea-ice edge region, as well as rapidly changing weather effects such as cloud liquid water content or wind roughening. To compute 15 OWFs at Level 2 can also help mitigate the potential impacts of changes in satellite crossing time between different missions. The impact of the dynamic tuning of the OWF is evaluated in section\_4.2.1.

**3.4.3 Reducing systematic errors at high concentration range**

[revised manuscript text omitted]

Figure 5 (center) shows the result of the ice curve correction on SIC variability averaged for January 2015. It plots the difference between the variability (standard deviation) of the un-corrected SIC values (SICureen) and that of the SIC after correction. Black solid lines show the mean sea ice edge region (at 15% and 70% SIC values). In the regions covered with sea-ice (>= 70% SIC), the shades of light blue indicate that the variability at high concentration is rather consistently reduced by about 1-2% SIC by the ice curve correction...: the SIC after correction is a more accurate description of a nearly 100% ice cover. A limited number of regions show no improvement (white color) or slight degradation. This reduction of the variability comes in addition to the correction for the systematic errors (e.g. underestimation north for-of Canadian Arctic Archipelago, see panel a)left panel for which the ice curve correction was designed. The analysis of the closed-ice (>=70% SIC) region in Figure 5, thus confirms that the ice curve correction works as expected at high concentration range, and is potentially linked to the age of sea ice.

- 15 In the open water regions of Figure 5 (<= (outside the 15% SIC contour), the reduction of variability (center panel) is even larger (3-4% SIC) than over closed-ice regions. This reduction is the result of the atmospheric correction step, that what described in section 3.4.1s and has most impact over open water (Andersen et al. 2006). From left panel, it appears that the atmospheric correction step in average increases SIC (shades of red) over open water regions close to the sea-ice edge, e.g. in East-Greenland Sea, Barents Sea, and in Labrador Sea. These regions generally present negative SICs before correction, and are brought closer to 0% SIC by the process of atmospheric correction. This is due to selecting the training OW samples in
- lower latitude conditions (ocean surface, atmosphere conditions) than prevailing closer to the ice edge, and is also discussed in section 4.2.3 when evaluating uncertainties.

Still on Figure 5 (center panel), The the increased variability (red tones) between the 15% and 70% isolines follows logically
 from the two above mentioned reductions: the corrections enable more accurate retrievals of SICs, thus the ice edge is more sharply defined in the daily SIC fields, and this results in higher variability on a monthly basis.

In this section, we described the strategies we implemented to improve the accuracy of the SIC algorithms. In the next section, we discuss how the remaining noise is quantified and reported to the users of the data records in the form of uncertainties.

**3.5 Uncertainties**

30

Spatially and temporally varying uncertainty estimates for each and every SIC value are required of state-of-the-art CDRs (GCOS-IP, 2016). Uncertainties are needed as soon as the data are compared to other sources (e.g. other similar data

records), or when data is are assimilated into numerical models. However, there is no unique way to derive nor to present uncertainties in EO data (Merchant et al., 2017).

The approach to derive and present uncertainties in the new SIC CDRs is mostly similar to those that of Tonboe et al. (2016): we make the assumption that the total uncertainty  $\sigma_{tot}$  is given by two uncertainty components, i.e.:

 $\sigma_{\text{tot}}^2 = \sigma_{\text{algo}}^2 + \sigma_{\text{smear}}^2$

5

10

15

(4)

where  $\sigma_{algo}$  is the inherent uncertainty of the SIC algorithm (algorithm uncertainty) including sensor noise and the residual geophysical noise quantified as variability around the OW and CI mean signatures, and  $\sigma_{smear}$  is the representativeness uncertainty due to resampling from satellite swath to a grid (smearing uncertainty) and the mismatch between footprints at different channels-mismatch.

The derivation of  $\sigma_{algo}$  is to a large extent similar to that described in Tonboe et al. (2016). This term is derived from the accuracy (estimated as statistical variance) of the algorithm to for retrieve retrieving 0% (*resp* 100%) when applied onto the OW (*resp* CI) training data samples (section\_-3.3). This uncertainty term is computed at Level 2 (Figure 2). Each Level 2 SIC estimate in the data record has an associated  $\sigma_{algo}$  value.

The uncertainty term  $\sigma_{smear}$  is a representativeness uncertainty. It measures the increase of uncertainty due to mismatching spatial dimensions such as when a) the satellite sensor footprint potentially covers a larger area than that of a target grid cell, or when b) the imaging channels used by the SIC algorithms do not have the same FoV diameter. Table 2 lists the dimensions relevant to-for discussion of these two effects. Effect a) : the size of the 3-dB-footprint of the 19-GHz channels 20 of the SMMR, SSM/I, and SSMIS instruments is larger than the resolution of the grid used to present the SIC field (25x25\_km, see Table 1). Effect b) : the 3 dB footprint of the 37\_GHz channels is smaller than that of the 19\_GHz ones, so that the two frequencies entering the SIC algorithms do not cover the same area of Earth's surface. Intuitively, both effects should have no or limited impact where the sea ice cover is homogeneous (fully consolidated sea ice, or open water). It should be at a maximum where sharp spatial gradients occur, typically at the sea-ice edge. The smearing contribution  $\sigma_{smear}$  is 25 difficult to derive analytically and we carry on the approach of Tonboe et al. (2016) that is to parametrize  $\sigma_{smear}$  as a function of a proxy. For the three new CDRs we parametrize  $\sigma_{smear}$  as a function of the (MAX-MIN)3x3 value, that is the difference between the highest and lowest SIC value in a 3x3 grid cells neighbourhood around each location in the grid. Specifically:  $\sigma_{\rm smear} = K \times (MAX - MIN)_{3 \times 3},$ (5)

30 where *K* is a scalar whose value depends on the FoV diameter of the instrument channels used for the SIC computation, and the spatial spacing of the target grid. Several other proxies for the local variability of the SIC field (among others the 3x3 standard deviation, the Laplacian, power-to-mean-ratio...) were tested and this one was selected for its simplicity and robustness. Values of K were tuned using a foot-print simulator and selected cloud-free scenes of the marginal ice zone imaged by the Moderate-Resolution Imaging Spectroradiometer (MODIS) as described in Tonboe et al. (2016). The MODIS images are first classified as water/ice at full resolution. Two sets of coarser resolution SIC fields are then prepared: 1) the footprint simulator is applied to prepare a synthetic sea-ice concentration field at the resolution of the frequency channels, and 2) the high-resolution classified pixels are binned into regular grid cells, e.g. at the target resolution of the CDR (e.g. 25x25 km). The mismatch between the two fields is what we call the smearing uncertainty, and is parametrized against

[revised manuscript text omitted]

**20 4.2 Initial eEvaluation results**

5

25

30

The evaluation of a CDR needs to cover several-three aspects. One-The first is to demonstrate consistency of the methods used to derive the CDR. Key elements of our new suite of algorithms are i)-its-their application to different sensors (various SSM/I, AMSR-E and AMSR2), ii) a self-optimizing algorithm which dynamically tunes tie points to minimize SIC errors at 0% and 100%, and iii) a dynamic open-water filtering (OWF) to mitigate spurious SIC values caused by residual weather influences while keeping actual low SIC. For the three SIC CDRs published here we investigate time-series plots of the

optimized skills of the SIC algorithms, and the temporal stability of the OWF (section\_-4.2.1).

**4.2.1 Monitoring stability and internal consistency**

Many time-series plots can be produced to illustrate the stability and internal consistency of the three CDRs. As an example, Figure\_8 shows the time-series of the algorithm training statistics at the Open Water target. As described in sections 3.2\_<del>3.2</del> and -3.3, the algorithms implemented in the three CDRs dynamically tune their parameters to yield zero bias and minimum standard deviation of the computed SICs (*aka* best accuracy) over the Open Water (OW) and Closed Ice (CI) training targets. Figure –8 shows the Northern Hemisphere (NH, top) and Southern Hemisphere (SH, bottom) temporal evolution of the

standard deviation (solid lines) and bias (dotted lines) of the SIC algorithms over OW target areas. Prior to further describing Figure\_-8, it is important to note that the biases and standard deviations discussed here are internal to the processing chains, not an evaluation of the CDRs against independent observations of SICs. An initial-evaluation of the CDRs against independent ground-truth observations is the topic of section\_-4.2.2.

5

From Figure\_8, it is easy to see that the algorithms implemented in the three CDRs achieve zero bias (dotted lines along the y=0 axis) for all instruments and both hemispheres, on a daily basis. To achieve zero bias albeit\_despite\_the changes in central wavelengths and calibrations from one satellite to the next is one of the key advantages of using dynamically-tuned algorithms (section\_3.3).

10

The impact of the explicit correction of brightness temperature from-for atmospheric noise effects is also clearly visible on Figure\_-8, since the standard deviations resulting from un-corrected  $T_B$  data (thin solid lines) are consistently above those from-for corrected data (thick solid lines) by about 3% to 4% on average, depending on the season and hemisphere. The seasonal variability is also larger from the un-corrected data, especially in the NH. It is noteworthy that the atmospheric

- 15 noise reduction step does not much improve much the OW standard deviation in the SH at the beginning of the OSI-450 period, for the SMMR instrument (1979-1987). As noted at the end of section\_-3.4.1, OSI-450 uses the Wentz (1983) RTM for SMMR, and the Wentz (1997) RTM for SSM/I and SSMIS. The parametrizations implemented in the SMMR RTM are probably less developed than in the SSM/I and SSMIS RTM, which might explain why the impact on our standard deviation is more limited for SMMR. Another plausible explanation is that the re-analysed fields for wind speed and water vapour from ERA\_
- 20 Interim are less accurate in the SMMR era than in the SSM/I and SSMIS era. We note that clear-sky radiances from SSM/I and SSMIS were directly assimilated in ERA-Interim over the ocean (Dee et al. 2011), but not SMMR radiances (Paul Poli, personal communication, 28/09/2018). This can especially have an impact in the SH, where other sources of conventional dwdwccccAutpathaevollbdumlyclifntRAtindscattileSMRthafmethSSMpail/thElsebaceCinebalbaebaecarEnfordegilipticipat on the accuracy of OW SICs during the SMMR era.
- 25

The SICCI-25km and SICCI-50km standard deviations are also plotted on Figure\_8 (only those after atmospheric correction so as not to clutter the plot area). SICCI-25km (reds) achieves sensibly roughly the same OW standard deviation as OSI-450. Since SICCI-25km uses very similar frequency channels to those of OSI-450 (Table 1), it is not surprising they achieve similar accuracy. The central frequency of the AMSR-E and AMSR2 channels (18.7\_-GHz) is slightly further away from the

30 water vapour absorption line (~22 GHz) than the SSM/I and SSMIS channels (19.3\_-GHz). This difference in frequency yields better accuracy for SICCI-25km than OSI-450 when using un-corrected TB data (not shown) but this effect is mostly cancelled after atmospheric correction (though not fully in SH, bottom panel). The same effect is observed for the standard deviations resulting from un-corrected SMMR TB data (purple thin line), which is consistent with a central frequency of 18.0 GHz (Table 2).

SICCI-50km (greens) is more accurate than both SICCI-25km and OSI-450, by nearly 1% in NH, and 0.5% in SH. This is expected from the choice of frequency channels, since SICCI-50km uses a C-band (6.9 GHz) channel, while SICCI-25km and OSI-450 use Ku-band (~19 GHz). Three effects lead to better accuracies of SIC retrievals at low-frequencies: 1) the atmosphere is more transparent, yielding better accuracy over OW, 2) the noise sources such as sea-ice type, snow depth, snow scattering, etc... have less impact at low frequencies, and 3) the permittivity (and hence TB) of sea ice and water are more different, resulting in a larger dynamic range for sea-ice concentration retrievals. SICCI-50km is designed to be the most accurate of the three SIC CDRs. However, it achieves a coarser spatial resolution (50\_-km) due to the limited size of the AMSR-E and AMSR2 antenna. The time-series in Figure 8 illustrate that the algorithms are internally consistent, behave as
expected across instruments, and are effectively tuned to achieve zero bias and a smallsmallest possible retrieval noise for each instrument in the time series.

The role of Open Water Filters (OWF) is to detect and remove weather-induced false sea ice over open water, while ideally preserving the true low concentration values (typically at the ice edge)-at best. As introduced in section\_3.4.2, the threshold 15 of the OWF is tuned dynamically against the daily updated training data samples (thus by instrument, and by hemisphere) to preserve true SIC values down to 10%. A water/ice separation limit at 10% SIC is an ambitious goal, but is necessary to ensure that time-series of Sea Ice Extent (SIE, usually defined with a threshold of 15% SIC) are not influenced by the OWF and only by the evolution of true SIC. Figure\_9 shows time-series of NH (solid lines) and SH (dashed lines, almost coinciding with NH lines) of the 1%-percentile value of all ice conc values (thus after all filters including the OWF is 20 applied) that are strictly positive and below 30% SIC for the OSI-450 (blue), SICCI-25km (red), and SICCI-50km (green) CDRs. These are thus time-series of the typical minimum detected SIC that are preserved by the OWFs. A solid horizontal line is drawn at 15% SIC value, the threshold commonly chosen for SIE computations. The OSI-450 curves are very stable with time and increase only slightly from around 9% SIC at the beginning of the period to around 10.5% SIC at the end. Seasonal variations are visible especially at the beginning of the time-series for NH cases, when typical winter values are 25 around 7.5% SIC and peak to 10% SIC in summer. They are in any case well below the 15% threshold throughout the data record and very little few jumpschange are-is observed when transitioning between sensors. The seasonal cycles are limited to few tens of a percent at the end of the period (few percent at the beginning). The SICCI-25km curves are close to the OSI-450 ones, but at a slightly larger value of 11%, with a seasonal variation range of about 2%. The SICCI-25km curves are also well below 15%. The SICCI-50km curves are those showing the largest variation. The average value for SICCI-50km is at 30 about 10%, but the seasonal variations are much larger, ranging from lowest 5% to highest 15%. The temporal stability of the time-series on Figure -9 document that the tuning of the OWFs at values close to 10% SIC is successful for the two data records that rely on the 19\_-GHz and 37\_-GHz for computing their SICs (OSI-450 and SICCI-25km) and not as good for SICCI-50km that uses the 6-GHz and 37-GHz channels to compute the SIC values. Although SICCI-50km does not compute SICs from 19\_-GHz and 37\_-GHz channels, its OWF is still based on the GR3719v threshold (section 3.4.2). The mismatch in frequency and resolution between the channels used to compute the OWF and those used to compute SIC explains the larger variability of the SICCI-50km time-series in Figure\_-9.

We note in addition that both the OSI-450 and SICCI-25km CDRs dynamically tune their optimal data plane for low
concentration range θOW in the (19V, 37V, 37H) 3D TB space, while the OWF is only tuned in the (19V, 37V) TB plane. The departure of the optimal SIC data-plane from the (19V, 37V) OWF plane (by convention at θ=0°, see right-hand side panel in Figure\_4) could be the causes for 
[revised manuscript text omitted]

Ice resulting from freezing of fresh and brackish waters does not have the same emissivity as that from sea water. The retrieval of ice area fraction in these conditions would call for dedicated tie-points (e.g. Ghaffari et al. 2011), which we did not implement here. In addition to the difficulty of computing dynamic tie-points over such small areas, it is unclear if such dedicated tie-points would make a large difference in the end, because of the combination of many error sources in these

25

30 <u>close water bodies (land spill-over, thin sea-ice, larger atmospheric influence, etc...). A layer in the status\_flag variable indicates fresh and brackish water bodies.

**5 Discussion, Outlook and Conclusions**

**5.1 Discussion**

5

This paper documents three new Sea Ice Concentration (SIC) Climate Data Records (CDR). One from EUMETSAT OSI SAF (OSI-450), and two from ESA CCI (SICCI-25km and SICCI-50km). All three share the same algorithm baseline, which is both a continuation of the EUMETSAT OSI SAF SIC approach (Andersen et al. 2006, Tonboe et al. 2016) and a series of

- innovations contributed mostly by the ESA CCI activities. The three CDRs are a family of data records that aim at addressing the GCOS Requirements for the Sea Ice Essential Climate Variable (ECV) (GCOS-IP, 2016). The improvements with respect to earlier versions of the CDRs include 1) using high-quality Fundamental Climate Data Records (FCDR) as input data (section\_-2.1), 2) a new family of self-tuning, self-optimizing SIC algorithms that dynamically adjust to the input TB data (section\_-3.2, and\_-3.3), 3) novel noise reduction and filtering approaches (section\_-3.4), and per-pixel uncertainty estimates (section\_-3.5). The product data files are designed so that interested users can revert some of the filtering steps and access the "raw" output of the SIC algorithms (section\_-4.1).
- The three CDRs are designed to ensure temporal continuity throughout the almost 40\_-years of passive microwave data records. The OSI-450 dataset currently covers 1979 throughout 2015 with a consistent set of frequencies at 19\_-GHz and 37\_-GHz. Conversely to other CDRs (e.g. Meier et al. 2017 and its two algorithm components Bootstrap and NasaTeam), the channels around 22\_-GHz are not used for filtering water vapour contamination. The 23.0\_-GHz channels of the SMMR instrument were highly unstable since launch, and eventually ceased to function ion March 1985. This is one of the reasons why the Meier et al. (2017) dataset only starts with SSM/I F08 on\_09 July 1987 as a fully-qualified CDR (according to https://nsidc.org/data/g02202). A key asset of the algorithms we adopted is that they are self-tuning and self-optimizing to the data, which greatly helps achieving temporal consistency between different satellite missions, both in the past and future (discussed later as outlook section 5.2).

The self-tuning and self-optimizing algorithms also\_allowed e.g. to consistently processconsistent processing of 
[revised manuscript text omitted]
 TB channels than 19–GHz and 37–GHz. Finally, research is needed to assign a true spatial resolution to SIC fields computed from combinations of p TB channels, themselves at different spatial resolutions. Some knowledge is embedded in our parametrization of  $\sigma_{gmean}$ , but it is currently not enough to e.g. choose and fully justify a grid spacing for SIC data records. In any case and even after almost 40–years of routinely available passive microwave observations of the polar regions, the underlying algorithms can still be improved to yield improved accuracy and there is scope for continued research and development in the field.
- Formatted: Font: Italic

Other steps in the processing chain can further be improved upon, e.g. the land spill-over correction schemes. In section 3.6 we described how land spill-over was corrected in two steps, first through a physically-based algorithm on swath  $T_{\underline{B}}$  data (adapted from Maass and Kaleschke, 2010), followed by a statistically-based correction of gridded SICs (adapted from

Cavalieri et al. 1999). Several reasons can have led to the swath-based correction to not be enough. For example, the method relies heavily on accurate geolocation of the  $T_{B}$  measurements, however its uncertainty for the SSM/I and SSMIS instrument is known to be large (Poe et al. 2008), and is not corrected for in the current version of the FCDR (R3) we used (Fennig et al. 2017). We used approximated iFoVs weighting functions instead of eFoVs (see section 2.1) when convolving antenna

[revised manuscript text omitted]

Cavalieri, D. J., Crawford, J., Drinkwater, M., Emery, W. J., Eppler, D. T., Farmer, L. D., Goodberlet, M., Jentz, R., Milman, A., Morris, C., Onstott, R., Schweiger, A., Shuchman, R., Steffen, K., Swift, C. T., Wackerman, C., and Weaver, R. L.: NASA sea ice validation program for the DMSP SSM/I: final report. NASA Technical Memorandum 104559. National

30 Aeronautics and Space Administration, Washington, D.C. 126 pages, 1992

Cavalieri, D. J.: A microwave technique for mapping thin sea ice. J. Geophys. Res., 99:12561-12572, 1994

| -  | Formatted: Font: Not Italic |
|----|-----------------------------|
| -1 | Formatted: Font: Italic     |
| -{ | Formatted: No underline     |
|    |                             |

|    | Cavalieri, D. J., St. Germain, K. M., and Swift, C. T.: Reduction of weather effects in the calculation of sea ice concentration |                           |
|----|----------------------------------------------------------------------------------------------------------------------------------|---------------------------|
|    | with the DMSP SSM/I. Journal of Glaciology 41(139): 455-464, 1995                                                                | Formatted: Font: Not Bold |
|    |                                                                                                                                  |                           |
|    | Cavalieri, D. J., Parkinson, C. L., Gloersen, P., Comiso, J. C., and Zwally, H. J.: Deriving long-term time series of sea ice    |                           |
| 5  | cover from satellite passive-microwave multisensor data sets, J. Geophys. Res., 104(C7), 15803-15814,                            |                           |
|    | doi:10.1029/1999JC900081, 1999                                                                                                   | Formatted: No underline   |
|    |                                                                                                                                  |                           |
|    | Colton, M. C., and Poe, G. A.: Intersensor calibration of DMSP SSM/Is: F-8 to F-14, 1987-1997, IEEE Trans. Geosci.               |                           |
|    | Remote Sens 37 418-439 1999                                                                                                      |                           |
| 10 | Renote Sensi, 57, 110-157, 1777                                                                                           |                           |
| 10 | Comise I. C. Characteristics of arctic winter see ice from satellite multiconstral microways observations. Journal of            |                           |
|    | Combined December 01(C1) 075 004 1096                                                                                            |                           |
|    | Geophysical Research 91(C1), 973-994, 1980                                                                                |                           |
|    |                                                                                                                                  |                           |
|    | Comiso, J. C., and Nishio, F.: Trends in the sea ice cover using ennanced and compatible AMSR-E, SSM/I, and SMMR data,           |                           |
| 15 | J. Geophys. Res., 113, C02S07, doi:10.1029/2007JC004257, 2008                                                                    | Formatted: No underline   |
|    |                                                                                                                                  |                           |
|    | Comiso, J. C., Gersten, R.A., Stock, L.V., Turner, J., Perez, G.J., and Cho, K.: Positive Trend in the Antarctic Sea Ice Cover   |                           |
|    | and Associated Changes in Surface Temperature. J. Climate, 30, 2251–2267, doi:10.1175/JCLI-D-16-0408.1, 2017                     | Formatted: No underline   |
|    |                                                                                                                                  |                           |
| 20 | Comiso, J.C.: Large Decadal Decline of the Arctic Multiyear Ice Cover. J. Climate, 25, 1176–1193, doi:10.1175/JCLI-D-11-         |                           |
|    | 00113.1, 2012                                                                                                                    |                           |
|    |                                                                                                                                  |                           |
|    | Comiso, J. C., Meier, W. N., and Gersten, R.: Variability and trends in the Arctic Sea ice cover: Results from different         |                           |
|    | techniques, J. Geophys. Res. Oceans, 122, 6883-6900, 2017                                                                        |                           |
| 25 |                                                                                                                                  |                           |
|    | Dee, D. P., Uppala, S. M., Simmons, A. J., Berrisford, P., Poli, P., Kobayashi, S., Andrae, U., Balmaseda, M. A., Balsamo,       |                           |
|    | G., Bauer, P., Bechtold, P., Beljaars, A. C. M., van de Berg, L., Bidlot, J., Bormann, N., Delsol, C., Dragani, R., Fuentes, M., |                           |
|    | Geer, A. J., Haimberger, L., Healy, S. B., Hersbach, H., Hólm, E. V., Isaksen, L., Kållberg, P., Köhler, M., Matricardi, M.,     |                           |
|    | McNally, A. P., Monge-Sanz, B. M., Morcrette, JJ., Park, BK., Peubey, C., de Rosnay, P., Tavolato, C., Thépaut, JN.              |                           |
| 30 | and Vitart, F.: The ERA-Interim reanalysis: configuration and performance of the data assimilation system. Q.J.R. Meteorol.      |                           |
|    | Soc., 137: 553–597, 2011                                                                                                  | Formatted: No underline   |
|    |                                                                                                                                  |                           |

Donlon, C. J., Martin, M., Stark, J. D., Roberts-Jones, J., Fiedler, E. and Wimmer, W.: The Operational Sea Surface Temperature and Sea Ice analysis (OSTIA). Remote Sensing of the Environment. doi:10.1016/j.rse.2010.10.017, 2012

|    | Fennig, K., Schröder, M., Hollmann, R.: Fundamental Climate Data Record of Microwave Imager Radiances, Edition 3.                                                                                                                                                                                                                                                                            |                         |
|----|----------------------------------------------------------------------------------------------------------------------------------------------------------------------------------------------------------------------------------------------------------------------------------------------------------------------------------------------------------------------------------------------|-------------------------|
|    | Satellite Application Facility on Climate Monitoring. doi:10.5676/EUM_SAF_CM/FCDR_MWI/V003_2017                                                                                                                                                                                                                                                                                              | Formatted: No underline |
| 5  | GCOS-IP: GCOS Implementation Plan 2016. GCOS-200. Available at
https://library.wmo.int/opac/doc_num.php?explnum_id=3417, 2016                                                                                                                                                                                                                                                             |                         |
|    | Ghaffari P. Pedersen I. Eastwood S. Lavergne T. Sea Ice in the Caspian Sea Technical Report OSI AS11 P04                                                                                                                                                                                                                                                                                     |                         |
|    | available at https://ocieaf.mat.no. 2011                                                                                                                                                                                                                                                                                                                                                     | Parmathada Marundarilar |
| 10 | available at https://osisal.inet.no, 2011                                                                                                                                                                                                                                                                                                                                             | Formatted: No underline |
| 10 | Gregory, J. M., Stott, P. A., Cresswell, D. J., Rayner, N. A., Gordon, C., and Sexton, D. M. H.: Recent and Future Changes
in Arctic Sea Ice Simulated by the HadCM3 AOGCM, Geophys. Res. Lett. 29, no. 24, 2175, doi:10.1029/2001GL014575,
2002                                                                                                                                       |                         |
| 15 | Herrington, T., and Zickfeld, K.: Path independence of climate and carbon cycle response over a broad range of cumulative carbon emissions, Earth Syst. Dyn. 5, 409–422, doi:10.5194/esd-5-409-2014, 2014                                                                                                                                                                                    |                         |
| 20 | Hersbach, H. and Dee, D.: ERA5 reanalysis is in production, ECMWF Newsletter, Number 147 - Spring 2016
https://www.ecmwf.int/en/newsletter/147/news/era5-reanalysis-production, 2016                                                                                                                                                                                                      |                         |
|    | Hobbs, W., Massom, R., Stammerjohn, S., Reid, P., Williams, G., and Meier, W.: A review of recent changes in Southern
Ocean sea ice, their drivers and forcings, Global and Planetary Change, Volume 143, doi:10.1016/j.gloplacha.2016.06.008,
2016                                                                                                                                    |                         |
| 25 | Ivanova, N., Pedersen, L. T., Tonboe, R. T., Kern, S., Heygster, G., Lavergne, T., Sørensen, A., Saldo, R., Dybkjær, G.,
Brucker, L., and Shokr, M.: Inter-comparison and evaluation of sea ice algorithms: towards further identification of
challenges and optimal approach using passive microwave observations, The Cryosphere, 9, 1797-1817, doi:10.5194/tc-9-
1797-2015, 2015 |                         |
| 30 | Johannessen, O. M.: Decreasing Arctic Sea Ice Mirrors Increasing CO2 on Decadal Time Scale, Atmospheric and Oceanic Science Letters 1, no. 1, 51–56. doi:10.1080/16742834.2008.11446766, 2008                                                                                                                                                                                                |                         |

|    | Kaminski, T. and Mathieu, PP.: Reviews and syntheses: Flying the satellite into your model: on the role of observation     |                           |
|----|----------------------------------------------------------------------------------------------------------------------------|---------------------------|
|    | operators in constraining models of the Earth system and the carbon cycle. Biogeosciences, 14, 2343-2357, doi:10.5194/be-  |                           |
|    | 14-2343-2017. 2017                                                                                                         |                           |
|    |                                                                                                                            |                           |
| 5  | Kern S Rösel A Pedersen I T Ivanova N Saldo R and Tonboe R T. The impact of melt ponds on summertime                       |                           |
| 5  | migroupus brightness temperatures and see ice concentrations. The Cruesphere, 10, 2217-2220, doi:10.5104/to.10.2217        |                           |
|    | incrowave originaless temperatures and sea-ice concentrations, the Cryosphere, 10, 2217-2259, doi:10.5194/tc-10-2217-      |                           |
|    | 2016, 2016                                                                                                                 |                           |
|    |                                                                                                                            |                           |
|    | Korosov, A. A., Rampal, P., Pedersen, L. T., Saldo, R., Ye, Y., Heygster, G., Lavergne, T., Aaboe, S., and Girard-Ardhuin, |                           |
| 10 | F.: A new tracking algorithm for sea ice age distribution estimation, The Cryosphere, 12, 2073-2085, doi:10.5194/tc-12-    |                           |
|    | 2073-2018, 2018                                                                                                            |                           |
|    |                                                                                                                            |                           |
|    | Kwok, R.: Sea ice concentration estimates from satellite passive microwave radiometry and openings from SAR ice motion,    |                           |
|    | Geophys. Res. Lett., 29(9), 1311, doi:10.1029/2002GL014787, 2002                                                           |                           |
| 15 |                                                                                                                            |                           |
|    | Lavergne, T., Eastwood, S., Teffah, Z., Schyberg, H., and Breivik, L